# TOWARDS MINIMAX OPTIMAL REWARD-FREE REINFORCEMENT LEARNING IN LINEAR MDPS

**Pihe Hu\*, Yu Chen\*, Longbo Huang**$^{†}$
Institute for Interdisciplinary Institute for Interdisciplinary Information Sciences
Tsinghua University, Beijing, China
`{hph19,c-y19}@mails.tsinghua.edu.cn, longbohuang@tsinghua.edu.cn`

## ABSTRACT

We study reward-free reinforcement learning with linear function approximation for episodic Markov decision processes (MDPs). In this setting, an agent first interacts with the environment without accessing the reward function in the exploration phase. In the subsequent planning phase, it is given a reward function and asked to output an $\epsilon$-optimal policy. We propose a novel algorithm LSVI-RFE under the linear MDP setting, where the transition probability and reward functions are linear in a feature mapping. We prove an $\widetilde{O}(H^4 d^2/\epsilon^2)$ sample complexity upper bound for LSVI-RFE, where $H$ is the episode length and $d$ is the feature dimension. We also establish a sample complexity lower bound of $\Omega(H^3 d^2/\epsilon^2)$. To the best of our knowledge, LSVI-RFE is the first computationally efficient algorithm that achieves the minimax optimal sample complexity in linear MDP settings up to an $H$ and logarithmic factors. Our LSVI-RFE algorithm is based on a novel variance-aware exploration mechanism to avoid overly-conservative exploration in prior works. Our sharp bound relies on the decoupling of UCB bonuses during two phases, and a Bernstein-type self-normalized bound, which remove the extra dependency of sample complexity on $H$ and $d$, respectively.

## 1 INTRODUCTION

In reinforcement learning (RL), an agent tries to learn an optimal policy that maximizes the cumulative long-term rewards by interacting with an unknown environment. Designing efficient exploration mechanisms, being a critical task in RL algorithm design, is of great significance in improving the sample efficiency of RL, both theoretically Azar et al. (2017); Ménard et al. (2021) and empirically Schwarzer et al. (2020); Ye et al. (2021). In particular, for scenarios where reward signals are sparse and require manually-designed reward functions, e.g., Nair et al. (2018); Riedmiller et al. (2018), or multi-task settings where RL agents are required to accomplish different goals in different stages, e.g., Hessel et al. (2019); Yang et al. (2020), efficient exploration of the environments is crucial, as it can avoid the agent from repeated learning under different reward functions, resulting in inefficiency and even intractability of sample complexity. However, the theoretical understanding is still limited, especially for MDPs with large (or infinite) states or action spaces.

To understand the exploration mechanism in RL, *reward-free exploration* (RFE) is firstly proposed in Jin et al. (2020a) to explore the environment without reward signals. RFE contains two phases: exploration and planning. In the exploration phase, the agent first interacts with the environment without accessing the reward function. In the subsequent planning phase, the agent is given a reward function and asked to output an $\epsilon$-optimal policy. RFE has great significance in a host of reinforcement learning applications, e.g., multi-task RL Hessel et al. (2019); Yang et al. (2020), RL with sparse rewards Nair et al. (2018); Riedmiller et al. (2018), and systematic generalization of RL Jiang et al. (2019); Mutti et al. (2022). The minimax optimal sample complexity $O(H^3 S^2 A/\epsilon^2)$ of RFE is obtained in Ménard et al. (2021) for tabular settings where $S$ and $A$ are sizes of state and action space, respectively. However, this bound is intractable when state and action space are large.

In this paper, we consider the RFE problem in the linear MDP setting, where the transition probability and the reward function are linear in a feature mapping $\phi(\cdot, \cdot)$. The linear MDP model is an

---

\*Equal contribution.
$^{†}$Corresponding author.

important formulation that provides a linear function approximation for general MDP problems with large (or infinite) state and action space, and has received much recent attention in RL studies, e.g., Jin et al. (2020b); Zhou et al. (2021). Existing works in Wang et al. (2020a); Zanette et al. (2020c); Chen et al. (2021) also study RFE in linear MDPs. The best known sample complexity upper bound in this setting is $\widetilde{O}(H^4 d^3/\epsilon^2)$ obtained in Chen et al. (2021), and the best known lower bound is $\Omega(\max\{H^3 d, d^2\}/\epsilon^2)$ by combining results obtained in Zhang et al. (2021) and Wagenmaker et al. (2022). This means that the following fundamental question still remains open:

*Does there exist a computation-efficient and minimax optimal algorithm for RFE in linear MDPs?*

We make constructive contributions to minimax optimality by proposing a computationally efficient algorithm LSVI-RFE, achieving the state-of-the-art sample complexity upper bound of $\widetilde{O}(H^4 d^2/\epsilon^2)$, which is minimax optimal up to an $H$ and logarithmic factors. The LSVI-RFE algorithm is based on a novel variance-aware exploration mechanism with weighted linear regression to avoid overly-conservative exploration in prior works. Accordingly, our sharp bound relies on the decoupling of UCB bonuses during two phases, and a Bernstein-type self-normalized bound, which remove the extra dependency of sample complexity on $H$ and $d$, respectively. We summarize the main contributions of the paper below.

- We propose the LSVI-RFE algorithm for RFE in linear MDPs based on a novel variance-aware exploration mechanism with weighted linear regression to avoid overly-conservative exploration in prior works. It also builds the monotonicity of constructed value functions between two phases, providing new reward-free linear RL techniques.

- LSVI-RFE achieves a sample complexity of $\widetilde{O}(H^4 d^2/\epsilon^2)$, which relies on the decoupling of UCB bonuses during two phases. In addition, a Bernstein-type self-normalized bound and the conservatism of elliptical potentials are utilized to reach optimal dependency of sample complexity on $d$, which is potentially a general tool for linear reward-free RL.

- We prove a sample complexity lower bound of $\Omega(H^3 d^2/\epsilon^2)$, showing that LSVI-RFE is the first computationally efficient algorithm to achieve the minimax optimal sample complexity in linear MDPs up to an $H$ and logarithmic factors.

**Notations** Scalars are denoted in lower case letters, and vectors/matrices are denoted in boldface letters. Denote $\|\boldsymbol{x}\|_{\boldsymbol{\Lambda}}^2 = \boldsymbol{x}^\top \boldsymbol{\Lambda} \boldsymbol{x}$ for vector $\boldsymbol{x}$ and positive definite matrix $\boldsymbol{\Lambda}$, and $\|\cdot\|_F$ denotes the Frobenius norm. Denote $\{1, ..., n\}$ as $[n]$. $a_n = O(b_n)$ if there exists an absolute constant $c > 0$ such that $a_n \leqslant c b_n$ holds for all $n \geqslant 1$ and $a_n = \Omega(b_n)$ for inverse direction. $\widetilde{O}(\cdot)$ further suppresses the polylogarithmic factors in $O(\cdot)$. $\lesssim$ denotes approximately less than up to constant factors.

## 2 RELATED WORKS

**RL with Linear Function Approximation** There are a number of ways to parameterize an MDP linearly. The first sample efficient algorithm is introduced by Jiang et al. (2017), where low Bellman rank MDPs are considered. Subsequent works include Dann et al. (2018); Sun et al. (2019). Yang & Wang (2019) develops the first statistically and computationally efficient algorithm for linear MDPs with generative models, while Jin et al. (2020b) considers RL settings and proposes the LSVI-UCB algorithm. Concurrently, Zanette et al. (2020a) provides a Thompson sampling-based algorithm. He et al. (2021); Wagenmaker et al. (2021) provide a gap-dependent regret bound and a first-order regret bound for linear MDPs, respectively. Subsequently, the minimax optimal algorithm is proposed in Hu et al. (2022). More works on RL with linear function approximation include Zanette et al. (2020b) for low inherent bellman error case, Wang et al. (2020c) for linear Q function case, and Wang et al. (2020b) for bounded Eluder dimension case. Another popular linearly parameterized MDP is the linear mixture MDP, studied in Modi et al. (2020); Yang & Wang (2020); Jia et al. (2020); Ayoub et al. (2020); Cai et al. (2020); Zhou et al. (2021).

**Reward-Free Exploration in Tabular MDPs** Reward-free exploration is studied in Jin et al. (2020a); Kaufmann et al. (2021); Ménard et al. (2021); Zhang et al. (2020). Jin et al. (2020a) achieves an $\widetilde{O}(H^5 S^2 A/\epsilon^2)$ sample complexity in the tabular setting. Subsequently, the RF-UCRL algorithm in Kaufmann et al. (2021) improves this result by an $H$ factor. Ménard et al. (2021) modifies the Upper Confidence Bound (UCB)-bonus of UCRL and achieves a sample complexity of $\widetilde{O}(H^3 S^2 A/\epsilon^2)$, matching the minimax lower bound provided in Jin et al. (2020a) up to logarithmic factors. Concurrently, the SSTP algorithm Zhang et al. (2020) also achieves minimax optimality in the time-homogeneous setting. Besides, Wu et al. (2022) proposes the first gap dependent bound.

**RFE with Linear Function Approximation** There are recent works Wang et al. (2020a); Zanette et al. (2020c); Zhang et al. (2021); Chen et al. (2021); Huang et al. (2022); Wagenmaker et al. (2022) focusing on RFE in RL with linear function approximation. Chen et al. (2021) gives a sample complexity bound of $\widetilde{O}(H^4 d^3/\epsilon^2)$ which is sharpest result on $H$, while Wagenmaker et al. (2022) gives a sample complexity bound of $\widetilde{O}(H^5 d^2/\epsilon^2)$, which achieves optimal dependency on $d$. Our technical framework, i.e., an aggressive variance-aware exploration mechanism, is very different from that in Wagenmaker et al. (2022), and also achieves an optimal dependency on $d$ and a better dependency on $H$. The best known lower bound is $\Omega(\max\{H^3 d, d^2\}/\epsilon^2)$ by combining results obtained in Zhang et al. (2021) and Wagenmaker et al. (2022). Another line of works Zhang et al. (2021); Chen et al. (2021) focus on RFE in linear mixture MDPs, where the minimax optimal sample complexity is obtained in Chen et al. (2021) when $d > H$ but the algorithm is not computationally-efficient. A detailed comparison of some most related works is shown in Table 1.[1] Moreover, low-rank MDPs, which subsume linear MDPs, are considered in Modi et al. (2021); Chen et al. (2022). In addition, more related work focuses on block MDPs, where the representation $\phi$ is unknown, Du et al. (2019); Misra et al. (2020); Zhang et al. (2022).

Table 1: Comparison of reward-free exploration in episodic RL with linear function approximation.

| Setting | Algorithm | Computation-Efficient | Sample Complexity |
|---|---|---|---|
| Linear Mixture MDP | UCRL-RFE$^+$ Zhang et al. (2021) | No | $\widetilde{O}(H^5 d(H+d)/\epsilon^2)$ |
| | Chen et al. Chen et al. (2021) | No | $\widetilde{O}(H^3 d(H+d)/\epsilon^2)$ |
| | *Lower bound* Chen et al. (2021) | | $\Omega(H^3 d^2/\epsilon^2)$ |
| Linear MDP | Wang et al. Wang et al. (2020a) | Yes | $\widetilde{O}(H^6 d^3/\epsilon^2)$ |
| | FRANCIS Zanette et al. (2020c) | Yes | $\widetilde{O}(H^5 d^3/\epsilon^2)$ |
| | Chen et al. Chen et al. (2021) | Yes | $\widetilde{O}(H^4 d^3/\epsilon^2)$ |
| | RFLIN Wagenmaker et al. (2022) | Yes | $\widetilde{O}(H^5 d^2/\epsilon^2)$ |
| | **LSVI-RFE** (Alg. 1, 2) | Yes | $\widetilde{O}(H^4 d^2/\epsilon^2)$ |
| | **Lower bound** (Th. 6.1) | | $\Omega(H^3 d^2/\epsilon^2)$ |

## 3 PRELIMINARIES

We consider an episodic finite-horizon MDP $\mathcal{M} = \{\mathcal{S}, \mathcal{A}, H, \{\mathbb{P}_h\}_h, \{r_h\}_h\}$, where $\mathcal{S}$ is the state space, $\mathcal{A}$ is the action space, $H \in \mathbb{Z}^+$ is the episode length, $\mathbb{P}_h : \mathcal{S} \times \mathcal{A} \to \Delta(\mathcal{S})$ and $r_h : \mathcal{S} \times \mathcal{A} \to [0,1]$ are time-dependent transition probability and deterministic reward function. We assume that $\mathcal{S}$ is a measurable space with a possibly infinite number of elements and $\mathcal{A}$ is a finite set.

For a time-inhomogeneous MDP, the policy is time-dependent, which is denoted as $\pi = \{\pi_1, ..., \pi_H\}$, where $\pi_h(s)$ is the action that agent takes at state $s$ at the $h$-th step. We define the state-action function (i.e., $Q$-function) and value function as

$$Q_h^\pi(s, a; r) = \mathbb{E}\left[\sum_{h'=h}^{H} r_{h'}(s_{h'}, a_{h'}) \mid s_h = s, a_h = a, \pi\right], \tag{1}$$

$$V_h^\pi(s; r) = \mathbb{E}\left[\sum_{h'=h}^{H} r_{h'}(s_{h'}, a_{h'}) \mid s_h = s, \pi\right], \tag{2}$$

respectively for a specific set of the reward function $r = \{r_h\}_{h=1}^{H}$. For any function $V(\cdot; r) : \mathcal{S} \to \mathbb{R}$, we further denote $\mathbb{P}_h V(s, a; r) = \mathbb{E}_{s' \sim \mathbb{P}_h(\cdot|s,a)} V(s'; r)$ and value function variance $[\mathbb{V}_h V](s, a; r) = \mathbb{P}_h V^2(s, a; r) - [\mathbb{P}_h V(s, a; r)]^2$, where $V^2$ stands for the function whose value at $s$ is $V^2(s; r)$. The Bellman equation associated with a policy $\pi$ for reward function $r$ is

$$Q_h^\pi(s, a; r) = r_h(s, a) + \mathbb{P}_h V_{h+1}^\pi(s, a; r), \quad V_h^\pi(s; r) = Q_h^\pi(s, \pi_h(s); r), \tag{3}$$

for any $(s, a) \in \mathcal{S} \times \mathcal{A}$ and $h \in [H]$. Since the action space and the episode length are both finite, there always exists an optimal policy $\pi^*$ for the reward function $r = \{r_h\}_{h=1}^{H}$, such that the associated optimal state-action function and value function are $Q_h^*(s, a; r) = \sup_\pi Q_h^\pi(s, a; r)$

---

[1] For time-inhomogenous case, the bound of UCRL-RFE$^+$ Zhang et al. (2021) is degraded by an $H$ factor.

and $V_h^*(s; r) = \sup_\pi V_h^\pi(s; r)$, respectively. For any $(s, a) \in \mathcal{S} \times \mathcal{A}$ and $h \in [H]$, the Bellman optimality equation for the reward function $r = \{r_h\}_{h=1}^H$ is

$$Q_h^*(s, a; r) = r_h(s, a) + \mathbb{P}_h V_{h+1}^*(s, a; r), \quad V_h^*(s; r) = \max_{a \in \mathcal{A}} Q_h^*(s, a; r), \tag{4}$$

The structural assumption we make in this paper is a linear structure in both transition and reward, which has been considered in prior works, e.g., Yang & Wang (2019); Jin et al. (2020b) as below:

**Definition 3.1** (Linear MDP). A MDP $\mathcal{M} = \{\mathcal{S}, \mathcal{A}, H, \{\mathbb{P}_h\}_h, \{r_h\}_h\}$ is a linear MDP with a known feature mapping $\phi : \mathcal{S} \times \mathcal{A} \to \mathbb{R}^d$, if for any $h \in [H]$, there exist unknown $d$-dimensional measures $\boldsymbol{\mu}_h = (\mu_h(s)_{s \in \mathcal{S}}) \in \mathbb{R}^{d \times |\mathcal{S}|}$ over and an unknown vector $\boldsymbol{\theta}_h \in \mathbb{R}^d$, such that for any $(s, a) \in \mathcal{S} \times \mathcal{A}$,

$$\mathbb{P}_h(s' \mid s, a) = \langle \phi(s, a), \boldsymbol{\mu}_h(s') \rangle, \ r_h(s, a) = \langle \phi(s, a), \boldsymbol{\theta}_h \rangle.$$

We make the following norm assumptions: for any $h \in [H]$, (i) $\sup_{s,a} \|\phi(s, a)\|_2 \leqslant 1$, (ii) $\|\boldsymbol{\mu}_h \boldsymbol{v}\|_2 \leqslant \sqrt{d}$ for any vector $\boldsymbol{v} \in \mathbb{R}^{|\mathcal{S}|}$ such that $\|\boldsymbol{v}\|_\infty \leqslant 1$, (iii) $\|\boldsymbol{\theta}_h\|_2 \leqslant W$, where $W$ is a constant. These assumptions are mild and are common in the existing literature Jin et al. (2020b).

**Reward-Free Exploration (RFE)**  We consider the following RFE model, which has been considered in previous literature Jin et al. (2020a); Kaufmann et al. (2021); Ménard et al. (2021); Zhang et al. (2020). Specifically, there are two phases in the RFE paradigm. (i) *Exploration Phase*: The agent interacts with the environment for exploration up to $K$ episodes without accessing the reward function. (ii) *Planning Phase*: The agent is given a reward function $r = \{r_h\}_{h=1}^H$ with the goal of outputting an $\epsilon$-optimal policy $\pi$ based on learned information from the exploration phase.

We define the *sample complexity* to be the number of episodes $K$ required in the exploration phase to output an $\epsilon$-optimal policy $\pi$ in the planning phase for any possible reward function $r$, i.e.,

$$\mathbb{E}_{s \sim \mu} [V_1^*(s; r)] - \mathbb{E}_{s \sim \mu} [V_1^\pi(s; r)] \leqslant \epsilon$$

where $\mu \in \Delta(\mathcal{S})$ denotes the initial state distribution.

## 4 ALGORITHM AND MAIN RESULTS

This section presents our LSVI-RFE algorithm for reward-free reinforcement learning in linear MDPs. It builds upon the procedure of optimistic learning as Wang et al. (2020a); Zhang et al. (2021), but with critical novelty of introducing an aggressive variance-aware exploration mechanism. The mechanism is inspired by Chen et al. (2021); Hu et al. (2022), and LSVI-RFE further makes critical improvements in variance-aware weights, value function monotonicity, and computational tractability. In addtion, the mechanism is implemented by an aggressive exploration bonus $b_{k,h}$ and an aggressive reward function $r_{k,h}$ in the exploration phase: (i) The aggressive exploration bonus guarantees the monotonicity of value functions between two phases and removes the additional dependency of sample complexity on feature dimension $d$, due to building a uniform convergence argument by the covering net in prior works, e.g., Jin et al. (2020b); Wang et al. (2020a). (ii) The reward function is also more aggressive than those in prior works Wang et al. (2020a); Zanette et al. (2020c) by an $H$ factor to avoid overly-conservative exploration, which removes the extra dependency of sample complexity on episode length $H$.

### 4.1 EXPLORATION PHASE

**Overall Exploration Sketch**  The observed state-action pairs are collected in each episode for estimating the parameter $\{\boldsymbol{\mu}_h\}_{h \in [H]}$ by weighted linear regression. Then, the optimistic state-action function $\widehat{Q}_{k,h}$ is constructed (Line 9), and the agent executes the greedy policy $\pi_h^k$ with respect to the optimistic state-action function (Line 11). Two critical steps in the exploration are the variance-aware exploration mechanism and the weighted linear regression, which are illustrated below.

**Aggressive Variance-Aware Exploration Mechanism**  LSVI-RFE designs a variance-aware weight $\widehat{\sigma}_{k,h}$, and subsequently builds an exploration bonus $b_{k,h}$ and a reward function $r_{k,h}$ to encourage the exploration in Lines 7 and 8 of Algorithm 1, respectively. Our exploration mechanism is variance-aware and aggressive with the following critical differences to prior works:

(i) *Variance-aware weights:* $\widehat{\sigma}_{k,h}$ (Line 24 of Algorithm 1) contains two terms: $w_{k,h}$ and $W_{k,h}$. The motivation to introduce $\widehat{\sigma}_{k,h}$ remains that we utilize a Bernstein-type self-normalized bound (Lemma C.2 in Appendix C.1) to build the confidence set, and the Bernstein bound contains a

variance term ($\sigma^2$) and elliptical potential term ($R$). Thus, LSVI-RFE estimates the variance upper bound of the value function $\widehat{V}_{k,h}$ by $W_{k,h}$ in Line 16, and dynamically adjusts $w_{k,h}$ in Lines 19-23 to keep $\|\widehat{\sigma}_{k,h}^{-1}\phi(s_h^k, a_h^k)\|_{\widehat{\Lambda}_{k,h}^{-1}}/\widehat{\sigma}_{k,h}$ small. This fine-grained control of $w_{k,h}$ is critical for removing the extra dependency on $d$, which is inspired by Hu et al. (2022) for regret minimization in linear MDPs, and is detailed in step 2 in Section 5. However, $\widehat{\sigma}_{k,h}$ in Hu et al. (2022) contains three terms and the difference is that we abandon the variance upper estimator term concerning the real value function. Due to the agnosticism of the reward function in the exploration phase, a uniform upper bound of the value function (concerning the actual reward function) variance is infeasible.

---

**Algorithm 1** Least-Squares Value Iteration - RFE (LSVI-RFE): Exploration Phase

---

**Require:** Regularization parameter $\lambda$, exploration radius $\widehat{\beta}^E$
1: **for** step $h = H, ..., 1$ **do**
2: $\quad \widehat{\Lambda}_{1,h}, \widetilde{\Lambda}_{1,h} \leftarrow \lambda\mathbf{I}; \widehat{\boldsymbol{\mu}}_{1,h} \leftarrow \mathbf{0}; \widehat{V}_{0,h}(\cdot) \leftarrow H;$
3: **end for**
4: $\widehat{V}_{1,H+1}(\cdot) \leftarrow 0$
5: **for** episode $k = 1, ..., K$ **do**
6: $\quad$ **for** step $h = H, ..., 1$ // *Value iteration* **do**
7: $\qquad b_{k,h}(\cdot, \cdot) = 2\widehat{\beta}^E \|\phi(\cdot, \cdot)\|_{\widehat{\Lambda}_{k,h}^{-1}}$ // *Exploration driven bonus*
8: $\qquad r_{k,h}(\cdot, \cdot) = b_{k,h}(\cdot, \cdot)/2$ // *Exploration driven reward function*
9: $\qquad \widehat{Q}_{k,h}(\cdot, \cdot) = r_{k,h}(\cdot, \cdot) + \langle\widehat{\boldsymbol{\mu}}_{k,h}\widehat{\boldsymbol{V}}_{k,h+1}, \phi(\cdot, \cdot)\rangle + b_{k,h}(\cdot, \cdot)$ // *Optimistic Q function*
10: $\qquad \widehat{V}_{k,h}(\cdot) \leftarrow \min\left\{\max_{a\in\mathcal{A}} \widehat{Q}_{k,h}(\cdot, a), H\right\}$
11: $\qquad \pi_h^k(\cdot) \leftarrow \arg\max_{a\in\mathcal{A}} \widehat{Q}_{k,h}(\cdot, a)$
12: $\quad$ **end for**
13: $\quad$ Receive the initial state $s_1^k$
14: $\quad$ **for** step $h = 1, ..., H$ **do**
15: $\qquad a_h^k \leftarrow \pi_h^k(s_h^k)$, and observe $s_{h+1}^k \sim \mathbb{P}_h(\cdot|s_h^k, a_h^k)$
16: $\qquad W_{k,h} = \min\{H \cdot (\langle\widehat{\boldsymbol{\mu}}_{k,h}\widehat{\boldsymbol{V}}_{k,h+1}, \phi(s_h^k, a_h^k)\rangle + \widehat{\beta}^E\|\phi(s_h^k, a_h^k)\|_{\widehat{\Lambda}_{k,h}^{-1}} + H\sqrt{\lambda}/2K\sqrt{d}), H^2\}$
17: $\qquad \widetilde{\sigma}_{k,h} \leftarrow \sqrt{\max\{H, (d^2/H)W_{k,h}\}}$
18: $\qquad \widetilde{\Lambda}_{k+1,h} \leftarrow \widetilde{\Lambda}_{k,h} + \widetilde{\sigma}_{k,h}^{-2}\phi(s_h^k, a_h^k)\phi(s_h^k, a_h^k)^\top$
19: $\qquad$ **if** $\|\widetilde{\sigma}_{k,h}^{-1}\phi(s_h^k, a_h^k)\|_{\widetilde{\Lambda}_{k,h}^{-1}} > 1/d^3$ **then**
20: $\qquad\quad w_{k,h} \leftarrow \sqrt{Hd^3}$
21: $\qquad$ **else**
22: $\qquad\quad w_{k,h} \leftarrow \sqrt{H}$
23: $\qquad$ **end if**
24: $\qquad \widehat{\sigma}_{k,h} \leftarrow \max\{w_{k,h}, \widetilde{\sigma}_{k,h}\}$
25: $\qquad \widehat{\Lambda}_{k+1,h} \leftarrow \widehat{\Lambda}_{k,h} + \widehat{\sigma}_{k,h}^{-2}\phi(s_h^k, a_h^k)\phi(s_h^k, a_h^k)^\top$
26: $\qquad \widehat{\boldsymbol{\mu}}_{k+1,h} \leftarrow \widehat{\Lambda}_{k+1,h}^{-1}\sum_{i=1}^k \widehat{\sigma}_{i,h}^{-2}\phi(s_h^i, a_h^i)\boldsymbol{\delta}(s_{h+1}^i)^\top$ // *Solution to the weighted regression*
27: $\quad$ **end for**
28: **end for**
29: Return $\{\widehat{\Lambda}_{K+1,h}, \widehat{\boldsymbol{\mu}}_{K+1,h}\}_{h\in[H]}$.

---

(ii) *Aggressive exploration bonus $b_{k,h}$:* It is aggressive by a factor of 2 enlargement to ensure the monotonicity of the estimated value function between the exploration phase and planning phase, i.e., $V_h^*(\cdot; r) + \widehat{V}_{k,h}(\cdot) \geqslant \widehat{V}_h(\cdot)$ (Lemma A.15 in Appendix), which is also necessary for removing the extra dependency of the sample complexity on $d$. This monotonicity is similar to "over-optimism" in Hu et al. (2022), but our monotonicity is built between the exploration phase and the planning phase, which is different from that in Hu et al. (2022), which builds monotonicity in each episode. The reason remains that only the planning phase bonus determines the sharpness of the sample complexity as detailed in Section 5, such that monotonicity is only required between the exploration phase and the planning phase, instead of each episode in the exploration phase.

(iii) *Aggressive reward function $r_{k,h}$:* It is more aggressive by a factor of $H$ enlargement than existing works for RFE in linear MDPs, e.g., Wang et al. (2020a), which sets the reward function as $\min\{b_{k,h}(\cdot, \cdot)/H, 1\}$ so that it belongs to $[0, 1]$. $r_{k,h}$ takes the same order as the exploration bonus,

i.e., without the $1/H$ factor. Using a factor of $H$ enlargement to achieve faster learning rates was firstly proposed in Chen et al. (2021) for linear mixture MDPs, achieving minimax optimal sample complexity, yet in a computationally-inefficient manner. We prove that the $H$ enlargement also works for our variance-aware exploration mechanism in a computationally-efficient way, and saves an $H^2$ factor in the sample complexity.

**Weighted Linear Regression** To enable aggressive variance-aware exploration, we employ weighted linear regression to assemble variance-aware weights in linear regression. Note that the weighted ridge regression estimator has been built for regret minimization algorithms for RL with linear function approximations, e.g., Zhou et al. (2021); Hu et al. (2022). Denote $\boldsymbol{\delta}(s) \in \mathbb{R}^{|\mathcal{S}|}$ as a one-hot vector that is zero everywhere except the entry corresponding to state $s$ is one, and define $\boldsymbol{\epsilon}_h^k := \mathbb{P}_h(\cdot \mid s_h^k, a_h^k) - \boldsymbol{\delta}(s_{h+1}^k)$. Considering $\mathbb{E}[\boldsymbol{\epsilon}_h^k \mid s_h^k, a_h^k] = 0$, $\boldsymbol{\delta}(s_{h+1}^k)$ is an unbiased estimate of $\mathbb{P}_h(\cdot \mid s_h^k, a_h^k) = \boldsymbol{\mu}_h^\top \boldsymbol{\phi}(s_h^k, a_h^k)$. Thus, $\boldsymbol{\mu}_h$ can be learned via regression from $\boldsymbol{\phi}(s_h^k, a_h^k)$ to $\boldsymbol{\delta}(s_{h+1}^k)$, and the sequence $\{\widehat{\sigma}_{i,h}\}_{i \in [k]}$ serves as the weight sequence. The estimated parameter $\widehat{\boldsymbol{\mu}}_{k,h}$ in Line 26 of Algorithm 1 is the solution to the following weighted linear regression:

$$\min_{\boldsymbol{\mu} \in \mathbb{R}^{d \times |\mathcal{S}|}} \sum_{i=1}^{k-1} \left\| \left[ \boldsymbol{\mu}_h^\top \boldsymbol{\phi}(s_h^k, a_h^k) - \boldsymbol{\delta}(s_{h+1}^k) \right] \widehat{\sigma}_{i,h}^{-1} \right\|_2^2 + \lambda \|\boldsymbol{\mu}\|_F^2, \tag{5}$$

with solution in Line 26 and the estimated transition probability $\widehat{\mathbb{P}}_{k,h}(\cdot \mid s_h^k, a_h^k) = \widehat{\boldsymbol{\mu}}_{k,h}^\top \boldsymbol{\phi}(s_h^k, a_h^k)$.

## 4.2 PLANNING PHASE

**Overall Planning Sketch** During the planning phase, the $\epsilon$-optimal policy $\pi$ is the greedy policy concerning the optimistic value iteration with respect to the estimated transition matrix from the exploration phase, i.e., parameters $\{\widehat{\boldsymbol{\mu}}_{K+1,h}\}_{h \in [H]}$. We introduce a UCB bonus term $b_h(\cdot, \cdot)$, which still takes a variance-aware mechanism by utilizing the covariance matrix $\widehat{\boldsymbol{\Lambda}}_{K+1,h}$, to ensure optimism of $\widehat{V}_h(\cdot)$. In particular, Step 4 in Section 5 reveals that the sub-optimality gap is upper bounded by the summation of UCB bonus term $b_h(\cdot, \cdot)$ in the planning phase, which is small on average and ensures the near optimality of the returned greedy policy $\pi$.

---

**Algorithm 2** Least-Squares Value Iteration - RFE (LSVI-RFE): Planning Phase

---

**Require:** Planning radius $\widehat{\beta}^P$, parameter $\{\widehat{\boldsymbol{\Lambda}}_{K+1,h}, \widehat{\boldsymbol{\mu}}_{K+1,h}\}_{h \in [H]}$, reward function $r = \{r_h\}_{h \in [H]}$

1: $\widehat{V}_{H+1}(\cdot) \leftarrow 0$
2: **for** step $h = H, ..., 1$ // *Value iteration* **do**
3:     $b_h(\cdot, \cdot) = \min\{\widehat{\beta}^P \|\boldsymbol{\phi}(\cdot, \cdot)\|_{\widehat{\boldsymbol{\Lambda}}_{K+1,h}^{-1}}, H\}$
4:     $\widehat{Q}_h(\cdot, \cdot) = r_h(\cdot, \cdot) + \langle \widehat{\boldsymbol{\mu}}_{K+1,h} \widehat{V}_{h+1}, \boldsymbol{\phi}(\cdot, \cdot) \rangle + b_h(\cdot, \cdot)$ // *Optimistic Q function*
5:     $\widehat{V}_h(\cdot) \leftarrow \min \{ \max_{a \in \mathcal{A}} \widehat{Q}_h(\cdot, a), H \}$
6:     $\pi_h(\cdot) \leftarrow \arg\max_{a \in \mathcal{A}} \widehat{Q}_h(\cdot, a)$
7: **end for**
8: Return $\boldsymbol{\pi} = \{\pi_h\}_{h \in [H]}$

---

*Remark* 4.1. In our analysis, we introduce auxiliary value functions (Definition A.5 in Appendix). Although it is inspired by Chen et al. (2021), we make critical changes, i.e., building optimism in our variance-aware exploration mechanism with a computationally-efficient manner. By auxiliary value functions, we can further decouple UCB bonuses $b_{k,h}$ and $b_h$ of exploration and planning phases, i.e., $b_{k,h}$ and $b_h$ can take different orders, as revealed in Eq. (9) and Eq. (10). In particular, Step 4 in Section 5 further shows that only the planning phase bonus $b_h$ determines the sharpness of the sample complexity. This is very different from prior works Wang et al. (2020a); Zhang et al. (2021), which take the same order of bonuses in two phases. Our heterogenous UCB bonuses in two phases and accompanying auxiliary value functions together are able to reduce an $H^2$ factor and a $d$ factor in the sample complexity, compared to prior work in Wang et al. (2020a).

## 4.3 MAIN RESULTS

The sample complexity of the proposed LSVI-RFE algorithm is given below in Theorem 4.2 with a proof sketch in Section 5. The detailed proof is given in Appendix A.

**Theorem 4.2** (Upper bound). *Set $\widehat{\beta}^E = \widetilde{O}(d\sqrt{H})$ and $\widehat{\beta}^P = \widetilde{O}(\sqrt{H}d)$ [2] . After collecting $\widetilde{O}\left(H^4 d^2/\epsilon^2\right)$ trajectories during exploration phase, with probability $1 - 7\delta$, LSVI-RFE outputs an $\epsilon$-optimal policy for arbitrary reward function satisfying Definition 3.1 during the planning phase.*

*Remark* 4.3. Theorem 4.2 shows that LSVI-RFE achieves the state-of-the-art sample complexity for RFE in linear MDPs. Compared to the RFE algorithms in Wang et al. (2020a); Zanette et al. (2020c); Wagenmaker et al. (2022) for linear MDPs, our sample complexity is sharper in $H$, which comes from the more aggressive reward function. Moreover, the optimal dependency on $d$ is achieved by our variance-aware UCB bonus such that the extra overhead due to building the uniform convergence argument in Wang et al. (2020a); Chen et al. (2021) is removed. Compared to Wagenmaker et al. (2022), which also achieves optimal dependency on $d$, our technical framework, i.e., an aggressive variance-aware exploration mechanism, is very different, and gives better dependency on $H$.

*Remark* 4.4 (Computational Tractability). LSVI-RFE is a computationally efficient algorithm, i.e., it has polynomial space and computation complexities. Besides, it suffices to only analyze the space and computation complexity of Algorithm 1 in the exploration phase, since Algorithm 2 in the planning phase is equivalent to a single-episode run of Algorithm 1. The space and computation complexities (detailed in Appendix D) of Algorithm 1 are $O(d^2 H + d|\mathcal{A}|HK)$ and $O(d^2|\mathcal{A}|HK^2)$, respectively, where $K$ is the number of episodes that Algorithm 1 has run. By Theorem 4.2, $K$ can take the order of $\widetilde{O}\left(H^4 d^2/\epsilon^2\right)$ to output an $\epsilon$-optimal policy with high probability.

## 5 MECHANISM

In this section, we overview the key techniques and ideas used in the analysis of reward-free exploration and the proof of Theorem 4.2. As preliminary steps in optimistic learning, we construct the confidence sets $\widehat{\mathcal{C}}_{k,h}^E$ and $\widehat{\mathcal{C}}_h^P$ for the exploration phase and planning phase respectively in Steps 1 and 2. Subsequently, we bound the exploration error, i.e., summation of the exploration bonus, during the exploration phase based on the confidence set $\widehat{\mathcal{C}}_{k,h}^E$ in Step 3. Finally, we bound the sub-optimality gap of the recovered policy in the planning phase with confidence set $\widehat{\mathcal{C}}_h^P$ and the monotonicity of the exploration bonus in Step 4. The full proof is in Appendix A.

**Step 1: Build Confidence Set $\widehat{\mathcal{C}}_{k,h}^E$ in the Exploration Phase**   LSVI-RFE estimates the parameter $\{\boldsymbol{\mu}_h\}_{h \in [H]}$ of the transition probability matrix in the exploration phase. The confidence set $\widehat{\mathcal{C}}_{k,h}^E$ is built with a Hoeffding-type self-normalized bound, i.e., Lemma C.1 in Appendix C and a standard covering net argument, e.g., Lemma B.3. in Jin et al. (2020b), such that we have for any $k \in [K], h \in [H]$, with high probability,

$$\boldsymbol{\mu}_h \in \widehat{\mathcal{C}}_{k,h}^E := \left\{\boldsymbol{\mu} : \left\|(\boldsymbol{\mu} - \widehat{\boldsymbol{\mu}}_{k,h})\,\widehat{\boldsymbol{V}}_{k,h+1}\right\|_{\widehat{\Lambda}_{k,h}} \leqslant \widehat{\beta}^E\right\}. \tag{6}$$

This is detailed in Lemma C.3 with $\widehat{\beta}^E = \widetilde{O}(d\sqrt{H})$. Consequently, we have $(\widehat{\mathbb{P}}_{k,h} - \mathbb{P}_h)\widehat{V}_{k,h+1} \leqslant \widehat{\beta}^E \|\boldsymbol{\phi}(s_h^k, a_h^k)\|_{\widehat{\Lambda}_{k,h}^{-1}}$ by the Cauchy-Schwarz inequality. Under confidence set $\widehat{\mathcal{C}}_{k,h}^E$, $\widehat{V}_{k,h}(\cdot)$ is a upper confidence estimator of the optimal value function with exploration-driven reward, i.e., $V_h^*(\cdot; r^k)$, where $r_k = \{r_{k,h}\}_{h \in [H]}$, which is detailed in Lemma A.7.

**Step 2: Build the Confidence Set $\widehat{\mathcal{C}}_h^P$ in the Planning Phase**   We prove that, with high probability, for any $h \in [H]$ in the planning phase,

$$\boldsymbol{\mu}_h \in \widehat{\mathcal{C}}_h^P := \left\{\boldsymbol{\mu} : \left\|(\boldsymbol{\mu} - \widehat{\boldsymbol{\mu}}_{K+1,h})\,\widehat{\boldsymbol{V}}_{h+1}\right\|_{\widehat{\Lambda}_{K+1,h}} \leqslant \widehat{\beta}^P\right\}, \tag{7}$$

where the bonus radius $\widehat{\beta}^P = \widetilde{O}(\sqrt{dH})$ is sharper than $\widehat{\beta}^E = \widetilde{O}(d\sqrt{H})$ in the exploration phase. Specifically, we build the confidence set $\widehat{\mathcal{C}}_h^P$ by the intersection of two confidence sets $\widehat{\mathcal{C}}_h^{P(1)}$ and $\widehat{\mathcal{C}}_h^{P(2)}$, similar to that in Azar et al. (2017) for tabular MDPs, where

$$\widehat{\mathcal{C}}_h^{P(1)} := \left\{\boldsymbol{\mu} : \left\|(\boldsymbol{\mu} - \widehat{\boldsymbol{\mu}}_{K+1,h})\,\boldsymbol{V}_{h+1}^*\right\|_{\widehat{\Lambda}_{K+1,h}} \leqslant \widehat{\beta}^{P(1)}\right\},$$

$$\widehat{\mathcal{C}}_h^{P(2)} := \left\{\boldsymbol{\mu} : \left\|(\boldsymbol{\mu} - \widehat{\boldsymbol{\mu}}_{K+1,h})\left(\widehat{\boldsymbol{V}}_{h+1} - \boldsymbol{V}_{h+1}^*\right)\right\|_{\widehat{\Lambda}_{K+1,h}} \leqslant \widehat{\beta}^{P(2)}\right\},$$

where $V_{h+1}^*(\cdot)$ refers to $V_{h+1}^*(\cdot; r)$. A couple of remarks are in place here. (i) In particular, since $V_{h+1}^*(\cdot)$ is a fixed function, where $r$ is the real reward function given in planning phase, we can

---

[2]The exact forms of $\widehat{\beta}^E, \widehat{\beta}^{P(1)}, \widehat{\beta}^{P(2)}$ are given in Eq. (20), (30) and (32), respectively in Appendix A

build confidence set $\widehat{\mathcal{C}}_h^{P(1)}$ with radius $\widehat{\beta}^{P(1)} = \widetilde{O}(\sqrt{Hd})$ by Hoeffding-type self-normalized bound directly without a uniform convergence argument, as detailed in Lemma A.11. (ii) When building the confidence set $\widehat{\mathcal{C}}_h^{P(2)}$, the Bernstein-type self-normalized bound (Lemma C.2 in Appendix C.1) is applied, where the variance term ($\sigma^2$) and elliptical potential term ($R$) need to be controlled. In fact, we build the monotonicity of the estimated value function between the exploration phase and planning phase in Lemma A.15, i.e., $V_h^*(\cdot; r) + \widehat{V}_{k,h}(\cdot) \geqslant \widehat{V}_h(\cdot)$ such that the variance term ($\sigma^2$) can be bounded by the variance of $\widehat{V}_{k,h}(\cdot)$. Moreover, by dynamically adjusting $w_{k,h}$, we can keep the elliptical potential term ($R$) small, which is detailed in Lemma A.9. This gives $\widehat{\beta}^{P(2)} = \widetilde{O}(\sqrt{Hd})$ as detailed in Lemma A.17. Consequently, the overhead due to building a uniform convergence argument by covering net is removed and $\widehat{\beta}^P = \widehat{\beta}^{P(1)} + \widehat{\beta}^{P(2)} = \widetilde{O}(\sqrt{Hd})$.

**Step 3: Bound the Exploration Error**  The exploration error refers to the summation of the constructed optimistic value function in the exploration phase, i.e., $\widehat{V}_{k,1}$, which is upper bounded by the summation of the bonus term under the confidence sets $\widehat{\mathcal{C}}_{k,h}^E$ for any $h \in [H]$. It is named as the exploration error since the summation of the bonus term upper bounds the estimation error $(\widehat{\mathbb{P}}_{k,h} - \mathbb{P}_h)\widehat{V}_{k,h+1}$. In particular, the exploration error is upper bounded in Lemma A.20 by

$$\sum_{k=1}^K \widehat{V}_{k,1}(s_1^k) \leqslant 4\widehat{\beta}^E \underbrace{\sqrt{\sum_{k=1}^K \sum_{h=1}^H \widehat{\sigma}_{k,h}^2}}_{I_1} \underbrace{\sqrt{\sum_{k=1}^K \sum_{h=1}^H \min\left\{\left\|\widehat{\sigma}_{k,h}^{-1}\phi(s,a)\right\|_{\widehat{\Lambda}_{k,h}^{-1}}^2, 1\right\}}}_{I_2} \leqslant \widetilde{O}(\sqrt{d^3 H^4 K}) \quad (8)$$

where the second inequality holds since $I_2$ can be bounded as $\widetilde{O}(\sqrt{Hd})$ by the elliptical potential lemma (Lemma C.6), and $I_1 = \widetilde{O}(\sqrt{HT})$ since the enlargement operation in Line 20 of Algorithm 1 rarely happens due to the conservatism of elliptical potentials according to Lemma C.7, and the summation of $W_{k,h}$ is in the order of $\sqrt{T}$ by Lemma A.19.

**Step 4: Bound the Sub-optimality Gap**  The sub-optimality gap refers to the expected gap between the optimal value function $V_1^*(\cdot; r)$ and the value function $V_1^\pi(\cdot; r)$ associated with the recovered policy $\pi$ in the planning phase. Notice that $\|\phi(\cdot, \cdot)\|_{\widehat{\Lambda}_{K+1,h}^{-1}} \leqslant \|\phi(\cdot, \cdot)\|_{\widehat{\Lambda}_{k,h}^{-1}}$ since $\widehat{\Lambda}_{k,h} \preceq \widehat{\Lambda}_{K+1,h}$ for any $k \in [K]$. We thus have

$$r_{k,h}(\cdot, \cdot) = \widehat{\beta}^E \|\phi(\cdot, \cdot)\|_{\widehat{\Lambda}_{k,h}^{-1}} \geqslant \sqrt{d} b_h(\cdot, \cdot) = \sqrt{d}\widehat{\beta}^P \|\phi(\cdot, \cdot)\|_{\widehat{\Lambda}_{K+1,h}^{-1}}, \quad (9)$$

where $b_h(\cdot, \cdot)$ is the exploration bonus in the planning phase. Then, we can apply standard analysis as in Wang et al. (2020a) to bound $\mathbb{E}_{s_1 \sim \mu}[V_1^*(s_1; r) - V_1^\pi(s_1; r)]$ by

$$\mathbb{E}_{s_1 \sim \mu}[V_1^*(s_1; r) - V_1^\pi(s_1; r)] \leqslant \mathbb{E}_{s_1 \sim \mu}\left[\widehat{V}_1(s_1) - V_1^\pi(s_1; r)\right] \quad (10)$$

$$\leqslant \mathbb{E}_{s_1 \sim \mu}\left[\widetilde{V}_1^\pi(s_1; b)\right] \leqslant \mathbb{E}_{s_1 \sim \mu}\left[\widetilde{V}_1^\pi(s_1; r_k)\right]/\sqrt{d} \leqslant \mathbb{E}_{s_1 \sim \mu}\left[\sum_{k=1}^K \widehat{V}_{k,1}(s_1)\right]/(K\sqrt{d}),$$

where the first inequality holds due to optimism under the confidence sets $\widehat{\mathcal{C}}_h^P$ for any $h \in [H]$ (Lemma A.15), the second inequality is the application of regret composition and $\widetilde{V}_1^\pi$ is an auxiliary value function defined in Appendix, the third inequality holds by Eq. (9), and the last inequality holds due to the optimism under the confidence sets $\widehat{\mathcal{C}}_{k,h}^E$ for any $h \in [H]$ (Lemma A.7). Indeed, Eq. (10) establishes the connection between the exploration phase and planning phase. Thus, if $\pi$ is an $\epsilon$-optimal policy, we obtain $K \geqslant \widetilde{O}(d^2 H^4/\epsilon^2)$ by combining Eq. (8) and Eq. (10).

## 6 Lower Bound and Sub-Optimality Gap

We provide a sample complexity lower bound for reward-free RL under the linear MDP setting in Theorem 6.1. We show that there exists an instance of linear MDP, such that any reward-free RL algorithm requires $\Omega(d^2 H^3/\epsilon^2)$ episodes of interaction during the exploration phase to find a near-optimal policy during the planning phase.

**Theorem 6.1** (Lower Bound). *Suppose $H \geqslant 4, d \geqslant 2, 1/32K < \delta < 1/H$. Then, there exists a linear MDP instance $\mathcal{M} = (\mathcal{S}, \mathcal{A}, H, \{\mathbb{P}_h\}, \{r_h\})$, such that any algorithm $\mathcal{ALG}$ that learns an $\epsilon$-optimal policy with probability at least $1 - \delta$ needs to collect at least $K = Cd^2 H^3/\epsilon^2$ episodes during the exploration phase, where $C$ is an absolute constant, and $\delta$ has no dependence on $\epsilon, H, d, K$.*

*Proof Sketch.* Notice that if the reward function is given in the exploration phase, the RFE setting degrades to Probably Approximately Correct (PAC) RL setting Dann et al. (2019). Thus, a lower bound for PAC RL also serves as a lower bound for RFE since an algorithm for RFE also works for PAC RL by neglecting the reward function in the exploration phase. The proof of Theorem 6.1 is inspired by the lower bound of RFE in linear mixture MDPs in Chen et al. (2021). First, we construct a hard-to-learn MDP $\mathcal{M}$ such that any algorithm which runs $K$ episodes will obtain the regret at least $\Omega(dH\sqrt{HK})$ as shown in Zhou et al. (2021). Then, for any algorithm $\mathcal{ALG}_1$ running $K$ episodes to learn an $\epsilon$-optimal policy $\pi(K)$ with probability at least $1 - \delta$, it suffices to prove that under the instance $\mathcal{M}$, $K \geqslant Cd^2H^3/\epsilon^2$. We prove this by constructing a new algorithm $\mathcal{ALG}_2$ which runs $\mathcal{ALG}_1$ in the first $K$ episodes and executes the generated policy $\pi(K)$ in the rest $(c-1)K$ episodes, where $c > 1$ is a positive constant. The regret under $\mathcal{M}$ in the last $(c - 1)K$ episodes by executing $\pi(K)$ satisfies $\Omega(dH\sqrt{HK}) \lesssim \sum_{k=K+1}^{K_2} \mathbb{E}_{x_1 \sim \nu} \left[ V_1^*(x_1) - V_1^{\hat{\pi}}(x_1) \right] \lesssim K\epsilon$, where $\nu$ is the initial state distribution, the first inequality holds due to the hardness of the constructed MDP, and the second inequality holds due to $\pi(K)$ is an $\epsilon$-optimal policy. Thus, we obtain $K \geqslant \Omega(d^2H^3/\epsilon^2)$. For detailed proof, please refer to Appendix B. □

*Remark* 6.2. Theorem 6.1 presents an improved sample complexity lower bound for RFE in linear MDPs than the results of $\Omega(H^2d/\epsilon^2)$ in Zhang et al. (2021) and $\Omega(d^2/\epsilon^2)$ in Wagenmaker et al. (2022). The lower bound and Theorem 4.2 together show that the sample complexity of LSVI-RFE, i.e., $\widetilde{O}(H^4d^2/\epsilon^2)$, matches the lower bound $\Omega(H^3d^2/\epsilon^2)$ except for an $H$ and logarithmic factors. To our best knowledge, the upper bound in Theorem 4.2 and lower bound in Theorem 6.1 are both sharper than those in existing works Wang et al. (2020a); Zanette et al. (2020c); Zhang et al. (2021); Chen et al. (2021).

## 6.1 TOWARDS MINIMAX OPTIMALITY

The sample complexity of LSVI-RFE is $\widetilde{O}(H^4d^2/\epsilon^2)$, which matches the lower bound up to an $H$ and logarithmic factors. The factor $H$ between the upper and lower bounds is potentially due to utilizing a Hoeffding-type bonus instead of a Bernstein-type one for building the confidence set $\widehat{\mathcal{C}}^{P(1)}$. Intuitively, a Bernstein-type bonus based on the variance of the value function, combined with the Law of Total Variance (LTV) Lattimore & Hutter (2012), can effectively reduce a $\sqrt{H}$ factor in the statistical complexity of RL algorithms. This phenomenon has been observed in existing works Zhou et al. (2021); Hu et al. (2022) for regret minimization in linear MDPs. However, to utilize the Bernstein-type inequality in building $\widehat{\mathcal{C}}^{P(1)}$, we estimate the variance of the optimal value function $V_{h+1}^*(\cdot, r)$ with real reward function. Unfortunately, under the RFE, the agent is unaware of the real reward during the exploration phase, which brings obstacles to building the variance estimator. In the Linear mixture setting, Chen et al. (2021); Zhang et al. (2021) successfully utilize the Bernstein-type self-normalized bound and presents a nearly minimax optimal algorithm. However, their works still cannot estimate the variance of the optimal value function. Instead, they build an upper-bound estimator from a candidate set to avoid estimating the variance, which is computationally inefficient.

**Degradation to PAC RL** Our sample complexity upper and lower bounds in Theorem 4.2 and 6.1 are also applicable to the PAC RL setting, i.e., the agent is aware of the reward function during the exploration phase, both upper and lower bounds are sharp, yet there is still an $H$ gap. However, a direct adaption of the Bernstein-type bonus will not improve the sample complexity under PAC RL due to policy inconsistency. Specifically, when we apply the total variance lemma (Lemma C.5 in Jin et al. (2018)), the exploration policy $\pi^k$ for episode $k$ is inconsistent with the recovered policy $\pi$ in the planning phase, since $\pi^k$ is the greedy with respect to constructed value function by exploration-driven reward function, instead of the real reward function given in the planning phase. A potential solution may be bounding the distance between two policies by policy distance measures, e.g., KL divergence, which helps tabular RFE to reach minimax optimality in Ménard et al. (2021).

## 7 CONCLUSION

This work studies reward-free reinforcement learning with linear function approximation for episodic MDPs. We propose a novel computation-efficient algorithm LSVI-RFE with $\widetilde{O}(H^4d^2/\epsilon^2)$ sample complexity upper bound for linear MDPs. We also establish a sample complexity lower bound of $\Omega(H^3d^2/\epsilon^2)$, showing that LSVI-RFE's complexity is optimal up to an $H$ and logarithmic factors. LSVI-RFE introduces a novel variance-aware exploration mechanism with weighted linear regression to avoid overly-aggressive exploration in prior works. Our sharp bound relies on the decoupling of UCB bonuses during two phases, a Bernstein-type self-normalized bound and the conservatism of elliptical potentials We leave removing the $H$ gap as future work.

## REPRODUCIBILITY STATEMENT

The assumption we make to the MDP structure can be found in Definition 3.1. The complete proof of our main theoretical results, Theorem 4.2 and Theorem 6.1 can be found in Appendix A and Appendix B, respectively. We also provide auxiliary lemmas required for our complete proof in Appendix C.

## ACKNOWLEDGEMENTS

The work is supported by the Technology and Innovation Major Project of the Ministry of Science and Technology of China under Grant 2020AAA0108400 and 2020AAA0108403, the Tsinghua University Initiative Scientific Research Program, and Tsinghua Precision Medicine Foundation 10001020109.

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

# Supplementary Materials

# A PROOF OF UPPER BOUND (THEOREM 4.2)

## A.1 NOTATIONS AND PRELIMINARIES

We summarize the key notations used in our analysis in Table 2.

Table 2: Notations

| Symbol | Explanation |
|--------|-------------|
| $\mathcal{F}_{k,h}$ | $\sigma$-algebra generated by random states and actions up to stage $h$ of episode $k$ |
| $\widehat{\mathcal{V}}(L, B)$ | Function class defined in Definition A.4 |
| $\widehat{V}_{k,h}(\cdot)$ | Constructed optimistic value function in exploration phase |
| $\widehat{V}_h(\cdot)$ | Constructed optimistic value function in planning phase |
| $V_h^*(s; r)$ | Optimal value function in planning phase |
| $V_h^*(s)$ | Abbreviation of $V_h^*(s; r)$ |
| $\widetilde{V}_h^*(s; r)$ | Truncated optimal value function in planning phase, defined in Eq. (19) |
| $\widehat{\beta}^E$ | $\widetilde{O}(d\sqrt{H})$, exploration bonus radius built in Lemma A.6 |
| $\widehat{\beta}^{P(1)}$ | $\widetilde{O}(\sqrt{Hd})$, part one of planning bonus radius built in Lemma A.11 |
| $\widehat{\beta}^{P(2)}$ | $\widetilde{O}(\sqrt{Hd})$, part two of planning bonus radius built in Lemma A.12 |
| $\widehat{\beta}^P$ | $\widetilde{O}(\sqrt{Hd})$, planning bonus radius built in Lemma A.13 |
| $\widehat{\mathcal{C}}_{k,h}^E$ | Confidence set in exploration phase, defined in Eq. (11) |
| $\widehat{\mathcal{C}}_h^{P(1)}$ | Part one of confidence set in planning phase, defined in Eq. (14) |
| $\widehat{\mathcal{C}}_h^{P(2)}$ | Part two of confidence set in planning phase, defined in Eq. (15) |
| $\widehat{\mathcal{C}}_h^P$ | Confidence set in planning phase, defined in Eq. (16) |
| $\Psi_{k,h}^E$ | Episodic optimism event in exploration phase, defined in Eq. (12) |
| $\Psi_h^E$ | Optimism event in exploration phase, defined in Eq. (13) |
| $\Psi_h^P$ | Optimism event in planning phase, defined in Eq. (17) |
| $\epsilon_h^k$ | Defined as $\mathbb{P}_h(\cdot \mid s_h^k, a_h^k) - \boldsymbol{\delta}(s_{h+1}^k)$ |
| $\widehat{\mathbb{P}}_{k,h}(\cdot \mid s, a)$ | Estimated transition probability, $\widehat{\boldsymbol{\mu}}_{k,h}^\top \boldsymbol{\phi}(s, a)$ |

Before the formal proof begins, we first start with some necessary definitions of measurable space and filtration required during our proofs.

**Measurable Space** Note that the stochasticity in the transition probability of the MDP is the only source of randomness. Denote $\mathbb{P}$ as the gather of the distributions over state-action pair sequence $(\mathcal{S} \times \mathcal{A})^{\mathbb{N}}$, induced by the interconnection of policy obtained from Algorithm 1 and the episodic linear MDP $\mathcal{M}$. Denote $\mathbb{E}$ as the corresponding expectation operator. Hence, all random variables can be defined over the sample space $\Omega = (\mathcal{S} \times \mathcal{A})^{\mathbb{N}}$. Thus, we work with the probability space given by the triplet $(\Omega, \mathcal{F}, \mathbb{P})$, where $\mathcal{F}$ is the product $\sigma$-algebra generated by the discrete $\sigma$-algebras underlying $\mathcal{S}$ and $\mathcal{A}$.

**Definition A.1** (Filtration). For any $k \in [K]$ and any $h \in [H]$, let $\mathcal{F}_{k,h}$ be the $\sigma$-algebra generated by the random variables representing the state-action pairs up to and including that appears in stage $h$ of episode $k$.

**Measurability** $\widetilde{\sigma}_{k,h}, \widehat{\sigma}_{k,h}, \widetilde{\boldsymbol{\Lambda}}_{k+1,h}, \widehat{\boldsymbol{\Lambda}}_{k+1,h}$ are $\mathcal{F}_{k,h}$-measurable, $\widehat{\boldsymbol{\mu}}_{k+1,h}$ is $\mathcal{F}_{k,h+1}$-measurable, $\widehat{Q}_{k,h}, \widehat{V}_{k,h}, \pi_h^k$ are $\mathcal{F}_{k-1,H}$-measurable, yet not $\mathcal{F}_{k-1,h}$-measurable due to their backwards construction.

We provide some necessary definitions of high probability events in Definition A.2 and Definition A.3 for exploration and planning phases, respectively. In particular, high probability events for the exploration phase are built in Appendix A.2, and those for the planning phase are built in Appendix A.3.

**Definition A.2** (High Probability Events in Exploration Phase).

- Confidence Set in Exploration Phase

$$\widehat{\mathcal{C}}_{k,h}^{E} := \left\{ \boldsymbol{\mu} : \left\| (\boldsymbol{\mu} - \widehat{\boldsymbol{\mu}}_{k,h}) \widehat{\boldsymbol{V}}_{k,h+1} \right\|_{\widehat{\Lambda}_{k,h}} \leqslant \widehat{\beta}^{E} \right\}, \quad \forall (k,h) \in [K] \times [H] \quad (11)$$

- Optimistic Events in Exploration Phase

$$\Psi_{k,h}^{E} := \{ \boldsymbol{\mu}_{h'} \in \widehat{\mathcal{C}}_{k,h'}^{E}, \forall h \leqslant h' \leqslant H \}, \qquad \forall (k,h) \in [K] \times [H] \quad (12)$$

$$\Psi_{h}^{E} := \{ \boldsymbol{\mu}_{h'} \in \widehat{\mathcal{C}}_{i,h'}^{E}, \forall i \in [K], \forall h \leqslant h' \leqslant H \}, \qquad \forall h \in [H] \quad (13)$$

**Definition A.3** (High Probability Events in Planning Phase).

- Confidence Sets in Planning Phase

$$\widehat{\mathcal{C}}_{h}^{P(1)} := \left\{ \boldsymbol{\mu} : \left\| (\boldsymbol{\mu} - \widehat{\boldsymbol{\mu}}_{K+1,h}) \boldsymbol{V}_{h+1}^{*} \right\|_{\widehat{\boldsymbol{\Lambda}}_{K+1,h}} \leqslant \widehat{\beta}^{P(1)} \right\}, \qquad \forall h \in \times [H] \quad (14)$$

$$\widehat{\mathcal{C}}_{h}^{P(2)} := \left\{ \boldsymbol{\mu} : \left\| (\boldsymbol{\mu} - \widehat{\boldsymbol{\mu}}_{K+1,h}) \left( \widehat{\boldsymbol{V}}_{h+1} - \boldsymbol{V}_{h+1}^{*} \right) \right\|_{\widehat{\boldsymbol{\Lambda}}_{K+1,h}} \leqslant \widehat{\beta}^{P(2)} \right\}, \quad \forall h \in \times [H] \quad (15)$$

$$\widehat{\mathcal{C}}_{h}^{P} := \left\{ \boldsymbol{\mu} : \left\| (\boldsymbol{\mu} - \widehat{\boldsymbol{\mu}}_{K+1,h}) \widehat{\boldsymbol{V}}_{h+1} \right\|_{\widehat{\boldsymbol{\Lambda}}_{K+1,h}} \leqslant \widehat{\beta}^{P} \right\}, \qquad \forall h \in \times [H] \quad (16)$$

- Optimistic Event in Planning Phase

$$\Psi_{h}^{P} := \{ \boldsymbol{\mu}_{h'} \in \widehat{\mathcal{C}}_{h'}^{P}, \forall h \leqslant h' \leqslant H \}, \quad \forall h \in [H] \quad (17)$$

We also define the optimistic value function class below, which is required to build a uniform convergence argument by the covering net.

**Definition A.4** (Optimistic Value Function Class). For any $L, B > 0$, let $\widehat{\mathcal{V}}(L, B)$ denote a class of functions mapping from $\mathcal{S}$ to $\mathbb{R}$ with the following parametric form

$$V(\cdot) = \min \left\{ \max_{a} \boldsymbol{w}^{\top} \boldsymbol{\phi}(\cdot, a) + \beta \sqrt{\boldsymbol{\phi}(\cdot, a)^{\top} \boldsymbol{\Lambda}^{-1} \boldsymbol{\phi}(\cdot, a)}, H \right\}, \quad (18)$$

where the parameters $(\boldsymbol{w}, \beta, \boldsymbol{\Lambda})$ satisfy $\|\boldsymbol{w}\|_{2} \leqslant L$, $\beta \in [0, B]$, the minimum eigenvalue satisfies $\lambda_{\min}(\boldsymbol{\Lambda}) \geqslant \lambda$, and $\sup_{s,a} \|\boldsymbol{\phi}(s, a)\|_{2} \leqslant 1$.

Moreover, we also utilize the truncated optimal value function in the planning phase, which is defined recursively as below. Compared to the definition of $V_{h}^{*}(s; r)$, we take minimization over the value function and $H$ in each step in this definition. We can similarly define $\widetilde{V}_{h}^{\pi}(s; r), \widetilde{Q}_{h}^{\pi}(s, a; r)$.

**Definition A.5** (Truncated Optimal Value Function). We introduce the value function $\widetilde{V}_{h}^{*}(s; r)$ which is recursively defined from step $H + 1$ to step 1:

$$\begin{aligned}
\widetilde{V}_{H+1}^{*}(s; r) &= 0, \quad \forall s \in \mathcal{S} \\
\widetilde{Q}_{h}^{*}(s, a; r) &= r_{h}(s, a) + \mathbb{P}_{h} \widetilde{V}_{h+1}^{*}(s, a; r), \quad \forall (s, a) \in \mathcal{S} \times \mathcal{A} \\
\widetilde{V}_{h}^{*}(s; r) &= \min \left\{ \max_{a \in \mathcal{A}} r_{h}(s, a) + \mathbb{P}_{h} \widetilde{V}_{h+1}^{*}(s, a; r), H \right\}, \quad \forall s \in \mathcal{S}, h \in [H].
\end{aligned} \quad (19)$$

**Proof Overview** We first build the confidence sets $\widehat{\mathcal{C}}_{k,h}^{E}$ and $\widehat{\mathcal{C}}_{h}^{P}$ for both exploration and planning phases, respectively in Appendix A.2 and A.3. Subsequently, we bound the exploration error, i.e., summation of the exploration bonus, during the exploration phase based on the built confidence set $\widehat{\mathcal{C}}_{k,h}^{E}$ in Appendix A.4. Finally, we can bound the sub-optimality gap of the recovered policy in the planning phase with confidence set $\widehat{\mathcal{C}}_{h}^{P}$ and the monotonicity of the exploration bonus in Appendix A.5, which also ends the proof of Theorem 4.2.

## A.2 HIGH PROBABILITY EVENTS IN EXPLORATION PHASE

In this subsection, we build high probability events in exploration phase, including confidence set $\widehat{\mathcal{C}}_{k,h}^{E}$ and optimism in Lemma A.6 and Lemma A.7, respectively.

### A.2.1 CONFIDENCE SET IN EXPLORATION PHASE

**Lemma A.6.** *In Algorithm 1, for any $\delta \in (0,1)$ and fixed $h \in [H]$, with probability at least $1 - \delta/H$, we have that for any $k \in [K]$,*

$$\boldsymbol{\mu}_h \in \widehat{\mathcal{C}}_{k,h}^E = \left\{ \boldsymbol{\mu} : \left\| (\boldsymbol{\mu} - \widehat{\boldsymbol{\mu}}_{k,h}) \, \widehat{\boldsymbol{V}}_{k,h+1} \right\|_{\widehat{\Lambda}_{k,h}} \leqslant \widehat{\beta}^E \right\},$$

*where*

$$
\widehat{\beta}^E = \sqrt{H} \sqrt{ d \log\left(1 + \frac{K}{Hd\lambda}\right) + \log\left(\frac{H}{\delta}\right) + d \log\left(1 + \frac{8K^2\sqrt{d}}{H\lambda^{3/2}}\right) + d^2 \log\left(1 + \frac{32K^2 B_E^2}{H^2\lambda^2\sqrt{d}}\right) }
$$
$$
+ H\sqrt{\lambda d} + 1
$$

(20)

*with $B_E$ satisfying $3\widehat{\beta}^E \leqslant B_E$.*

*Proof.* (Lemma A.6) In Line 10 of Algorithm 1, for any $(k,h) \in \times [K] \times [H]$, we have

$$\widehat{V}_{k,h}(\cdot) = \min\left\{ \max_a \langle \widehat{\boldsymbol{\mu}}_{k,h} \widehat{\boldsymbol{V}}_{k,h+1}, \boldsymbol{\phi}(\cdot, a) \rangle + 3\widehat{\beta}^E \| \boldsymbol{\phi}(\cdot, a) \|_{\widehat{\Lambda}_{k,h}^{-1}}, H \right\}.$$

Moreover,

$$\left\| \widehat{\boldsymbol{\mu}}_{k,h} \widehat{\boldsymbol{V}}_{k,h+1} \right\|_2 = \left\| \widehat{\Lambda}_{k,h}^{-1} \sum_{i=1}^{k-1} \widehat{\sigma}_{i,h}^{-2} \boldsymbol{\phi}(s_h^i, a_h^i) \widehat{V}_{k,h+1}(s_{h+1}^i) \right\|_2 \leqslant \frac{H}{(\sqrt{H})^2} \left\| \widehat{\Lambda}_{k,h}^{-1} \sum_{i=1}^{k-1} \boldsymbol{\phi}(s_h^i, a_h^i) \right\|_2 \leqslant \frac{K}{\lambda},$$

where the first inequality holds since $\widehat{V}_{k,h+1}(\cdot) \leqslant H$ and $\widehat{\sigma}_{i,h} \geqslant \sqrt{H}$ for any $i \in [k]$, and the second inequality holds since $\lambda_{\min}(\boldsymbol{\Lambda}_{k,h}) \geqslant \lambda$ and $\sup_{s,a} \| \boldsymbol{\phi}(s,a) \|_2 \leqslant 1$.

Thus, we claim that $\widehat{V}_{k,h} \in \widehat{\mathcal{V}}(L_E, B_E)$ for any $(k,h) \in [K] \times [H]$, where function set $\widehat{\mathcal{V}}(\cdot, \cdot)$ is defined in Definition A.4, $L_E = K/\lambda$ and $B_E$ is a constant satisfying $3\widehat{\beta}^E \leqslant B_E$ with $\widehat{\beta}^E$ given in Eq. (20.

For a fixed function $V \in \widehat{\mathcal{V}}(L_E, B_E)$, let $\mathcal{G}_i = \mathcal{F}_{i,h}$, $x_i = \widehat{\sigma}_{i,h}^{-1} \boldsymbol{\phi}(s_h^i, a_h^i)$ and

$$\eta_i = \widehat{\sigma}_{i,h}^{-1} {\boldsymbol{\epsilon}_h^i}^\top \boldsymbol{V} = \widehat{\sigma}_{i,h}^{-1} (\langle \boldsymbol{\mu}_h \boldsymbol{V}, \boldsymbol{\phi}(s_h^i, a_h^i) \rangle - V(s_{h+1}^i))$$

for any $i \in [k]$. Then, $x_i$ is $\mathcal{G}_i$-measurable and $\eta_i$ is $\mathcal{G}_{i+1}$-measurable, and we have $\mathbb{E}[\eta_i | \mathcal{G}_i] = 0$. Since $\widehat{\sigma}_{i,h} \geqslant \sqrt{H}$, we also have $\| x_i \|_2 \leqslant 1/\sqrt{H}$ and $|\eta_i| \leqslant \sqrt{H}$.

By Lemma C.1, for any $k \in [K]$ and fixed $h \in [H]$, with probability at least $1 - \delta/H$,

$$\left\| \sum_{i=1}^{k-1} \widehat{\sigma}_{i,h}^{-2} \boldsymbol{\phi}(s_h^i, a_h^i) {\boldsymbol{\epsilon}_h^i}^\top \boldsymbol{V} \right\|_{\widehat{\Lambda}_{k,h}^{-1}} \leqslant \sqrt{H} \sqrt{d \log(1 + (k-1)/(Hd\lambda)) + \log(H/\delta)}$$

$$\leqslant \sqrt{H} \sqrt{d \log(1 + K/(Hd\lambda)) + \log(H/\delta)}$$

Denote the $\varepsilon$-cover of function class $\widehat{\mathcal{V}}(L_E, B_E)$ as $\widehat{\mathcal{N}}_\varepsilon(L_E, B_E)$. For an arbitrary $f(\cdot) \in \widehat{\mathcal{V}}(L_E, B_E)$, there exists a $V(\cdot) \in \widehat{\mathcal{N}}_\varepsilon$, such that $\| \boldsymbol{f} - \boldsymbol{V} \|_\infty \leqslant \varepsilon$. Since $\| {\boldsymbol{\epsilon}_h^i}^\top (\boldsymbol{f} - \boldsymbol{V}) \|_2 \leqslant 2\varepsilon$ and $\left\| \sum_{i=1}^{k-1} \widehat{\sigma}_{i,h}^{-2} \boldsymbol{\phi}(s_h^i, a_h^i) {\boldsymbol{\epsilon}_h^i}^\top \right\|_{\widehat{\Lambda}_{k,h}^{-1}} \leqslant K\sqrt{d}/(H\sqrt{\lambda})$, we have

$$\left\| \sum_{i=1}^{k-1} \widehat{\sigma}_{i,h}^{-2} \boldsymbol{\phi}(s_h^i, a_h^i) {\boldsymbol{\epsilon}_h^i}^\top (\boldsymbol{f} - \boldsymbol{V}) \right\|_{\widehat{\Lambda}_{k,h}^{-1}} \leqslant \frac{2\varepsilon K\sqrt{d}}{H\sqrt{\lambda}}.$$

(21)

Thus,

$$
\left\| \sum_{i=1}^{k-1} \widehat{\sigma}_{i,h}^{-2} \phi(s_h^i, a_h^i) \boldsymbol{\epsilon}_h^i{}^\top \boldsymbol{f} \right\|_{\widehat{\Lambda}_{k,h}^{-1}}
$$

$$
\leqslant \left\| \sum_{i=1}^{k-1} \widehat{\sigma}_{i,h}^{-2} \phi(s_h^i, a_h^i) \boldsymbol{\epsilon}_h^i{}^\top \boldsymbol{V} \right\|_{\widehat{\Lambda}_{k,h}^{-1}} + \left\| \sum_{i=1}^{k-1} \widehat{\sigma}_{i,h}^{-2} \phi(s_h^i, a_h^i) \boldsymbol{\epsilon}_h^i{}^\top (\boldsymbol{f} - \boldsymbol{V}) \right\|_{\widehat{\Lambda}_{k,h}^{-1}}
$$

$$
\leqslant \left\| \sum_{i=1}^{k-1} \widehat{\sigma}_{i,h}^{-2} \phi(s_h^i, a_h^i) \boldsymbol{\epsilon}_h^i{}^\top \boldsymbol{V} \right\|_{\widehat{\Lambda}_{k,h}^{-1}} + \frac{2\varepsilon K \sqrt{d}}{H \sqrt{\lambda}} \tag{22}
$$

$$
\leqslant \sqrt{H} \sqrt{d \log(1 + K/(Hd\lambda)) + \log(H/\delta) + \log |\widehat{\mathcal{N}}_\varepsilon(L_E, B_E)|} + \frac{2\varepsilon K \sqrt{d}}{H \sqrt{\lambda}}.
$$

where the first inequality is due to triangle inequality, the second one holds by Eq. (21), and the third inequality follows from a union bound over all functions in $\widehat{\mathcal{N}}_\varepsilon(L_E, B_E)$ with

$$
\log |\widehat{\mathcal{N}}_\varepsilon(L_E, B_E)| \leqslant d \log \left(1 + 4 L_E \varepsilon\right) + d^2 \log(1 + 8 d^{1/2} B_E^2 / (\lambda \varepsilon^2)), \tag{23}
$$

according to Lemma C.5.

Moreover, we have

$$
\left\| \left(\widehat{\boldsymbol{\mu}}_{k,h} - \boldsymbol{\mu}_h\right) \widehat{\boldsymbol{V}}_{k,h+1} \right\|_{\widehat{\boldsymbol{\Lambda}}_{k,h}}
$$

$$
= \left\| \widehat{\boldsymbol{\Lambda}}_{k,h}^{-1} \left[ -\lambda \boldsymbol{\mu}_h + \sum_{i=1}^{k-1} \widehat{\sigma}_{i,h}^{-2} \phi\left(s_h^i, a_h^i\right) \boldsymbol{\epsilon}_h^i{}^\top \right] \widehat{\boldsymbol{V}}_{k,h+1} \right\|_{\widehat{\boldsymbol{\Lambda}}_{k,h}}
$$

$$
\leqslant \left\| -\lambda \boldsymbol{\mu}_h \widehat{\boldsymbol{V}}_{k,h+1} \right\|_{\widehat{\boldsymbol{\Lambda}}_{k,h}^{-1}} + \left\| \sum_{i=1}^{k-1} \widehat{\sigma}_{i,h}^{-2} \phi\left(s_h^i, a_h^i\right) \boldsymbol{\epsilon}_h^i{}^\top \widehat{\boldsymbol{V}}_{k,h+1} \right\|_{\widehat{\boldsymbol{\Lambda}}_{k,h}^{-1}} \tag{24}
$$

$$
\leqslant \frac{1}{\sqrt{\lambda}} \cdot \lambda H \sqrt{d} + \left\| \sum_{i=1}^{k-1} \widehat{\sigma}_{i,h}^{-2} \phi\left(s_h^i, a_h^i\right) \boldsymbol{\epsilon}_h^i{}^\top \widehat{\boldsymbol{V}}_{k,h+1} \right\|_{\widehat{\boldsymbol{\Lambda}}_{k,h}^{-1}},
$$

where the equality is due to Lemma C.4, the first inequality is due to the triangle inequality, and the second inequality holds since $\|\boldsymbol{\mu}_h \widehat{\boldsymbol{V}}_{k,h+1}\|_2 \leqslant H \sqrt{d}$ and the minimum eigenvalue of $\widehat{\boldsymbol{\Lambda}}_{k,h}$ is no less than $\lambda$.

Thus, since $\widehat{\boldsymbol{V}}_{k,h+1} \in \widehat{\mathcal{V}}$, we have that with probability at least $1 - \delta/H$, any $k \in [K]$ and fixed $h \in [H]$:

$$
\left\| \left(\widehat{\boldsymbol{\mu}}_{k,h} - \boldsymbol{\mu}_h\right) \widehat{\boldsymbol{V}}_{k,h+1} \right\|_{\widehat{\Lambda}_{k,h}}
$$

$$
\leqslant \sqrt{\lambda} H \sqrt{d} + \left\| \sum_{i=1}^{k-1} \widehat{\sigma}_{i,h}^{-2} \phi(s_h^i, a_h^i) \boldsymbol{\epsilon}_h^i{}^\top \widehat{\boldsymbol{V}}_{k,h+1} \right\|_{\widehat{\Lambda}_{k,h}^{-1}}
$$

$$
\leqslant \sqrt{H} \sqrt{d \log \left(1 + \frac{K}{Hd\lambda}\right) + \log \left(\frac{H}{\delta}\right) + d \log \left(1 + \frac{8 K L_E \sqrt{d}}{H \sqrt{\lambda}}\right) + d^2 \log \left(1 + \frac{32 K^2 B_E^2}{H^2 \lambda^2 \sqrt{d}}\right)}
$$

$$
+ H \sqrt{\lambda d} + 1 = \widehat{\beta}^E,
$$

where the last inequality follows by Eq. (22) and setting $\varepsilon = H \sqrt{\lambda}/(2 K \sqrt{d})$. $\qquad\square$

### A.2.2 OPTIMISM IN EXPLORATION PHASE

**Lemma A.7** (Optimism in Exploration Phase). *In Algorithm 1, for any $k \in [K]$ and any $h \in [H]$, under $\Psi_{k,h}^E$, we have*

$$
\widetilde{V}_h^*(s; r_k) \leqslant \widehat{V}_{k,h}(s), \quad \forall s \in \mathcal{S}.
$$

*Proof.* (Lemma A.7) We first prove the optimism for some fixed episode $k \in [K]$ by induction.

Initially, the statement holds for $h = H + 1$ since $\widehat{V}_{k,H+1}(\cdot) = \widetilde{V}_{H+1}^*(\cdot; r_k) = 0$ by definition. Assume the statement holds for $h + 1$, which means $\widehat{V}_{k,h+1}(\cdot) \geqslant \widetilde{V}_{h+1}^*(\cdot; r_k)$ under $\Psi_{k,h+1}^E$. Recall the definitions of $\widehat{Q}_{k,h}(\cdot, \cdot)$ and $\widetilde{Q}_h^*(\cdot, \cdot; r_k)$, i.e.,

$$\widehat{Q}_{k,h}(\cdot, \cdot) = r_{k,h}(\cdot, \cdot) + \langle \widehat{\boldsymbol{\mu}}_{k,h} \widehat{\boldsymbol{V}}_{k,h+1}, \boldsymbol{\phi}(\cdot, \cdot) \rangle + \widehat{\beta}^E \|\boldsymbol{\phi}(\cdot, \cdot)\|_{\widehat{\boldsymbol{\Lambda}}_{k,h}^{-1}}, \tag{25}$$

$$\widetilde{Q}_h^*(\cdot, \cdot; r_k) = \min \left\{ r_{k,h}(\cdot, \cdot) + \mathbb{P}_h \widetilde{V}_{h+1}^*(\cdot, \cdot; r_k), H \right\}. \tag{26}$$

We have for any $(s, a) \in \mathcal{S} \times \mathcal{A}$ that

$$\widehat{Q}_{k,h}(s, a) - \widetilde{Q}_h^*(s, a; r_k)$$
$$\geqslant r_{k,h}(s, a) + \langle \widehat{\boldsymbol{\mu}}_{k,h} \widehat{\boldsymbol{V}}_{k,h+1}, \boldsymbol{\phi}(s, a) \rangle + \widehat{\beta}^E \|\boldsymbol{\phi}(s, a)\|_{\widehat{\boldsymbol{\Lambda}}_{k,h}^{-1}} - \left[ r_{k,h}(s, a) + \mathbb{P}_h \widetilde{V}_{h+1}^*(s, a; r_k) \right]$$
$$= \langle \widehat{\boldsymbol{\mu}}_{k,h} \widehat{\boldsymbol{V}}_{k,h+1}, \boldsymbol{\phi}(s, a) \rangle - \langle \boldsymbol{\mu}_h \widehat{\boldsymbol{V}}_{k,h+1}, \boldsymbol{\phi}(s, a) \rangle + \widehat{\beta}^E \|\boldsymbol{\phi}(s, a)\|_{\widehat{\boldsymbol{\Lambda}}_{k,h}^{-1}}$$
$$\quad + \mathbb{P}_h \widehat{V}_{k,h+1}(s, a) - \mathbb{P}_h \widetilde{V}_{h+1}^*(s, a; r_k)$$
$$\geqslant - \left\| (\widehat{\boldsymbol{\mu}}_{k,h} - \boldsymbol{\mu}_h) \widehat{\boldsymbol{V}}_{k,h+1} \right\|_{\widehat{\boldsymbol{\Lambda}}_{k,h}} \|\boldsymbol{\phi}(s, a)\|_{\widehat{\boldsymbol{\Lambda}}_{k,h}^{-1}} + \widehat{\beta}^E \|\boldsymbol{\phi}(s, a)\|_{\widehat{\boldsymbol{\Lambda}}_{k,h}^{-1}}$$
$$\quad + \mathbb{P}_h \widehat{V}_{k,h+1}(s, a) - \mathbb{P}_h \widetilde{V}_{h+1}^*(s, a; r_k)$$
$$\geqslant \mathbb{P}_h \widehat{V}_{k,h+1}(s, a) - \mathbb{P}_h \widetilde{V}_{h+1}^*(s, a; r_k)$$
$$\geqslant 0.$$

Here the first inequality holds by Eq. (26), the second inequality follows from Cauchy-Schwarz inequality, the third inequality holds by the assumption that $\boldsymbol{\mu}_h \in \widehat{\mathcal{C}}_{k,h}^E$ under $\Psi_{k,h}^E$, the last inequality holds by the induction assumption that $\widehat{V}_{k,h+1}(\cdot) \geqslant \widetilde{V}_{h+1}^*(\cdot; r_k)$ under $\Psi_{k,h+1}^E$ and $\mathbb{P}_h$ is a valid distribution. Therefore, $\widehat{V}_{k,h}(\cdot) = \min\{\max_{a \in \mathcal{A}} \widehat{Q}_{k,h}(\cdot, a), H\} \geqslant \max_{a \in \mathcal{A}} \widetilde{Q}_h^*(\cdot, a; r_k) = \widetilde{V}_h^*(\cdot; r_k)$.

The lemma follows by applying the above argument to any $k \in [K]$. $\qquad \square$

## A.3 HIGH PROBABILITY EVENTS IN PLANNING PHASE

In this subsection, we built high probability events in planning phase, including confidence set $\widehat{\mathcal{C}}_h^P$ and optimism in Lemma A.13 and Lemma A.14, respectively. In particular, confidence set $\widehat{\mathcal{C}}_h^P$ in Lemma A.13 is built based on two confidence set $\widehat{\mathcal{C}}_h^{P(1)}$ and $\widehat{\mathcal{C}}_h^{P(2)}$ in Lemma A.11 and Lemma A.12, respectively. Apart from the optimism in Lemma A.14, we also build over-optimism between the exploration phase and planning phase in Lemma A.15. In addition, we denote $V_h^*(\cdot) := V_h^*(\cdot; r)$ for convenience, where $r = \{r_h\}_{h \in [H]}$ is the real reward function given in the planning phase.

### A.3.1 CONFIDENCE SET IN PLANNING PHASE

First, we present a lemma regarding the difference between $\widehat{\boldsymbol{\Lambda}}_{k,h}$ and $\widetilde{\boldsymbol{\Lambda}}_{k,h}$.

**Lemma A.8.** *In Algorithm 1, for any $k \in [K + 1]$ and any $h \in [H]$, we have*

$$d^3 \cdot \widehat{\boldsymbol{\Lambda}}_{k,h} \succeq \widetilde{\boldsymbol{\Lambda}}_{k,h}$$

*Proof.* (Lemma A.8) Since $\widehat{\sigma}_{k,h} \leqslant \sqrt{d^3} \widetilde{\sigma}_{k,h}$ by definition, we have

$$d^3 \cdot \widehat{\boldsymbol{\Lambda}}_{k,h} - \widetilde{\boldsymbol{\Lambda}}_{k,h} = \sum_{i=1}^{k-1} ((\widehat{\sigma}_{i,h}/\sqrt{d^3})^{-2} - \widetilde{\sigma}_{i,h}^{-2}) \boldsymbol{\phi}(s_h^i, a_h^i) \boldsymbol{\phi}(s_h^i, a_h^i)^\top \succeq 0$$

is a semi-positive definite matrix. $\qquad \square$

**Lemma A.9.** *In Algorithm 1, for any $i \in [K]$ and any $h \in [H]$, we have*

$$\widehat{\sigma}_{i,h}^{-1} \min \left\{ \|\widehat{\sigma}_{i,h}^{-1} \boldsymbol{\phi}(s_h^i, a_h^i)\|_{\widehat{\boldsymbol{\Lambda}}_{i,h}^{-1}}, 1 \right\} \leqslant \frac{1}{\sqrt{H d^3}}.$$

*Proof.* We discuss two cases that are considered in Algorithm 1.

1. If $\|\widetilde{\sigma}_{i,h}^{-1}\phi(s_h^i, a_h^i)\|_{\widetilde{\mathbf{\Lambda}}_{i,h}^{-1}} \leqslant \frac{1}{d^3}$, then $w_{i,h} = \sqrt{H}$, such that $\widehat{\sigma}_{i,h} = \widetilde{\sigma}_{i,h}$. Thus we have

$$\left\|\widehat{\sigma}_{i,h}^{-1}\phi(s_h^i, a_h^i)\right\|_{\widehat{\mathbf{\Lambda}}_{i,h}^{-1}} = \left\|\widetilde{\sigma}_{i,h}^{-1}\phi(s_h^i, a_h^i)\right\|_{\widehat{\mathbf{\Lambda}}_{i,h}^{-1}} \leqslant \sqrt{d^3}\left\|\widetilde{\sigma}_{i,h}^{-1}\phi(s_h^i, a_h^i)\right\|_{\widetilde{\mathbf{\Lambda}}_{i,h}^{-1}} \leqslant 1/\sqrt{d^3}.$$

   This leads to $\widehat{\sigma}_{i,h}^{-1} \min\left\{\|\widehat{\sigma}_{i,h}^{-1}\phi(s_h^i, a_h^i)\|_{\widehat{\mathbf{\Lambda}}_{i,h}^{-1}}, 1\right\} \leqslant \frac{1}{\sqrt{Hd^3}}$ by $\widehat{\sigma}_{i,h} \geqslant \sqrt{H}$.

2. If $\|\widetilde{\sigma}_{i,h}^{-1}\phi(s_h^i, a_h^i)\|_{\widetilde{\mathbf{\Lambda}}_{i,h}^{-1}} > \frac{1}{d^3}$, then $w_{k,h} = \sqrt{Hd^3}$. Then we have

$$\widehat{\sigma}_{i,h}^{-1} \min\left\{\|\widehat{\sigma}_{i,h}^{-1}\phi(s_h^i, a_h^i)\|_{\widehat{\mathbf{\Lambda}}_{i,h}^{-1}}, 1\right\} \leqslant \widehat{\sigma}_{i,h}^{-1} \leqslant w_{i,h}^{-1} = \frac{1}{\sqrt{Hd^3}}.$$

$\square$

**Lemma A.10.** *For any $i \in [K]$, $h \in [H-1]$, fixed function $V : \mathcal{S} \to [0, H]$, and $\zeta = H\sqrt{\lambda}/(2K\sqrt{d})$, we have*

$$[\mathbb{V}_h(V - V_{h+1}^*)](s_h^i, a_h^i) \cdot \mathbb{1}\left\{V_{h+1}^* - \zeta \leqslant V \leqslant \widehat{V}_{i,h+1} + V_{h+1}^* + \zeta\right\} \cdot \mathbb{1}\left\{\Psi_{i,h}^E\right\} \leqslant W_{i,h}, \quad (27)$$

*where $W_{i,h}$ is defined in Algorithm 1,*

$$W_{i,h} = \min\left\{H \cdot \left(\left\langle\widehat{\boldsymbol{\mu}}_{i,h}\widehat{\boldsymbol{V}}_{i,h+1}, \boldsymbol{\phi}(s_h^i, a_h^i)\right\rangle + \widehat{\beta}^E\left\|\boldsymbol{\phi}(s_h^i, a_h^i)\right\|_{\widehat{\mathbf{\Lambda}}_{i,h}^{-1}} + \zeta\right), H^2\right\} \quad (28)$$

*Proof.* We define $\widetilde{V} := V - V_{h+1}^*$ and $\widetilde{\mathcal{E}} := \left\{-\zeta \leqslant \widetilde{V} \leqslant \widehat{V}_{i,h+1} + \zeta\right\}$ for brevity. Thus we can write

$$[\mathbb{V}_h(V - V_{h+1}^*)](s_h^i, a_h^i) \cdot \mathbb{1}\left\{V_{h+1}^* - \zeta \leqslant V \leqslant \widehat{V}_{i,h+1} + V_{h+1}^* + \zeta\right\} \cdot \mathbb{1}\left\{\Psi_{i,h}^E\right\}$$

$$= [\mathbb{V}_h\widetilde{V}](s_h^i, a_h^i) \cdot \mathbb{1}\left\{\widetilde{\mathcal{E}} \cap \Psi_{i,h}^E\right\}$$

$$= \left(\mathbb{P}_h(\widetilde{V}^2)(s_h^i, a_h^i) - (\mathbb{P}_h\widetilde{V}(s_h^i, a_h^i))^2\right)\mathbb{1}\left\{\widetilde{\mathcal{E}} \cap \Psi_{i,h}^E\right\}$$

$$\leqslant H \cdot (\mathbb{P}_h\widetilde{V}(s_h^i, a_h^i)) \cdot \mathbb{1}\left\{\widetilde{\mathcal{E}} \cap \Psi_{i,h}^E\right\},$$

where the inequality holds by $|\widetilde{V}| \leqslant H$. Moreover, we can further condition on event $\widetilde{\mathcal{E}} \cap \Psi_{i,h}^E$, or the left term of Eq. 27 will be zero. This leads to

$$H \cdot (\mathbb{P}_h\widetilde{V}(s_h^i, a_h^i)) \cdot \mathbb{1}\left\{\widetilde{\mathcal{E}} \cap \Psi_{i,h}^E\right\}$$

$$\leqslant H \cdot (\mathbb{P}_h\widehat{V}_{i,h+1}(s_h^i, a_h^i) + \zeta) \cdot \mathbb{1}\left\{\widetilde{\mathcal{E}} \cap \Psi_{i,h}^E\right\}$$

$$= H \cdot \left(\left\langle\widehat{\boldsymbol{\mu}}_{i,h}\widehat{\boldsymbol{V}}_{i,h+1}, \boldsymbol{\phi}(s_h^i, a_h^i)\right\rangle + \left\langle(\boldsymbol{\mu}_h - \widehat{\boldsymbol{\mu}}_{i,h})\widehat{\boldsymbol{V}}_{i,h+1}, \boldsymbol{\phi}(s_h^i, a_h^i)\right\rangle + \zeta\right) \cdot \mathbb{1}\left\{\widetilde{\mathcal{E}} \cap \Psi_{i,h}^E\right\}$$

$$\leqslant H \cdot \left(\left\langle\widehat{\boldsymbol{\mu}}_{i,h}\widehat{\boldsymbol{V}}_{i,h+1}, \boldsymbol{\phi}(s_h^i, a_h^i)\right\rangle + \left\|(\boldsymbol{\mu}_h - \widehat{\boldsymbol{\mu}}_{i,h})\widehat{\boldsymbol{V}}_{i,h+1}\right\|_{\widehat{\mathbf{\Lambda}}_{i,h}} \left\|\boldsymbol{\phi}(s_h^i, a_h^i)\right\|_{\widehat{\mathbf{\Lambda}}_{i,h}^{-1}} + \zeta\right) \cdot \mathbb{1}\left\{\widetilde{\mathcal{E}} \cap \Psi_{i,h}^E\right\}$$

$$\leqslant H \cdot \left(\left\langle\widehat{\boldsymbol{\mu}}_{i,h}\widehat{\boldsymbol{V}}_{i,h+1}, \boldsymbol{\phi}(s_h^i, a_h^i)\right\rangle + \widehat{\beta}^E\left\|\boldsymbol{\phi}(s_h^i, a_h^i)\right\|_{\widehat{\mathbf{\Lambda}}_{i,h}^{-1}} + \zeta\right) \cdot \mathbb{1}\left\{\widetilde{\mathcal{E}} \cap \Psi_{i,h}^E\right\},$$

where the first inequality holds since we consider event $\widetilde{\mathcal{E}}$, the second inequality holds by Cauchy-Schwarz inequality, and the last inequality holds under event $\Psi_{i,h}^E$. Moreover, the left term of Eq. 27 is smaller than $H^2$, since $|V - V_{h+1}^*| \leqslant H$. Then we have Eq. 27 holds. $\square$

**Lemma A.11.** *In Algorithm 2, for any $k \in [K]$ and fixed $h \in [H]$, with probability at least $1 - \delta/H$:*

$$\boldsymbol{\mu}_h \in \widehat{\mathcal{C}}_h^{P(1)} = \left\{\boldsymbol{\mu} : \left\|(\boldsymbol{\mu} - \widehat{\boldsymbol{\mu}}_{K+1,h})\boldsymbol{V}_{h+1}^*\right\|_{\widehat{\mathbf{\Lambda}}_{K+1,h}} \leqslant \widehat{\beta}^{P(1)}\right\}, \quad (29)$$

*where*

$$\widehat{\beta}^{P(1)} = \sqrt{H}\sqrt{d\log(1 + K/(Hd\lambda)) + \log(H/\delta)} + H\sqrt{\lambda d}. \quad (30)$$

*Proof.* (Lemma A.11) For $i \in [K]$, let $\mathcal{G}_i = \mathcal{F}_{i,h}$, $\boldsymbol{x}_i = \widehat{\sigma}_{i,h}^{-1} \boldsymbol{\phi}\left(s_h^i, a_h^i\right)$ and

$$\eta_i = \widehat{\sigma}_{i,h}^{-1} {\boldsymbol{\epsilon}_h^i}^\top \boldsymbol{V}_{h+1}^* = \widehat{\sigma}_{i,h}^{-1} \left[\left\langle \boldsymbol{\mu}_h \boldsymbol{V}_{h+1}^*, \boldsymbol{\phi}(s_h^i, a_h^i)\right\rangle - V_{h+1}^*\left(s_{h+1}^i\right)\right].$$

Since $V_{h+1}^*(\cdot)$ is a fixed function, it is clear that $\boldsymbol{x}_i$ are $\mathcal{G}_i$-measurable and $\eta_i$ is $\mathcal{G}_{i+1}$-measurable. In addition, we have $\mathbb{E}\left[\eta_i \mid \mathcal{G}_i\right] = 0$. Since $\widehat{\sigma}_{i,h} \geqslant \sqrt{H}$, we see that $|\eta_i| \leqslant \sqrt{H}$ and $\|\boldsymbol{x}_i\|_2 \leqslant 1/\sqrt{H}$. Then, by Lemma C.1, with probability at least $1 - \delta/H$, for all $k \in [K]$ and fixed $h \in [H]$,

$$\left\|\sum_{i=1}^{k-1} \widehat{\sigma}_{i,h}^{-2} \boldsymbol{\phi}\left(s_h^i, a_h^i\right) {\boldsymbol{\epsilon}_h^i}^\top \boldsymbol{V}_{h+1}^*\right\|_{\widehat{\boldsymbol{\Lambda}}_{k,h}^{-1}} \leqslant \sqrt{H}\sqrt{d \log(1 + (k-1)/(Hd\lambda)) + \log(1/\delta)}$$

$$\leqslant \sqrt{H}\sqrt{d \log(1 + K/(Hd\lambda)) + \log(1/\delta)}$$

Using a similar argument as in Eq. (24), we have

$$\left\|(\widehat{\boldsymbol{\mu}}_{K+1,h} - \boldsymbol{\mu}_h) \boldsymbol{V}_{h+1}^*\right\|_{\widehat{\boldsymbol{\Lambda}}_{k,h}} \leqslant H\sqrt{\lambda d} + \left\|\sum_{i=1}^{k-1} \widehat{\sigma}_{i,h}^{-2} \boldsymbol{\phi}\left(s_h^i, a_h^i\right) {\boldsymbol{\epsilon}_h^i}^\top \boldsymbol{V}_{h+1}^*\right\|_{\widehat{\boldsymbol{\Lambda}}_{k,h}^{-1}}.$$

Therefore, with probability at least $1 - \delta/H$, for any $k \in [K]$ and fixed $h \in [H]$:

$$\left\|(\widehat{\boldsymbol{\mu}}_{K+1,h} - \boldsymbol{\mu}_h) \boldsymbol{V}_{h+1}^*\right\|_{\widehat{\boldsymbol{\Lambda}}_{k,h}} \leqslant \sqrt{H}\sqrt{d \log(1 + K/(Hd\lambda)) + \log(H/\delta)} + H\sqrt{\lambda d} = \widehat{\beta}^{P(1)}.$$

$\square$

**Lemma A.12.** *In Algorithm 2, for any $k \in [K]$ and fixed $h \in [H]$, under $\Psi_h^E \cap \Psi_{h+1}^P$, with probability at least $1 - \delta/H$:*

$$\boldsymbol{\mu}_h \in \widehat{\mathcal{C}}_h^{P(2)} = \left\{\boldsymbol{\mu} : \left\|(\boldsymbol{\mu} - \widehat{\boldsymbol{\mu}}_{K+1,h})\left(\widehat{\boldsymbol{V}}_{h+1} - \boldsymbol{V}_{h+1}^*\right)\right\|_{\widehat{\boldsymbol{\Lambda}}_{K+1,h}} \leqslant \widehat{\beta}^{P(2)}\right\}, \tag{31}$$

*where*

$$\widehat{\beta}^{P(2)} = 8\sqrt{\frac{H}{d} \log\left(1 + \frac{K}{Hd\lambda}\right)}$$

$$\cdot \sqrt{\log\left(\frac{4K^2 H}{\delta}\right) + d \log\left(1 + \frac{8K\sqrt{d}L_P}{H\sqrt{\lambda}}\right) + d^2 \log\left(1 + \frac{32d^{3/2}K^2 B_P^2}{H^2\lambda^2}\right)}$$

$$+ 4\sqrt{\frac{H}{d^3}}\left[\log(\frac{4K^2 H}{\delta}) + d \log\left(1 + \frac{8K\sqrt{d}L_P}{H\sqrt{\lambda}}\right) + d^2 \log\left(1 + \frac{32d^{3/2}K^2 B_P^2}{H^2\lambda^2}\right)\right]$$

$$+ H\sqrt{\lambda d} + 1$$

$$\tag{32}$$

*with $L_P = W + K/\lambda$ and an arbitrary $B_P \geqslant \widehat{\beta}^P$.*

*Proof.* (Lemma A.12) In Line 6 of Algorithm 2, for any $h \in [H]$, we have

$$\widehat{V}_h(s) = \min\left\{\max_{a \in \mathcal{A}}\left\{r_h(s,a) + \langle \widehat{\boldsymbol{\mu}}_{K+1,h} \widehat{\boldsymbol{V}}_{h+1}, \boldsymbol{\phi}(s,a)\rangle + b_h(s,a)\right\}, H\right\}$$

$$= \min\left\{\max_{a \in \mathcal{A}}\left\{\langle \boldsymbol{\theta}_h + \widehat{\boldsymbol{\mu}}_{K+1,h} \widehat{\boldsymbol{V}}_{h+1}, \boldsymbol{\phi}(s,a)\rangle + \widehat{\beta}^P \|\boldsymbol{\phi}(s,a)\|_{\widehat{\boldsymbol{\Lambda}}_{K+1,h}^{-1}}\right\}, H\right\}.$$

Moreover, we have

$$\left\|\boldsymbol{\theta}_h + \widehat{\boldsymbol{\mu}}_{K+1,h} \widehat{\boldsymbol{V}}_{h+1}\right\|_2 = \left\|\boldsymbol{\theta}_h + \widehat{\boldsymbol{\Lambda}}_{K+1,h}^{-1} \sum_{i=1}^K \widehat{\sigma}_{i,h}^{-2} \boldsymbol{\phi}(s_h^i, a_h^i) \widehat{V}_{h+1}(s_{h+1}^i)\right\|_2$$

$$\leqslant W + \frac{H}{(\sqrt{H})^2}\left\|\widehat{\boldsymbol{\Lambda}}_{K+1,h}^{-1} \sum_{i=1}^K \boldsymbol{\phi}(s_h^i, a_h^i)\right\|_2$$

$$\leqslant W + K/\lambda,$$

where the first inequality holds by the triangle inequality with $\|\boldsymbol{\theta}_h\|_2 \leq W$, $\widehat{V}_{h+1}(\cdot) \leq H$, and $\widehat{\sigma}_{i,h} \geq \sqrt{H}$ for any $i \in [K]$, and the second inequality holds since $\lambda_{\min}(\boldsymbol{\Lambda}_{K+1,h}) \geq \lambda$ and $\sup_{s,a} \|\boldsymbol{\phi}(s,a)\|_2 \leq 1$. This implies that $\widehat{V}_h \in \widehat{\mathcal{V}}(L_P, B_P)$ for any $h \in [H]$ with $L_P = W + K/\lambda$ and an arbitrary $B_P \geq \widehat{\beta}^P$, where function set $\widehat{\mathcal{V}}(\cdot, \cdot)$ is defined in Definition A.4.

For fixed $h \in [H]$, fixed $V(\cdot) \in \widehat{\mathcal{V}}(L_P, B_P)$ and constant $\zeta = H\sqrt{\lambda}/(2K\sqrt{d})$, let $\mathcal{G}_i = \mathcal{F}_{i,h}$, $x_i = \widehat{\sigma}_{i,h}^{-1}\boldsymbol{\phi}(s_h^i, a_h^i)$, and

$$\eta_i = \widehat{\sigma}_{i,h}^{-1}{\boldsymbol{\epsilon}_h^i}^\top \left(\boldsymbol{V} - \boldsymbol{V}_{h+1}^*\right) \cdot \mathbb{1}\left\{\boldsymbol{V}_{h+1}^* - \zeta \leq \boldsymbol{V} \leq \widehat{\boldsymbol{V}}_{i,h+1} + \boldsymbol{V}_{h+1}^* + \zeta\right\} \cdot \mathbb{1}\left\{\Psi_{i,h}^E\right\},$$

for any $i \in [k]$. Note that $V_{h+1}^*(\cdot)$ is a fixed function, $\widehat{V}_{i,h+1}(\cdot)$ and $\Psi_{i,h}^E$ are $\mathcal{G}_i$-measurable. Thus, $x_i$ is $\mathcal{G}_i$-measurable and $\eta_i$ is $\mathcal{G}_{i+1}$-measurable.

Also, we have $\mathbb{E}[\eta_i|\mathcal{G}_i] = 0$. By Lemma A.9, we have $\widehat{\sigma}_{i,h}^{-1}\min\left\{\|\widehat{\sigma}_{i,h}^{-1}\boldsymbol{\phi}(s_h^i, a_h^i)\|_{\widehat{\boldsymbol{\Lambda}}_{i,h}^{-1}}, 1\right\} \leq \frac{1}{\sqrt{Hd^3}}$. Since $|{\boldsymbol{\epsilon}_h^i}^\top (\boldsymbol{V} - \boldsymbol{V}_{h+1}^*)| \leq H$, we have $|\eta_i \min\{\|x_i\|_{\boldsymbol{\Lambda}_{i,h}^{-1}}, 1\}| \leq \sqrt{H}/\sqrt{d^3}$.

Furthermore, since $\widehat{\sigma}_{i,h}^2 \geq (d^2/H)W_{i,h}$, it holds that

$$\mathbb{E}[\eta_i^2|\mathcal{G}_i] = \widehat{\sigma}_{i,h}^{-2} \cdot [\mathbb{V}_h(V - V_{h+1}^*)](s_h^i, a_h^i) \cdot \mathbb{1}\left\{\boldsymbol{V}_{h+1}^* - \zeta \leq \boldsymbol{V} \leq \widehat{\boldsymbol{V}}_{i,h+1} + \boldsymbol{V}_{h+1}^* + \zeta\right\} \cdot \mathbb{1}\left\{\Psi_{i,h}^E\right\}$$
$$\leq \widehat{\sigma}_{i,h}^{-2} \cdot W_{i,h} \leq H/d^2,$$

where the first inequality holds by Lemma A.10.

By Lemma C.2, for fixed $h \in [H]$, with probability at least $1 - \delta/H$,

$$\Big\| \sum_{i=1}^K \widehat{\sigma}_{i,h}^{-2}\boldsymbol{\phi}(s_h^i, a_h^i){\boldsymbol{\epsilon}_h^i}^\top \left(\boldsymbol{V} - \boldsymbol{V}_{h+1}^*\right)$$
$$\cdot \mathbb{1}\left\{\boldsymbol{V}_{h+1}^* - \zeta \leq \boldsymbol{V} \leq \widehat{\boldsymbol{V}}_{i,h+1} + \boldsymbol{V}_{h+1}^* + \zeta\right\} \cdot \mathbb{1}\left\{\Psi_{i,h}^E\right\} \Big\|_{\widehat{\boldsymbol{\Lambda}}_{K+1,h}^{-1}}$$
$$\leq 8\sqrt{\frac{H}{d}\log\left(1 + \frac{K}{Hd\lambda}\right)\log\left(\frac{4K^2H}{\delta}\right)} + 4\sqrt{\frac{H}{d^3}\log(\frac{4K^2H}{\delta})}.$$

Denote the $\zeta$-cover of function class $\widehat{\mathcal{V}}(L_P, B_P)$ as $\widehat{\mathcal{N}}_\zeta$, we have

$$\log|\widehat{\mathcal{N}}_\zeta| \leq d\log(1 + 4L_P/\zeta) + d^2\log(1 + 8d^{1/2}B_P^2/(\lambda\zeta^2)),$$

according to Lemma C.5. Here $L_P = W + K/\lambda$ and $B_P \geq \widehat{\beta}^P$. Then for any $k \in [K]$ and fixed $h \in [H]$, with probability at least $1 - \delta/H$, for any $\boldsymbol{V} \in \widehat{\mathcal{N}}_\zeta$, conditioned on $\Psi_h^E$ (i.e. $\mathbb{1}\left\{\Psi_{i,h}^E\right\} = 1$ for any $i \in [k]$), we can build our argument under $\Psi_{h+1}$ in the following), we have

$$\left\| \sum_{i=1}^K \widehat{\sigma}_{i,h}^{-2}\boldsymbol{\phi}(s_h^i, a_h^i){\boldsymbol{\epsilon}_h^i}^\top \left(\boldsymbol{V} - \boldsymbol{V}_{h+1}^*\right) \cdot \mathbb{1}\left\{\boldsymbol{V}_{h+1}^* - \zeta \leq \boldsymbol{V} \leq \widehat{\boldsymbol{V}}_{i,h+1} + \boldsymbol{V}_{h+1}^* + \zeta\right\} \right\|_{\widehat{\boldsymbol{\Lambda}}_{K+1,h}^{-1}}$$
$$\leq 8\sqrt{\frac{H}{d}\log\left(1 + \frac{K}{Hd\lambda}\right)\left(\log\left(\frac{4K^2H}{\delta}\right) + \log|\mathcal{N}_\zeta|\right)} + 4\sqrt{\frac{H}{d^3}\left(\log(\frac{4K^2H}{\delta}) + \log|\mathcal{N}_\zeta|\right)}.$$
$$(33)$$

For any $f(\cdot) \in \widehat{\mathcal{V}}(L_P, B_P)$, there exists a $V(\cdot) \in \widehat{\mathcal{N}}_\zeta$, such that $\|\boldsymbol{f} - \boldsymbol{V}\|_\infty \leq \zeta$. Since $\|{\boldsymbol{\epsilon}_h^i}^\top(\boldsymbol{f} - \boldsymbol{V})\|_2 \leq 2\zeta$ and $\left\|\sum_{i=1}^K \widehat{\sigma}_{i,h}^{-2}\boldsymbol{\phi}(s_h^i, a_h^i){\boldsymbol{\epsilon}_h^i}^\top \cdot \mathbb{1}\left\{\boldsymbol{V}_{h+1}^* - \zeta \leq \boldsymbol{V} \leq \widehat{\boldsymbol{V}}_{i,h+1} + \boldsymbol{V}_{h+1}^* + \zeta\right\}\right\|_{\widehat{\boldsymbol{\Lambda}}_{K+1,h}^{-1}} \leq K\sqrt{d}/(H\sqrt{\lambda})$, we have

$$\left\| \sum_{i=1}^K \widehat{\sigma}_{i,h}^{-2}\boldsymbol{\phi}(s_h^i, a_h^i){\boldsymbol{\epsilon}_h^i}^\top(\boldsymbol{f} - \boldsymbol{V}) \cdot \mathbb{1}\left\{\boldsymbol{V}_{h+1}^* - \zeta \leq \boldsymbol{V} \leq \widehat{\boldsymbol{V}}_{i,h+1} + \boldsymbol{V}_{h+1}^* + \zeta\right\} \right\|_{\widehat{\boldsymbol{\Lambda}}_{K+1,h}^{-1}} \leq \frac{2\zeta K\sqrt{d}}{H\sqrt{\lambda}}.$$
$$(34)$$

Conditioning on $\Psi_h^E$, we have

$$
\left\| \sum_{i=1}^{K} \widehat{\sigma}_{i,h}^{-2} \phi(s_h^i, a_h^i) {\boldsymbol{\epsilon}_h^i}^{\top} \left( \boldsymbol{f} - \boldsymbol{V}_{h+1}^* \right) \cdot \mathbb{1} \left\{ \boldsymbol{V}_{h+1}^* - \zeta \leqslant \boldsymbol{V} \leqslant \widehat{\boldsymbol{V}}_{i,h+1} + \boldsymbol{V}_{h+1}^* + \zeta \right\} \right\|_{\widehat{\Lambda}_{K+1,h}^{-1}}
$$

$$
\leqslant \left\| \sum_{i=1}^{K} \widehat{\sigma}_{i,h}^{-2} \phi(s_h^i, a_h^i) {\boldsymbol{\epsilon}_h^i}^{\top} \left( \boldsymbol{V} - \boldsymbol{V}_{h+1}^* \right) \cdot \mathbb{1} \left\{ \boldsymbol{V}_{h+1}^* - \zeta \leqslant \boldsymbol{V} \leqslant \widehat{\boldsymbol{V}}_{i,h+1} + \boldsymbol{V}_{h+1}^* + \zeta \right\} \right\|_{\widehat{\Lambda}_{K+1,h}^{-1}}
$$

$$
+ \left\| \sum_{i=1}^{K} \widehat{\sigma}_{i,h}^{-2} \phi(s_h^i, a_h^i) {\boldsymbol{\epsilon}_h^i}^{\top} (\boldsymbol{f} - \boldsymbol{V}) \cdot \mathbb{1} \left\{ \boldsymbol{V}_{h+1}^* - \zeta \leqslant \boldsymbol{V} \leqslant \widehat{\boldsymbol{V}}_{i,h+1} + \boldsymbol{V}_{h+1}^* + \zeta \right\} \right\|_{\widehat{\Lambda}_{K+1,h}^{-1}}
$$

$$
\leqslant \left\| \sum_{i=1}^{K} \widehat{\sigma}_{i,h}^{-2} \phi(s_h^i, a_h^i) {\boldsymbol{\epsilon}_h^i}^{\top} \left( \boldsymbol{V} - \boldsymbol{V}_{h+1}^* \right) \cdot \mathbb{1} \left\{ \boldsymbol{V}_{h+1}^* - \zeta \leqslant \boldsymbol{V} \leqslant \widehat{\boldsymbol{V}}_{i,h+1} + \boldsymbol{V}_{h+1}^* + \zeta \right\} \right\|_{\widehat{\Lambda}_{K+1,h}^{-1}}
$$

$$
+ \frac{2\zeta K \sqrt{d}}{H \sqrt{\lambda}}
$$

$$
\leqslant 8 \sqrt{\frac{H}{d} \log \left( 1 + \frac{K}{H d \lambda} \right) \left( \log \left( \frac{4 K^2 H}{\delta} \right) + \log |\mathcal{N}_\zeta| \right)} + 4 \sqrt{\frac{H}{d^3} \left( \log(\frac{4 K^2 H}{\delta}) + \log |\mathcal{N}_\zeta| \right)}
$$

$$
+ \frac{2\zeta K \sqrt{d}}{H \sqrt{\lambda}}.
$$

$$(35)$$

Here the first inequality is due to triangle inequality, the second holds by Eq. (34), and the third holds by Eq. (33).

We further assume $\Psi_{h+1}^P$ holds. In addition, for any $\widehat{V}_{h+1}(\cdot)$, there exists a $V'(\cdot) \in \widehat{\mathcal{N}}_\zeta$ such that $\left\| \widehat{\boldsymbol{V}}_{h+1} - \boldsymbol{V}' \right\|_\infty \leqslant \zeta$. Since $\zeta = H\sqrt{\lambda}/(2K\sqrt{d})$, we have

$$
\boldsymbol{V}' \leqslant \widehat{\boldsymbol{V}}_{h+1} + \zeta \leqslant \boldsymbol{V}_{h+1}^* + \widehat{\boldsymbol{V}}_{i,h+1} + \zeta
$$

for any $i \in [K]$ where the second inequality holds by Lemma A.15 under $\Psi_{h+1}^E \cap \Psi_{h+1}^P$. On the other hand, $\boldsymbol{V}_{h+1}^* - \zeta \leqslant \widehat{\boldsymbol{V}}_{h+1} - \zeta \leqslant \boldsymbol{V}'$, where the first inequality holds by Lemma A.14 under $\Psi_{h+1}^P$. Thus, $\mathbb{1} \left\{ \boldsymbol{V}_{h+1}^* - \zeta \leqslant \boldsymbol{V} \leqslant \widehat{\boldsymbol{V}}_{i,h+1} + \boldsymbol{V}_{h+1}^* + \zeta \right\} = 1$ for any $i \in [K]$ under $\Psi_{h+1}^E \cap \Psi_{h+1}^P$.

Moreover, we have that, with probability at least $1 - \delta/H$, under $\Psi_h^E \cap \Psi_{h+1}^P$, for any $k \in [K]$ and fixed $h \in [H]$:

$$
\left\| (\widehat{\boldsymbol{\mu}}_{K+1,h} - \boldsymbol{\mu}_h) \left( \widehat{\boldsymbol{V}}_{h+1} - \boldsymbol{V}_{h+1}^* \right) \right\|_{\widehat{\Lambda}_{K+1,h}}
$$

$$
\leqslant \sqrt{\lambda} H \sqrt{d} + \left\| \sum_{i=1}^{K} \widehat{\sigma}_{i,h}^{-2} \phi(s_h^i, a_h^i) {\boldsymbol{\epsilon}_h^i}^{\top} \left( \widehat{\boldsymbol{V}}_{h+1} - \boldsymbol{V}_{h+1}^* \right) \right\|_{\widehat{\Lambda}_{K+1,h}^{-1}}
$$

$$
\leqslant 8 \sqrt{\frac{H}{d} \log \left( 1 + \frac{K}{H d \lambda} \right)}
$$

$$
\cdot \sqrt{\log \left( \frac{4 K^2 H}{\delta} \right) + d \log \left( 1 + \frac{8 K \sqrt{d} L_P}{H \sqrt{\lambda}} \right) + d^2 \log \left( 1 + \frac{32 d^{3/2} K^2 B_P^2}{H^2 \lambda^2} \right)}
$$

$$
+ 4 \sqrt{\frac{H}{d^3} \left( \log(\frac{4 K^2 H}{\delta}) + d \log \left( 1 + \frac{8 K \sqrt{d} L_P}{H \sqrt{\lambda}} \right) + d^2 \log \left( 1 + \frac{32 d^{3/2} K^2 B_P^2}{H^2 \lambda^2} \right) \right)}
$$

$$
+ H \sqrt{\lambda d} + 1
$$

$$
= \widehat{\beta}^{P(2)}.
$$

Here the first inequality holds similarly as Eq. (24), and the second inequality holds by Eq. (35) and $\zeta = H\sqrt{\lambda}/(2K\sqrt{d})$. $\qquad\square$

**Lemma A.13.** *Set $\widehat{\beta}^P = \widehat{\beta}^{P(1)} + \widehat{\beta}^{P(2)}$ in Eq equation 16, for any $\delta \in (0,1)$, with probability at least $1 - 3\delta$, $\Psi_1^E \cap \Psi_1^P$ holds, i.e., for any $h \in [H]$ in the planning phase,*

$$\boldsymbol{\mu}_h \in \widehat{\mathcal{C}}_h^P = \left\{ \boldsymbol{\mu} : \left\| (\boldsymbol{\mu} - \widehat{\boldsymbol{\mu}}_{K+1,h}) \widehat{\boldsymbol{V}}_{h+1} \right\|_{\widehat{\Lambda}_{K+1,h}} \leqslant \widehat{\beta}^P \right\}.$$

*Proof.* (Lemma A.13) We first prove the following claim:

*Under $\Psi_1^E$, for fixed $h \in [H]$ and any $k \in [K]$, with probability at least $1 - 2(H-h)\delta/H$, $\Psi_h^P$ holds.*

We prove this claim by induction. Firstly, when $h = H$ the result is trivial. Assume the claim holds for $h + 1 \leqslant H$. Then, for any $k \in [K]$, under $\Psi_1^E$, with probability $1 - 2(H - h + 1)\delta/H$, $\Psi_{h+1}^P$ holds.

Subsequently, for any $k \in [K]$, by Lemma A.11 and A.12 under $\Psi_1^E \cap \Psi_{h+1}^P$, we have with probability $1 - 2\delta/H$ that

$$\boldsymbol{\mu}_h \in \widehat{\mathcal{C}}_h^{P(1)} \cap \widehat{\mathcal{C}}_h^{P(2)}$$

By taking the union bound, for any $k \in [K]$, under $\Psi_1^E$, with probability $1 - 2(H-h)\delta/H$, $\Psi_h^P$ holds, which means the claim holds for $h$. Thus, the claim is proved by induction.

Since $\mathbb{P}\{\Psi_1^E\} \geqslant 1 - \delta$ by Lemma A.6, the conclusion is obtained by setting $h = 1$ and taking a union bound. $\qquad\square$

### A.3.2 Optimism in Planning Phase

**Lemma A.14** (Optimism in Planning Phase). *In Algorithm 2, for any $h \in [H]$, under $\Psi_h^P$, we have*

$$V_h^*(s; r) \leqslant \widehat{V}_h(s), \quad \forall s \in \mathcal{S}.$$

*Proof.* We prove the optimism by introduction. Notice that the statement holds trivially when $h = H + 1$ since $V_{H+1}^*(\cdot; r) = \widehat{V}_{H+1}(\cdot) = 0$.

Assume the statement holds for $h + 1$, which means $V_{h+1}^*(\cdot; r) \leqslant \widehat{V}_{h+1}(\cdot)$ under $\Psi_{h+1}^P$.

Since

$$\widehat{Q}_h(\cdot, \cdot) = r_h(\cdot, \cdot) + \langle \widehat{\boldsymbol{\mu}}_{K+1,h} \widehat{\boldsymbol{V}}_{h+1}, \boldsymbol{\phi}(\cdot, \cdot) \rangle + \widehat{\beta}^P \|\boldsymbol{\phi}(\cdot, \cdot)\|_{\widehat{\Lambda}_{K+1,h}^{-1}},$$

$$Q_h^*(\cdot, \cdot; r) = r_h(\cdot, \cdot) + \mathbb{P}_h V_{h+1}^*(\cdot, \cdot; r),$$

we have for any $(s, a) \in \mathcal{S} \times \mathcal{A}$ that

$$\begin{aligned}
&\widehat{Q}_h(s, a) - Q_h^*(s, a; r) \\
=& r_h(s, a) + \langle \widehat{\boldsymbol{\mu}}_{K+1,h} \widehat{\boldsymbol{V}}_{h+1}, \boldsymbol{\phi}(s, a) \rangle + \widehat{\beta}^P \|\boldsymbol{\phi}(s, a)\|_{\widehat{\Lambda}_{K+1,h}^{-1}} - \left[ r_h(s, a) + \mathbb{P}_h V_{h+1}^*(s, a; r) \right] \\
=& \langle \widehat{\boldsymbol{\mu}}_{K+1,h} \widehat{\boldsymbol{V}}_{h+1}, \boldsymbol{\phi}(s, a) \rangle - \langle \boldsymbol{\mu}_h \widehat{\boldsymbol{V}}_{h+1}, \boldsymbol{\phi}(s, a) \rangle + \widehat{\beta}^P \|\boldsymbol{\phi}(s, a)\|_{\widehat{\Lambda}_{K+1,h}^{-1}} \\
& + \mathbb{P}_h \widehat{V}_{h+1}(s, a) - \mathbb{P}_h V_{h+1}^*(s, a; r) \\
\geqslant& - \left\| (\widehat{\boldsymbol{\mu}}_{K+1,h} - \boldsymbol{\mu}_h) \widehat{\boldsymbol{V}}_{h+1} \right\|_{\widehat{\Lambda}_{K+1,h}} \|\boldsymbol{\phi}(s, a)\|_{\widehat{\Lambda}_{K+1,h}^{-1}} + \widehat{\beta}^P \|\boldsymbol{\phi}(s, a)\|_{\widehat{\Lambda}_{K+1,h}^{-1}} \\
& + \mathbb{P}_h \widehat{V}_{h+1}(s, a) - \mathbb{P}_h V_{h+1}^*(s, a; r) \\
\geqslant& \mathbb{P}_h \widehat{V}_{h+1}(s, a) - \mathbb{P}_h V_{h+1}^*(s, a; r) \\
\geqslant& 0,
\end{aligned}$$

where the first inequality is due to Cauchy-Schwarz inequality, the second inequality holds by $\boldsymbol{\mu}_h \in \widehat{\mathcal{C}}_h^P$ under $\Psi_h^P$, the last inequality holds by the induction assumption $\widehat{V}_{h+1}(\cdot) \geqslant V_{h+1}^*(\cdot; r)$ under $\Psi_{h+1}^P$ and $\mathbb{P}_h$ is a valid distribution. Therefore, $\widehat{V}_h(\cdot) = \max_{a \in \mathcal{A}} \widehat{Q}_h(\cdot, a) \geqslant \max_{a \in \mathcal{A}} Q_h^*(\cdot, a; r) = V_h^*(\cdot; r)$. $\qquad\square$

**Lemma A.15** (Over-optimism Between Two Phases). *In Algoirhms 1 and 2, for any and any $h \in [H]$, under $\Psi_h^E \cap \Psi_h^P$, we have*

$$V_h^*(s; r) + \widehat{V}_{k,h}(s) \geqslant \widehat{V}_h(s), \quad \forall s \in \mathcal{S}.$$

*Proof.* (Lemma A.15) We first prove the conclusion by introduction for some $k \in [K]$.

Notice that the statement holds trivially when $h = H+1$ since $V_{H+1}^*(\cdot; r) + \widehat{V}_{k,H+1}(\cdot) = \widehat{V}_{H+1}(\cdot) = 0$ by definitions. Assume the statement holds for $h + 1$, which means $V_{h+1}^*(\cdot; r) + \widehat{V}_{k,h+1}(\cdot) \geqslant \widehat{V}_{h+1}(\cdot)$ under $\Psi_{h+1}^E \cap \Psi_{h+1}^P$.

We see that

$$Q_h^*(\cdot, \cdot; r) = r_h(\cdot, \cdot) + \mathbb{P}_h V_{h+1}^*(\cdot, \cdot; r),$$
$$\widehat{Q}_{k,h}(\cdot, \cdot) = r_{k,h}(\cdot, \cdot) + \langle \widehat{\boldsymbol{\mu}}_{k,h} \widehat{\boldsymbol{V}}_{k,h+1}, \boldsymbol{\phi}(\cdot, \cdot) \rangle + \widehat{\beta}^E \|\boldsymbol{\phi}(\cdot, \cdot)\|_{\widehat{\boldsymbol{\Lambda}}_{k,h}^{-1}},$$
$$\widehat{Q}_h(\cdot, \cdot) = r_h(\cdot, \cdot) + \langle \widehat{\boldsymbol{\mu}}_{K+1,h} \widehat{\boldsymbol{V}}_{h+1}, \boldsymbol{\phi}(\cdot, \cdot) \rangle + \widehat{\beta}^P \|\boldsymbol{\phi}(\cdot, \cdot)\|_{\widehat{\boldsymbol{\Lambda}}_{K+1,h}^{-1}},$$

Thus, we have for any $(s, a) \in \mathcal{S} \times \mathcal{A}$ that,

$$Q_h^*(s, a; r) + \widehat{Q}_{k,h}(s, a) - \widehat{Q}_h(s, a)$$
$$= r_h(s, a) + \mathbb{P}_h V_{h+1}^*(s, a; r) + r_{k,h}(s, a) + \widehat{\mathbb{P}}_{k,h} \widehat{V}_{k,h+1}(s, a) + \widehat{\beta}^E \|\boldsymbol{\phi}(s, a)\|_{\widehat{\boldsymbol{\Lambda}}_{k,h}^{-1}}$$
$$\quad - \left[ r_h(s, a) + \langle \widehat{\boldsymbol{\mu}}_{K+1,h} \widehat{\boldsymbol{V}}_{h+1}, \boldsymbol{\phi}(s, a) \rangle + \widehat{\beta}^P \|\boldsymbol{\phi}(s, a)\|_{\widehat{\boldsymbol{\Lambda}}_{K+1,h}^{-1}} \right]$$
$$= \left[ \mathbb{P}_h V_{h+1}^*(s, a) + \mathbb{P}_h \widehat{V}_{k,h+1}(s, a) - \mathbb{P}_h \widehat{V}_{h+1}(s, a) \right] + 3\widehat{\beta}^E \|\boldsymbol{\phi}(s, a)\|_{\widehat{\boldsymbol{\Lambda}}_{k,h}^{-1}} - \widehat{\beta}^P \|\boldsymbol{\phi}(s, a)\|_{\widehat{\boldsymbol{\Lambda}}_{K+1,h}^{-1}}$$
$$\quad + \langle \widehat{\boldsymbol{\mu}}_{k,h} \widehat{\boldsymbol{V}}_{k,h+1}, \boldsymbol{\phi}(s, a) \rangle - \langle \boldsymbol{\mu}_h \widehat{\boldsymbol{V}}_{k,h+1}, \boldsymbol{\phi}(s, a) \rangle$$
$$\quad + \langle \boldsymbol{\mu}_h \widehat{\boldsymbol{V}}_{h+1}, \boldsymbol{\phi}(s, a) \rangle - \langle \widehat{\boldsymbol{\mu}}_{K+1,h} \widehat{\boldsymbol{V}}_{h+1}, \boldsymbol{\phi}(s, a) \rangle$$
$$\geqslant \left[ \mathbb{P}_h V_{h+1}^*(s, a) + \mathbb{P}_h \widehat{V}_{k,h+1}(s, a) - \mathbb{P}_h \widehat{V}_{h+1}(s, a) \right] + 3\widehat{\beta}^E \|\boldsymbol{\phi}(s, a)\|_{\widehat{\boldsymbol{\Lambda}}_{k,h}^{-1}} - \widehat{\beta}^P \|\boldsymbol{\phi}(s, a)\|_{\widehat{\boldsymbol{\Lambda}}_{K+1,h}^{-1}}$$
$$\quad - \left\| (\widehat{\boldsymbol{\mu}}_{k,h} - \boldsymbol{\mu}_h) \widehat{\boldsymbol{V}}_{k,h+1} \right\|_{\widehat{\boldsymbol{\Lambda}}_{k,h}} \|\boldsymbol{\phi}(s, a)\|_{\widehat{\boldsymbol{\Lambda}}_{k,h}^{-1}} - \left\| (\widehat{\boldsymbol{\mu}}_{K+1,h} - \boldsymbol{\mu}_h) \widehat{\boldsymbol{V}}_{h+1} \right\|_{\widehat{\boldsymbol{\Lambda}}_{K+1,h}} \|\boldsymbol{\phi}(s, a)\|_{\widehat{\boldsymbol{\Lambda}}_{K+1,h}^{-1}}$$
$$\geqslant \left[ \mathbb{P}_h V_{h+1}^*(s, a) + \mathbb{P}_h \widehat{V}_{k,h+1}(s, a) - \mathbb{P}_h \widehat{V}_{h+1}(s, a) \right] + 3\widehat{\beta}^E \|\boldsymbol{\phi}(s, a)\|_{\widehat{\boldsymbol{\Lambda}}_{k,h}^{-1}} - \widehat{\beta}^P \|\boldsymbol{\phi}(s, a)\|_{\widehat{\boldsymbol{\Lambda}}_{K+1,h}^{-1}}$$
$$\quad - \widehat{\beta}^E \|\boldsymbol{\phi}(s, a)\|_{\widehat{\boldsymbol{\Lambda}}_{k,h}^{-1}} - \widehat{\beta}^P \|\boldsymbol{\phi}(s, a)\|_{\widehat{\boldsymbol{\Lambda}}_{K+1,h}^{-1}}$$
$$\geqslant \left[ \mathbb{P}_h V_{h+1}^*(s, a) + \mathbb{P}_h \widehat{V}_{k,h+1}(s, a) - \mathbb{P}_h \widehat{V}_{h+1}(s, a) \right] \geqslant 0,$$

where the first inequality is due to Cauchy-Schwarz inequality, the second inequality holds by $\boldsymbol{\mu}_h \in \widehat{\mathcal{C}}_{k,h}^E \cap \widehat{\mathcal{C}}_h^P$ under $\Psi_h^E \cap \Psi_h^P$, the third inequality holds since $\widehat{\boldsymbol{\Lambda}}_{K+1,h} \succeq \widehat{\boldsymbol{\Lambda}}_{k,h}$ and $\widehat{\beta}^E \geqslant \widehat{\beta}^P$, and the last inequality holds by the induction assumption $V_{h+1}^*(\cdot; r) + \widehat{V}_{k,h+1}(\cdot) \geqslant \widehat{V}_{h+1}(\cdot)$ under $\Psi_{h+1}^E \cap \Psi_{h+1}^P$ and $\mathbb{P}_h$ is a valid distribution. Therefore,

$$V_h^*(\cdot; r) + \widehat{V}_{k,h}(\cdot) = \max_{a \in \mathcal{A}} Q_h^*(\cdot, a; r) + \max_{a \in \mathcal{A}} \widehat{Q}_{k,h}(\cdot, a) \geqslant \max_{a \in \mathcal{A}} \widehat{Q}_h(\cdot, a) = \widehat{V}_h(\cdot)$$

Finally, note that the same argument can be extended to any $k \in [K]$. $\qquad\square$

## A.4 UPPER BOUNDING THE EXPLORATION ERROR

In this subsection, we bound the exploration error in Lemma A.20. Before that, we provide a simulation lemma in linear MDPs in Lemma A.17. Besides, we denote the reward function $\{r_{k,h}\}_{h=1}^H$ in Algorithm 1 at episode $k$ as $r_k$, for all $k \in [K]$.

**Definition A.16** (Trajectory Distribution). For a fixed policy $\pi = \{\pi_1, \pi_2, ..., \pi_H\}$, define $d_h^\pi(s_h)$ as the probability measure over trajectory $\tau_h = (s_h, a_h, s_{h+1}, a_{h+1}, \cdots, s_H, a_H)$ induced by following $\pi$ starting at $s_h$ at stage $h$:

$$d_h^\pi(s_h)(\tau_h) := \prod_{t=h+1}^{H} \mathbb{P}_t^\pi(s_t, a_t \mid s_{t-1}, a_{t-1}),$$

where $\mathbb{P}_t^\pi(s_t, a_t \mid s_{t-1}, a_{t-1})$ means the probability that the probability of stat-action pair transition from $(s_{t-1}, a_{t-1})$ to $(s_t, a_t)$ in a trajectory $\tau_h = (s_h, a_h, s_{h+1}, a_{h+1}, \cdots, s_H, a_H)$ started at $s_h$ at stage $h$ by following policy $\pi$.

**Lemma A.17** (Simulation Lemma). *In Algorithm 1, for any $k \in [K]$ and any $h \in [H]$, under $\Psi_{k,h}^E$, we have*

$$0 \leqslant \widehat{V}_{k,h}(s_h^k) \leqslant \mathbb{E}_{\tau_h^k \sim d_h^{\pi^k}(s_h^k)} \min \left\{ \sum_{h'=h}^{H} 4\widehat{\beta}^E \left\| \phi(s_{h'}^k, a_{h'}^k) \right\|_{\widehat{\mathbf{\Lambda}}_{k,h'}^{-1}}, H \right\}$$

*Proof.* (Lemma A.17) First, recall that

$$\widehat{Q}_{k,h}(\cdot, \cdot) = r_{k,h}(\cdot, \cdot) + \langle \widehat{\boldsymbol{\mu}}_{k,h} \widehat{V}_{k,h+1}, \phi(\cdot, \cdot) \rangle + \widehat{\beta}^E \|\phi(\cdot, \cdot)\|_{\widehat{\mathbf{\Lambda}}_{k,h}^{-1}}$$

$$= r_{k,h}(\cdot, \cdot) + \widehat{\mathbb{P}}_{k,h} \widehat{V}_{k,h+1}(s_h^k, a_h^k) + \widehat{\beta}^E \|\phi(\cdot, \cdot)\|_{\widehat{\mathbf{\Lambda}}_{k,h}^{-1}}$$

$$\widehat{V}_{k,h}(\cdot) = \min \left\{ \max_{a \in \mathcal{A}} \widehat{Q}_{k,h}(\cdot, a), H \right\}$$

Thus, for any $k \in [K]$ and any $h \in [H]$ in Algorithm 1, we have

$$\widehat{V}_{k,h}(s_h^k) \leqslant \widehat{Q}_{k,h}(s_h^k, a_h^k) = r_{k,h}(s_h^k, a_h^k) + \widehat{\mathbb{P}}_{k,h} \widehat{V}_{k,h+1}(s_h^k, a_h^k) + \widehat{\beta}^E \|\phi(s_h^k, a_h^k)\|_{\widehat{\mathbf{\Lambda}}_{k,h}^{-1}}$$

$$= 3\widehat{\beta}^E \|\phi(s_h^k, a_h^k)\|_{\widehat{\mathbf{\Lambda}}_{k,h}^{-1}} + \left[ \widehat{\mathbb{P}}_{k,h} \widehat{V}_{k,h+1}(s_h^k, a_h^k) - \mathbb{P}_h \widehat{V}_{k,h+1}(s_h^k, a_h^k) \right] + \mathbb{P}_h \widehat{V}_{k,h+1}(s_h^k, a_h^k)$$

$$\cdots$$

$$\overset{(a)}{=} \mathbb{E}_{\tau_h^k \sim d_h^{\pi^k}(s_h^k)} \left[ \sum_{h'=h}^{H} \widehat{\mathbb{P}}_{k,h'} \widehat{V}_{k,h'+1}(s_{h'}^k, a_{h'}^k) - \mathbb{P}_{h'} \widehat{V}_{k,h'+1}(s_{h'}^k, a_{h'}^k) + 3\widehat{\beta}^E \left\| \phi(s_{h'}^k, a_{h'}^k) \right\|_{\widehat{\mathbf{\Lambda}}_{k,h}^{-1}} \right]$$

$$= \mathbb{E}_{\tau_h^k \sim d_h^{\pi^k}(s_h^k)} \left[ \sum_{h'=h}^{H} \left\langle (\widehat{\boldsymbol{\mu}}_{k,h'} - \boldsymbol{\mu}_{h'}) \widehat{V}_{k,h'+1}, \phi(s_{h'}^k, a_{h'}^k) \right\rangle + 3\widehat{\beta}^E \left\| \phi(s_{h'}^k, a_{h'}^k) \right\|_{\widehat{\mathbf{\Lambda}}_{k,h'}^{-1}} \right]$$

$$\leqslant \mathbb{E}_{\tau_h^k \sim d_h^{\pi^k}(s_h^k)} \left[ \sum_{h'=h}^{H} 4\widehat{\beta}^E \left\| \phi(s_{h'}^k, a_{h'}^k) \right\|_{\widehat{\mathbf{\Lambda}}_{k,h'}^{-1}} \right],$$

where the equality $(a)$ holds since we can expand $\widehat{V}_{k,h+1}(\cdot)$ in a recursive way until $\widehat{V}_{k,H+1}(\cdot)$ and the expectation is taken over trajectory distribution $d_h^{\pi^k}(s_h^k)$ over trajectory $\tau_h^k = (s_h^k, a_h^k, s_{h+1}^k, a_{h+1}^k, \ldots, s_H^k, a_H^k)$, and the last inequality holds since $\boldsymbol{\mu}_{h'} \in \mathcal{C}_{k,h'}^E$ for any $h \leqslant h' \leqslant H$ under $\Psi_{k,h}^E$. Since $\widehat{V}_{k,h}(s_h^k) \leqslant H$ and $\widehat{V}_{k,h}(s_h^k) \geqslant V_h^*(s_h^k; r_k) \geqslant 0$ by Lemma A.7 under $\Psi_{k,h}^E$, the conclusion is obtained. $\square$

**Lemma A.18.** *Fix $\delta > 0$. In Algorithm 1, under $\Psi_1^E$, we have with probability $1 - 2\delta$,*

$$\sum_{k=1}^{K} \sum_{h=1}^{H} \mathbb{P}_h \widehat{V}_{k,h+1}(s_h^k, a_h^k)$$

$$\leqslant 4H\widehat{\beta}^E \sqrt{\sum_{k=1}^{K} \sum_{h=1}^{H} \widehat{\sigma}_{k,h}^2} \sqrt{\sum_{k=1}^{K} \sum_{h=1}^{H} \min\{\|\phi(s_h^k, a_h^k)\|_{\widehat{\mathbf{\Lambda}}_{k,h}^{-1}}^2, 1\}} + (H^2 + H)\sqrt{2T \log(H/\delta)}.$$

(36)

*Proof.* First, we have

$$\sum_{k=1}^{K}\sum_{h=1}^{H}\mathbb{P}_h\widehat{V}_{k,h+1}(s_h^k,a_h^k) = \sum_{k=1}^{K}\sum_{h=1}^{H}\widehat{V}_{k,h+1}(s_{h+1}^k) + \left[\mathbb{P}_h\widehat{V}_{k,h+1}(s_h^k,a_h^k) - \widehat{V}_{k,h+1}(s_{h+1}^k)\right] \tag{37}$$
$$\leqslant \sum_{k=1}^{K}\sum_{h=1}^{H}\widehat{V}_{k,h+1}(s_{h+1}^k) + H\sqrt{2T\log(H/\delta)},$$

holds with probability $1-\delta$, by common Hoeffding inequality, as stated in Lemma C.3.

Notice that

$$\begin{aligned}\widehat{V}_{k,h}(s_h^k) &= \min\left\{\widehat{Q}_{k,h}(s_h^k,a_h^k), H\right\}\\&= \min\left\{3\widehat{\beta}^E\|\boldsymbol{\phi}(s_h^k,a_h^k)\|_{\widehat{\boldsymbol{\Lambda}}_{k,h}^{-1}} + \langle\widehat{\boldsymbol{\mu}}_{k,h}\widehat{\boldsymbol{V}}_{k,h+1}, \boldsymbol{\phi}(s_h^k,a_h^k)\rangle, H\right\}\\&\leqslant \min\left\{4\widehat{\beta}^E\|\boldsymbol{\phi}(s_h^k,a_h^k)\|_{\widehat{\boldsymbol{\Lambda}}_{k,h}^{-1}} + \mathbb{P}_h\widehat{V}_{k,h+1}(s_h^k,a_h^k), H\right\}\\&= \min\left\{4\widehat{\beta}^E\|\boldsymbol{\phi}(s_h^k,a_h^k)\|_{\widehat{\boldsymbol{\Lambda}}_{k,h}^{-1}} + \widehat{V}_{k,h+1}(s_{h+1}^k) + [\mathbb{P}_h\widehat{V}_{k,h+1}(s_h^k,a_h^k) - \widehat{V}_{k,h+1}(s_{h+1}^k)], H\right\}\end{aligned} \tag{38}$$

where the inequality holds under $\widehat{\Psi}_1^E$. For fixed $h$, We can recursively use this method and get

$$\begin{aligned}\sum_{k=1}^{K}\widehat{V}_{k,h}(s_h^k) &\leqslant \sum_{k=1}^{K}\min\Big\{\sum_{h'=h}^{H}4\widehat{\beta}^E\|\boldsymbol{\phi}(s_{h'}^k,a_{h'}^k)\|_{\widehat{\boldsymbol{\Lambda}}_{k,h'}^{-1}}\\&\qquad\qquad + [\mathbb{P}_{h'}\widehat{V}_{k,h'+1}(s_{h'}^k,a_{h'}^k) - \widehat{V}_{k,h'+1}(s_{h'+1}^k)], H\Big\}\\&\leqslant \sum_{k=1}^{K}\sum_{h'=h}^{H}4\widehat{\beta}^E\widehat{\sigma}_{k,h'}\min\{\|\boldsymbol{\phi}(s_{h'}^k,a_{h'}^k)\|_{\widehat{\boldsymbol{\Lambda}}_{k,h'}^{-1}},1\}\\&\qquad + \sum_{k=1}^{K}\sum_{h'=h}^{H}\mathbb{P}_{h'}\widehat{V}_{k,h'+1}(s_{h'}^k,a_{h'}^k) - \widehat{V}_{k,h'+1}(s_{h'+1}^k)\\&\leqslant 4\widehat{\beta}^E\sqrt{\sum_{k=1}^{K}\sum_{h'=h}^{H}\widehat{\sigma}_{k,h'}^2}\sqrt{\sum_{k=1}^{K}\sum_{h'=h}^{H}\min\{\|\boldsymbol{\phi}(s_{h'}^k,a_{h'}^k)\|_{\widehat{\boldsymbol{\Lambda}}_{k,h'}^{-1}}^2,1\}} + H\sqrt{2T\log(H/\delta)}.\end{aligned} \tag{39}$$

The inequalities above hold with probability $1-\delta$ by Cauchy-Schwarz inequality and Lemma C.3. Thus, we have

$$\begin{aligned}&\sum_{k=1}^{K}\sum_{h=1}^{H}\mathbb{P}_h\widehat{V}_{k,h+1}(s_h^k,a_h^k)\\&\leqslant \sum_{k=1}^{K}\sum_{h=1}^{H}\widehat{V}_{k,h+1}(s_{h+1}^k) + H\sqrt{2T\log(H/\delta)}\\&\leqslant 4H\widehat{\beta}^E\sqrt{\sum_{k=1}^{K}\sum_{h=1}^{H}\widehat{\sigma}_{k,h}^2}\sqrt{\sum_{k=1}^{K}\sum_{h=1}^{H}\min\{\|\boldsymbol{\phi}(s_h^k,a_h^k)\|_{\widehat{\boldsymbol{\Lambda}}_{k,h}^{-1}}^2,1\}} + (H^2+H)\sqrt{2T\log(H/\delta)}\end{aligned} \tag{40}$$

holds with probability $1-2\delta$. $\qquad\square$

**Lemma A.19.** *In Algorithm 1, under $\Psi_1^E$, we have*

$$\begin{aligned}\sum_{k=1}^{K}\sum_{h=1}^{H}\widehat{\sigma}_{k,h}^2 &\leqslant 2HT + 12H^3d^9\log(1 + 2d^7/(\lambda H)) + 2d^{3/2}H^2\sqrt{\lambda}\\&\quad + 2d^2(H^2+H)\sqrt{2T\log(H/\delta)} + 8d^5H(1+2H)^2(\widehat{\beta}^E)^2\log(1 + K/(Hd\lambda).\end{aligned}$$

*holds with probability $1-2\delta$.*

*Proof.* (Lemma A.19) From Algorithm 1, we can write the definition of $\widehat{\sigma}_{k,h}$ here,

$$\widehat{\sigma}_{k,h} := \max\left\{w_{k,h}, \sqrt{\frac{d^2}{H}W_{k,h}}\right\}.$$

Thus we can write

$$\sum_{k=1}^{K}\sum_{h=1}^{H}\widehat{\sigma}_{k,h}^2 \leqslant \underbrace{\sum_{k=1}^{K}\sum_{h=1}^{H}w_{k,h}^2}_{i} + \underbrace{\frac{d^2}{H}\sum_{k=1}^{K}\sum_{h=1}^{H}W_{k,h}}_{ii}.$$

To bound $i$, we utilize the conservatism of elliptical potentials, i.e., Lemma C.7. Initially, for fixed $h \in [H]$, set $\boldsymbol{x}_k$ as $\widetilde{\sigma}_{k,h}^{-1}\boldsymbol{\phi}(s_h^k, a_h^k)$ in Lemma C.7. Then for $C = 1/d^3$, during $K$ episodes, there are at most $3d\log\left(1 + d/(\lambda H\log(1+C^2))\right)/\log(1+C^2)$ episodes that $\left\|\widetilde{\sigma}_{k,h}^{-1}\boldsymbol{\phi}(s_h^k, a_h^k)\right\|_{\widetilde{\boldsymbol{\Lambda}}_{k,h}^{-1}} \geqslant 1/d^3$ for fixed $h \in [H]$. Thus, there are at most $3Hd\log\left(1 + d/(\lambda H\log(1 + C^2))\right)/\log(1+C^2)$ episodes that there exists $h' \in [H]$ such that $\left\|\widetilde{\sigma}_{k,h}^{-1}\boldsymbol{\phi}(s_{h'}^k, a_{h'}^k)\right\|_{\widetilde{\boldsymbol{\Lambda}}_{k,h'}^{-1}} > 1/d^3$. In particular, $w_{k,h}^2 = Hd^3$ during these episodes.

In summary, we obtain

$$\begin{aligned}
i = \sum_{k=1}^{K}\sum_{h=1}^{H}w_{k,h}^2 &\leqslant \sum_{k=1}^{K}\sum_{h=1}^{H}H + H\cdot Hd^3\frac{3Hd}{\log(1+1/d^6)}\log\left(1 + \frac{d}{\lambda H\log(1+1/d^6)}\right) \quad (41)\\
&\leqslant HT + 6H^3d^9\log(1 + 2d^7/(\lambda H)),
\end{aligned}$$

where the last inequality holds by $\log(1 + 1/x) \leqslant 2x$ for $x > 0$.

To bound $ii$, we can write

$$\begin{aligned}
ii = &\frac{d^2}{H}\sum_{k=1}^{K}\sum_{h=1}^{H}W_{k,h} = d^2\sum_{k=1}^{K}\sum_{h=1}^{H}\min\left\{\left\langle\widehat{\boldsymbol{\mu}}_{i,h}\widehat{\boldsymbol{V}}_{i,h+1}, \boldsymbol{\phi}(s_h^i, a_h^i)\right\rangle + \widehat{\beta}^E\left\|\boldsymbol{\phi}(s_h^i, a_h^i)\right\|_{\widehat{\boldsymbol{\Lambda}}_{i,h}^{-1}} + \zeta, H\right\}\\
&\leqslant d^2HK\zeta + d^2\sum_{k=1}^{K}\sum_{h=1}^{H}\min\{\widehat{\beta}^E\|\boldsymbol{\phi}(s_h^k, a_h^k)\|_{\widehat{\boldsymbol{\Lambda}}_{k,h}^{-1}}, H\}\\
&\quad + d^2\sum_{k=1}^{K}\sum_{h=1}^{H}\left\langle\boldsymbol{\mu}_h\widehat{\boldsymbol{V}}_{k,h+1}, \boldsymbol{\phi}(s_h^k, a_h^k)\right\rangle + \left\langle(\widehat{\boldsymbol{\mu}}_{k,h} - \boldsymbol{\mu}_h)\widehat{\boldsymbol{V}}_{k,h+1}, \boldsymbol{\phi}(s_h^k, a_h^k)\right\rangle\\
&\leqslant d^2HK\zeta + 2d^2\sum_{k=1}^{K}\sum_{h=1}^{H}\min\{\widehat{\beta}^E\|\boldsymbol{\phi}(s_h^k, a_h^k)\|_{\widehat{\boldsymbol{\Lambda}}_{k,h}^{-1}}, H\} + d^2\sum_{k=1}^{K}\sum_{h=1}^{H}\mathbb{P}_h\widehat{V}_{k,h+1}(s_h^k, a_h^k),
\end{aligned}$$
$$(42)$$

where $\zeta = H\sqrt{\lambda}/(2K\sqrt{d})$ and the last inequality holds under $\Psi_1^E$. Moreover, since $\widehat{\beta}^E\widehat{\sigma}_{k,h} \geqslant H\sqrt{d} \geqslant H$ holds for any $k \in [K]$ and $h \in [H]$, we can write

$$\begin{aligned}
2d^2\sum_{k=1}^{K}\sum_{h=1}^{H}\min\{\widehat{\beta}^E\|\boldsymbol{\phi}(s_h^k, a_h^k)\|_{\widehat{\boldsymbol{\Lambda}}_{k,h}^{-1}}, H\} &\leqslant 2d^2\widehat{\beta}^E\sum_{k=1}^{K}\sum_{h=1}^{H}\widehat{\sigma}_{k,h}\min\{\|\boldsymbol{\phi}(s_h^k, a_h^k)\|_{\widehat{\boldsymbol{\Lambda}}_{k,h}^{-1}}, 1\}\\
&\leqslant 2d^2\widehat{\beta}^E\sqrt{\sum_{k=1}^{K}\sum_{h=1}^{H}\widehat{\sigma}_{k,h}^2}\sqrt{\sum_{k=1}^{K}\sum_{h=1}^{H}\min\{\|\boldsymbol{\phi}(s_h^k, a_h^k)\|_{\widehat{\boldsymbol{\Lambda}}_{k,h}^{-1}}^2, 1\}},
\end{aligned}$$
$$(43)$$

By Lemma A.18, we have

$$\begin{aligned}
&d^2\sum_{k=1}^{K}\sum_{h=1}^{H}\mathbb{P}_h\widehat{V}_{k,h+1}(s_h^k, a_h^k)\\
&\leqslant 4d^2H\widehat{\beta}^E\sqrt{\sum_{k=1}^{K}\sum_{h=1}^{H}\widehat{\sigma}_{k,h}^2}\sqrt{\sum_{k=1}^{K}\sum_{h=1}^{H}\min\{\|\boldsymbol{\phi}(s_h^k, a_h^k)\|_{\widehat{\boldsymbol{\Lambda}}_{k,h}^{-1}}^2, 1\}} + d^2(H^2 + H)\sqrt{2T\log(H/\delta)}.
\end{aligned}$$
$$(44)$$

Thus, combine Eq. 41, 42, 43, 44, and Lemma C.6, we have

$$
\sum_{k=1}^{K} \sum_{h=1}^{H} \widehat{\sigma}_{k,h}^2 \leqslant HT + 6H^3 d^9 \log(1 + 2d^7/(\lambda H)) + d^{3/2} H^2 \sqrt{\lambda} + d^2(H^2 + H)\sqrt{2T\log(H/\delta)}
$$

$$
+ 2d^2(1 + 2H)\widehat{\beta}^E \sqrt{\sum_{k=1}^{K}\sum_{h=1}^{H} \widehat{\sigma}_{k,h}^2} \sqrt{2Hd\log(1 + K/(Hd\lambda))},
$$

Since for any $x > 0, a > 0, b > 0$, $x \leqslant a\sqrt{x} + b$ leads to $x \leqslant 2b + a^2$, we have

$$
\sum_{k=1}^{K} \sum_{h=1}^{H} \widehat{\sigma}_{k,h}^2 \leqslant 2HT + 12H^3 d^9 \log(1 + 2d^7/(\lambda H)) + 2d^{3/2} H^2 \sqrt{\lambda}
$$

$$
+ 2d^2(H^2 + H)\sqrt{2T\log(H/\delta)} + 8d^5 H(1 + 2H)^2(\widehat{\beta}^E)^2 \log(1 + K/(Hd\lambda)).
$$

which finishes the proof. $\qquad\square$

**Lemma A.20.** *In Algorithm 1, under $\Psi_1^E$, we have with probability at least $1 - 3\delta$,*

$$
\sum_{k=1}^{K} \widehat{V}_{k,1}(s_1^k) \leqslant \sum_{k=1}^{K} \mathbb{E}_{\tau_1^k \sim d_1^{\pi^k}(s_1^k)} \min\left\{ \sum_{h=1}^{H} 4\widehat{\beta}^E \left\| \phi(s_h^k, a_h^k) \right\|_{\widehat{\mathbf{\Lambda}}_{k,h}^{-1}}, H \right\}
$$
$$
\leqslant \widetilde{O}\left( \sqrt{H^4 d^3 K} \right)
$$
(45)

*Proof.* (Lemma A.20)

On the one hand, be Lemma A.17, we have

$$
\sum_{k=1}^{K} \widehat{V}_{k,1}(s_1^k) \leqslant \sum_{k=1}^{K} \mathbb{E}_{\tau_1^k \sim d_1^{\pi^k}(s_1^k)} \min\left\{ \sum_{h=1}^{H} 4\widehat{\beta}^E \left\| \phi(s_h^k, a_h^k) \right\|_{\widehat{\mathbf{\Lambda}}_{k,h}^{-1}}, H \right\},
$$
(46)

under $\Psi_1^E$.

On the other hand, we have

$$
\sum_{k=1}^{K} \min\left\{ \sum_{h=1}^{H} 4\widehat{\beta}^E \left\| \phi(s_h^k, a_h^k) \right\|_{\widehat{\mathbf{\Lambda}}_{k,h}^{-1}}, H \right\}
$$
$$
= \sum_{k=1}^{K} \min\left\{ \sum_{h=1}^{H} 4\widehat{\beta}^E \widehat{\sigma}_{k,h} \left\| \widehat{\sigma}_{k,h}^{-1} \phi(s_h^k, a_h^k) \right\|_{\widehat{\mathbf{\Lambda}}_{k,h}^{-1}}, H \right\}
$$
$$
\leqslant 4 \underbrace{\sum_{k=1}^{K} \sum_{h=1}^{H} \widehat{\beta}^E \widehat{\sigma}_{k,h} \min\left\{ \left\| \widehat{\sigma}_{k,h}^{-1} \phi(s_h^k, a_h^k) \right\|_{\widehat{\mathbf{\Lambda}}_{k,h}^{-1}}, 1 \right\}}_{I},
$$
(47)

where the inequality holds since $\widehat{\beta}^E \widehat{\sigma}_{k,h} \geqslant \sqrt{H}d \cdot \sqrt{H} \geqslant H$. To further bound $I$, we have

$$
I \leqslant \widehat{\beta}^E \sqrt{\sum_{k=1}^{K}\sum_{h=1}^{H} \widehat{\sigma}_{k,h}^2} \sqrt{\sum_{k=1}^{K}\sum_{h=1}^{H} \min\left\{ \left\| \widehat{\sigma}_{k,h}^{-1} \phi(s, a) \right\|_{\widehat{\mathbf{\Lambda}}_{k,h}^{-1}}^2, 1 \right\}}
$$
$$
\leqslant \widehat{\beta}^E \sqrt{\sum_{k=1}^{K}\sum_{h=1}^{H} \widehat{\sigma}_{k,h}^2} \sqrt{2Hd\log(1 + K/(\lambda Hd))},
$$
(48)

where the first inequality holds due to Cauchy-Schwarz inequality and the second inequality holds due to Lemma C.6 with the fact that $\left\| \widehat{\sigma}_{k,h}^{-1} \phi\left(s_h^k, a_h^k\right) \right\|_2 \leqslant 1/\sqrt{H}$.

By setting $\lambda = 1/(H\sqrt{d})$, we obtain $\widehat{\beta}^E = \widetilde{O}(d\sqrt{H}\log(T))$ by Lemma A.6, where $T = KH$. By Lemma A.19, we have with probability $1 - 2\delta$

$$
\begin{aligned}
\sum_{k=1}^{K}\sum_{h=1}^{H} \widehat{\sigma}_{k,h}^2 \leqslant{}& 2HT + 12H^3 d^9 \log(1 + 2d^7/(\lambda H)) + 2d^{3/2}H^2\sqrt{\lambda} \\
&+ 2d^2(H^2 + H)\sqrt{2T\log(H/\delta)} + 8d^5 H(1 + 2H)^2(\widehat{\beta}^E)^2 \log(1 + K/(Hd\lambda)) \\
={}& \widetilde{O}(HT).
\end{aligned}
\tag{49}
$$

Substituting Eq. (49) in (48), and combing Eq. (47), we obtain

$$
\begin{aligned}
&\sum_{k=1}^{K} \min\left\{ \sum_{h=1}^{H} 4\widehat{\beta}^E \left\| \phi(s_h^k, a_h^k) \right\|_{\widehat{\mathbf{\Lambda}}_{k,h}^{-1}}, H \right\} \\
&\leqslant 4\widehat{\beta}^E \sqrt{\sum_{k=1}^{K}\sum_{h=1}^{H} \widehat{\sigma}_{k,h}^2} \sqrt{2Hd\log(1 + K/(\lambda Hd))} \\
&= \widetilde{O}\left( d\sqrt{H}\log(T/\delta) \cdot \sqrt{Hd\log T} \cdot \sqrt{HT} \right) \leqslant \widetilde{O}\left( \sqrt{H^3 d^3 T \log^3(T/\delta)} \right),
\end{aligned}
\tag{50}
$$

holds with probability at least $1 - 2\delta$. Finally, by Hoeffding inequality(Lemma C.3), we have with probability at least $1 - 3\delta$,

$$
\begin{aligned}
\sum_{k=1}^{K} \widehat{V}_{k,1}(s_1^k) &\leqslant \sum_{k=1}^{K} \mathbb{E}_{\tau_1^k \sim d_1^{\pi^k}(s_1^k)} \min\left\{ \sum_{h=1}^{H} 4\widehat{\beta} \left\| \phi(s_h^k, a_h^k) \right\|_{\widehat{\mathbf{\Lambda}}_{k,h}^{-1}}, H \right\} \\
&\leqslant \widetilde{O}\left( \sqrt{H^3 d^3 T \log^3(T/\delta)} \right) + H\sqrt{2T\log(H/\delta)} \\
&= \widetilde{O}\left( \sqrt{H^4 d^3 K} \right).
\end{aligned}
$$

That finishes the proof. $\qquad\square$

We denote the event that the inequality in Lemma A.20 holds as $\Phi$, which holds with probability at least $1 - 3\delta$.

## A.5 Proof of Theorem 4.2

In this subsection, we bound the sub-optimality gap of the recovered policy in the planning phase under event $\Psi_1^E \cap \Psi_1^P \cap \Xi \cap \Phi$, which also ends the proof of Theorem 4.2.

**Lemma A.21.** *For any $0 < \delta < 1$, with probability at least $1 - \delta$, we have*

$$
\left| \sum_{k=1}^{K} \left( \mathbb{E}_{s\sim\mu}\left[ \widetilde{V}_1^*(s; r_k) \right] - \widetilde{V}_1^*\left( s_1^k; r_k \right) \right) \right| \leqslant H\sqrt{2KH\log(1/\delta)},
$$

*where $r_k = \{r_{k,h}\}_{h\in[H]}$.*

*Proof.* (Lemma A.21) Denote $\Delta_k = \mathbb{E}_{s\sim\mu}\left[ \widetilde{V}_1^*(s; r_k) \right] - \widetilde{V}_1^*\left( s_1^k; r_k \right)$. Since $s_1^k$ is $\mathcal{F}_{k,1}$-measurable, $\Delta_k$ is $\mathcal{F}_{k,1}$-measurable and $\mathbb{E}\left[ \Delta_k \mid \mathcal{F}_{k-1,H} \right] = 0$, i.e., $\Delta_k$ is a martingale difference sequence. Since $|\Delta_k| \leqslant H$ by $0 \leqslant \widetilde{V}^*(s; r_k) \leqslant H$ for all $s \in \mathcal{S}$, we can apply Azuma-Hoeffding inequality (Lemma C.3) to this martingale difference sequence and obtain

$$
\left| \sum_{k=1}^{K} \Delta_k \right| \leqslant H\sqrt{2KH\log(1/\delta)},
\tag{51}
$$

with probability at least $1 - \delta$. $\qquad\square$

Now we denote the event that the conclusion of Lemma A.21 holds as $\Xi$, which is a high probability event.

**Lemma A.22.** *Under $\Psi_1^E \cap \Xi \cap \Phi$, we have*

$$\mathbb{E}_{s\sim\mu}\left[\widetilde{V}_1^*\left(s;b\right)\right] \leqslant \widetilde{O}\left(\sqrt{H^4 d^2/K}\right),$$

*where $b = \{b_h\}_{h\in[H]}$ is the UCB bonus defined in Algorithm 2.*

*Proof.* (Lemma A.22)

Notice that $\widehat{\mathbf{\Lambda}}_{K+1,h} \geqslant \widehat{\mathbf{\Lambda}}_{k,h}$, $\widehat{\beta}^E = \widetilde{O}(d\sqrt{H})$, and $\widehat{\beta}^P = \widetilde{O}(\sqrt{Hd})$, for any $k \in [K]$ and any $h \in [H]$. We have

$$\widehat{\beta}^E \|\phi(\cdot,\cdot)\|_{\widehat{\mathbf{\Lambda}}_{k,h}^{-1}} \geqslant c\sqrt{d}\widehat{\beta}^P \|\phi(\cdot,\cdot)\|_{\widehat{\mathbf{\Lambda}}_{K+1,h}^{-1}},$$

where $c > 0$ is a constant, which further implies $r_{k,h}(\cdot,\cdot) \geqslant c\sqrt{d}b_h(\cdot,\cdot)$ for in LSVI-RFE. Subsequently, we have

$$\begin{aligned}
&c\sqrt{d}\mathbb{E}_{s\sim\mu}\left[\widetilde{V}_1^*(s;b)\right] \\
&= \mathbb{E}_{s\sim\mu}\left[\widetilde{V}_1^*(s;c\sqrt{d}\cdot b)\right] \\
&\leqslant \sum_{k=1}^{K}\mathbb{E}_{s\sim\mu}\left[\widetilde{V}_1^*(s;r_k)\right]/K \\
&= \left\{\sum_{k=1}^{K}\widetilde{V}_1^*(s_1^k;r_k) + \sum_{k=1}^{K}\left(\mathbb{E}_{s\sim\mu}\left[\widetilde{V}_1^*(s;r_k)\right] - \widetilde{V}_1^*(s_1^k;r_k)\right)\right\}/K \\
&\leqslant \left(\sum_{k=1}^{K}\widetilde{V}_1^*(s_1^k;r_k)\right)/K + H\sqrt{2H\log(1/\delta)/K} \\
&\leqslant \left(\sum_{k=1}^{K}\widehat{V}_{k,1}(s_1^k)\right)/K + H\sqrt{2H\log(1/\delta)/K} \\
&\leqslant \widetilde{O}\left(\sqrt{H^4 d^3/K}\right),
\end{aligned}$$

where the second inequality holds by Lemma A.21 under $\Xi$, and third inequality holds by Lemma A.7 under $\Psi_1^E$, and the last inequality holds by Lemma A.20 under $\Psi_1^E \cap \Phi$. Thus

$$\mathbb{E}_{s\sim\mu}\left[V_1^*\left(s;b\right)\right] \leqslant \widetilde{O}\left(\sqrt{H^4 d^2/K}\right).$$

$\square$

Now we are ready to prove the main theorem.

*Proof of Theorem 4.2.* It suffices to prove the conclusion under the event $\Psi_1^E \cap \Psi_1^P \cap \Xi \cap \Phi$ which holds at probability at least $1 - 7\delta$ by Lemma A.13 and Lemma A.21 and taking a union bound.

Initially, we have

$$\begin{aligned}
&\mathbb{E}_{s_1\sim\mu}\left[V_1^*\left(s_1;r\right) - V_1^\pi\left(s_1;r\right)\right] \\
&\leqslant \mathbb{E}_{s_1\sim\mu}\left[\widehat{V}_1(s_1) - V_1^\pi(s_1;r)\right] \\
&= \mathbb{E}_{s_1\sim\mu}\Big[\min\left\{r_1(s_1,\pi(s_1)) + \langle\widehat{\boldsymbol{\mu}}_{K+1,1}\widehat{\boldsymbol{V}}_2, \boldsymbol{\phi}(s_1,\pi(s_1))\rangle + b_1(s_1,\pi(s_1)), H\right\} \\
&\quad - r_1(s_1,\pi(s_1)) - \mathbb{P}_1 V_2^\pi(s_1,\pi(s_1);r)\Big] \\
&= \mathbb{E}_{s_1\sim\mu}\Big[\min\left\{H, \langle\widehat{\boldsymbol{\mu}}_{K+1,1}\widehat{\boldsymbol{V}}_2, \boldsymbol{\phi}(s_1,\pi(s_1))\rangle + b_1(s_1,\pi(s_1)) - \mathbb{P}_1 V_2^\pi(s_1,\pi(s_1);r)\right\}\Big]
\end{aligned}$$

$$
\begin{aligned}
&= \mathbb{E}_{s_1 \sim \mu}\Big[\min\Big\{H, \langle \widehat{\boldsymbol{\mu}}_{K+1,1}\widehat{\boldsymbol{V}}_2, \boldsymbol{\phi}(s_1, \pi(s_1))\rangle - \langle \boldsymbol{\mu}_1\widehat{\boldsymbol{V}}_2, \boldsymbol{\phi}(s_1, \pi(s_1))\rangle + b_1(s_1, \pi(s_1)) \\
&\qquad\qquad + \mathbb{P}_1\widehat{V}_2(s_1, \pi(s_1); r) - \mathbb{P}_1 V_2^\pi(s_1, \pi(s_1); r)\Big\}\Big] \\
&\leqslant \mathbb{E}_{s_1 \sim \mu}\Big[\min\Big\{H, \big\|(\widehat{\boldsymbol{\mu}}_{K+1,1} - \boldsymbol{\mu}_1)\widehat{\boldsymbol{V}}_2\big\|_{\widehat{\boldsymbol{\Lambda}}_{K+1,1}} \|\boldsymbol{\phi}(s_1, \pi(s_1))\|_{\widehat{\boldsymbol{\Lambda}}_{K+1,1}^{-1}} + b_1(s_1, \pi(s_1)) \\
&\qquad\qquad + \mathbb{P}_1\widehat{V}_2(s_1, \pi(s_1); r) - \mathbb{P}_1 V_2^\pi(s_1, \pi(s_1); r)\Big\}\Big] \\
&\leqslant \mathbb{E}_{s_1 \sim \mu}\Big[\min\Big\{H, 2b_1(s_1, \pi(s_1)) + \mathbb{P}_1\widehat{V}_2(s_1, \pi(s_1); r) - \mathbb{P}_1 V_2^\pi(s_1, \pi(s_1); r)\Big\}\Big] \\
&= \mathbb{E}_{s_1 \sim \mu, s_2 \sim \mathbb{P}(\cdot|s_1, \pi(s_1))}\Big[\min\Big\{H, 2b_1(s_1, \pi(s_1)) + \widehat{V}_2(s_2) - V_2^\pi(s_2; r)\Big\}\Big] \\
&\leqslant \mathbb{E}_{\tau \sim d^\pi}\left[\min\left\{\sum_{h=1}^H 2b_h(s_h, \pi(s_h)), H\right\}\right] \\
&\leqslant 2\mathbb{E}_{\tau \sim d^\pi}\left[\min\left\{\sum_{h=1}^H b_h(s_h, \pi(s_h)), H\right\}\right] \\
&\leqslant 2\mathbb{E}_{s_1 \sim \mu}\left[\widetilde{V}_1^\pi(s_1; b)\right],
\end{aligned}
$$

where the first inequality holds by Lemma A.14, the second inequality holds by Cauchy-Schwarz inequality, the third inequality holds since $\boldsymbol{\mu}_h \in \widehat{C}_1^P$ under $\Psi_1^P$, the fourth inequality holds by recursively decomposition, and the last inequality holds by definition of $\widetilde{V}_1^\pi(\cdot; b)$. In addition, we have

$$
\mathbb{E}_{s \sim \mu}\left[\widetilde{V}_1^\pi(s; b)\right] \leqslant \mathbb{E}_{s \sim \mu}\left[\widetilde{V}_1^*(s; b)\right] \leqslant \widetilde{O}\left(\sqrt{H^4 d^2/K}\right),
$$

where the first inequality holds by definition of $\widetilde{V}_1^*(s; b)$, and the second inequality holds by Lemma A.22 under $\Psi_1^E \cap \Xi \cap \Phi$.

If we ignore logarithmic terms and take $K = m(H^4 d^2/\epsilon^2)$ for a sufficiently large constant $m > 0$, we have

$$
\mathbb{E}_{s_1 \sim \mu}[V_1^*(s_1; r) - V_1^\pi(s_1; r)] \leqslant \widetilde{O}\left(\sqrt{H^4 d^2/K}\right) \leqslant \epsilon
$$

Thus, we need $K = \widetilde{O}(H^4 d^2/\epsilon^2)$ episodes to output an $\epsilon$-optimal policy $\pi$, when $\epsilon$ is small enough. $\qquad\square$

## B  PROOF OF LOWER BOUND (THEOREM 6.1)

In this section, we provide proof for Theorem 6.1 in the manuscript. Notice that if the reward function is given in the exploration phase, the RFE setting degrades to Probably Approximately Correct (PAC) RL setting Dann et al. (2019). Thus, it suffices to prove a lower bound for PAC RL since an algorithm for RFE also works for PAC RL by neglecting the reward function in the exploration phase.

Our proof is inspired by the proof of Theorem 3 in Appendix C of Chen et al. (2021), which provides a lower bound for RFE in linear mixture MDPs. In particular, we connect the lower bound of RFE with regret minimization in linear MDPs.

Firstly, we construct a hard-to-learn MDP $\mathcal{M} = \{\mathcal{S}, \mathcal{A}, H, \{\mathbb{P}_h\}_h, \{r_h\}_h, \nu\}^3$ in Lemma B.1 such that any algorithm which runs $K$ episodes will obtain the regret at least $\Omega(dH\sqrt{HK})$ as shows in Lemma B.1.

**Lemma B.1** (Remark 23 in Zhou et al. (2021)). *Let $d > 1$ and suppose $d \geqslant 4, H \geqslant 3$ and $K \geqslant \max\left\{(d-1)^2 H/2, (d-1)/(32H(d-1))\right\}$. Then, there exists an episodic linear MDP $\mathcal{M} = \{\mathcal{S}, \mathcal{A}, H, \{\mathbb{P}_h\}_h, \{r_h\}_h, \nu\}$ parameterized by $\{\boldsymbol{\mu}_h\}_{h \in [H]}, \{\boldsymbol{\theta}_h\}_{h \in [H]}$ and satisfy Definition 3.1, such that for any algorithm, the expected regret is lower bounded as follows:*

$$
\mathbb{E}_{s_1^k \sim \nu}\left[\sum_{k=1}^K V^*(s_1^k) - V^{\pi^k}(s_1^k)\right] \geqslant \Omega(dH\sqrt{T}),
$$

---

[3] $\nu$ denotes the initial state distribution.

*where $T = KH$ and the expectation is taken over the probability distribution generated by the interconnection of the algorithm and the MDP.*

**Hard MDP Instance** $\mathcal{M}$ In this MDP, $\mathcal{S} = \{x_1, x_2, \cdots, x_{H+2}\}$, $\mathcal{A} = \{-1, 1\}^{d-1}$, and the linear parameterized of $\mathcal{M}$ are specified as

$$\phi(s, \boldsymbol{a}) = \begin{cases} (\alpha, \beta \boldsymbol{a}^\top, 0)^\top, & s = x_h, h \in [H+1] \\ (0, \boldsymbol{0}^\top, 1)^\top, & s = x_{H+2} \end{cases}$$

$$\boldsymbol{\mu}_h(s') = \begin{cases} ((1-\iota)/\alpha, -\boldsymbol{\mu}_h^\top/\beta, 0)^\top, & s' = x_{H+1}, \\ (\iota/\alpha, \boldsymbol{\mu}_h^\top/\beta, 1)^\top, & s' = x_{H+2} \\ \boldsymbol{0}, & \text{otherwise} \end{cases},$$

$$\boldsymbol{\theta}_h = (\boldsymbol{0}^\top, 1)^\top$$

where $\iota = 1/H$, $\Delta = \sqrt{\iota/K}/(4\sqrt{2})$, $\alpha = \sqrt{1/(1 + \Delta(d-1))}$, and $\beta = \sqrt{\Delta/(1 + \Delta(d-1))}$. Under this parameterization, we have

$$\mathbb{P}_h(s'|s_i, \boldsymbol{a}) = \begin{cases} \iota + \langle \boldsymbol{\mu}_h, \boldsymbol{a} \rangle, & s' = x_{H+2} \\ 1 - (\iota + \langle \boldsymbol{\mu}_h, \boldsymbol{a} \rangle), & s' = x_{i+1} \\ 0, & \text{Otherwise} \end{cases},$$

and only the transition starting at $s_{H+2}$ generates a reward.

Then, for any algorithm $\mathcal{ALG}_1$ running $K_1$ episodes to learn an $\epsilon$-optimal policy $\pi(K_1)$ with probability at least $1 - \delta$, it suffices to prove that *under the instance $\mathcal{M}$, $K_1 \geqslant Cd^2H^3/\epsilon^2$.*

We prove this by constructing a new algorithm $\mathcal{ALG}_2$. $\mathcal{ALG}_2$ firstly runs $\mathcal{ALG}_1$ for $K_1$ episodes, then $\mathcal{ALG}_1$ outputs a $\epsilon$-optimal policy $\pi(K_1)$ with probability at least $1 - \delta$. Then $\mathcal{ALG}_2$ executes $\pi(K_1)$ in the following $(c-1)K_1$ episodes, which means $\mathcal{ALG}_2$ runs $K_2 = cK_1$ episodes.

Note that Lemma B.2 also gives an upper bound for the regret of every single step under $\mathcal{M}$.

**Lemma B.2.** *Suppose $3(d-1)\Delta \leqslant \iota$. Then, for all $k \in [K_2]$, we have*

$$\mathbb{E}_{x \sim \nu} \left[ V_1^*(x_1) - V_1^{\pi^k}(x_1) \right] \leqslant \frac{dH}{4\sqrt{2}} \sqrt{\frac{H}{K_2}}.$$

*Proof.* (Lemma B.2) We prove this lemma by computing $V_1^*(x_1)$, following the standard analysis of this hard-to-learn MDP in Zhou et al. (2021).

By definition of $\mathcal{M}$, we have $(d-1)\Delta = \max_{a \in \mathcal{A}} \langle \boldsymbol{\mu}_h, a \rangle$ such that

$$V_1^\pi(x_1) = \mathbb{E}\left[ \sum_{h=1}^{H} r_h(s_h, a_h) \mid s_1 = x_1, a_h = \pi_h(s_h) \right].$$

Since only $r_h(x_{H+2}, \cdot) = 1$, we have $V_1^\pi(x_1) = \sum_{h=1}^{H-1}(H-h)\mathbb{P}(N_h)$, where $N_h = \{s_{h+1} = x_{H+2}, s_h = x_h\}$. We also have

$$\mathbb{P}(s_{h+1} = x_{H+2} \mid s_h = x_h, s_1 = x_1)$$
$$= \sum_{a \in \mathcal{A}} \mathbb{P}(s_{h+1} = x_{H+2} \mid s_h = x_h, a_h = a)\mathbb{P}(a_h = a \mid s_h = x_h, s_1 = x_1)$$
$$= \iota + \langle \boldsymbol{\mu}_h, \overline{a}_h^\pi \rangle,$$

where $\overline{a}_h^\pi = \sum_{a \in \mathcal{A}} \mathbb{P}(a_h = a \mid s_h = x_h, s_1 = x_1)a$. Thus

$$\mathbb{P}(N_h) = (\iota + \langle \boldsymbol{\mu}_h, \overline{a}_h^\pi \rangle) \prod_{j=1}^{h-1} (1 - \iota - \langle \boldsymbol{\mu}_j, \overline{a}_j^\pi \rangle).$$

Subsequently, we get

$$V_1^\pi(x_1) = \sum_{h=1}^{H}(H-h)(a_h + \iota)\prod_{j=1}^{h-1}(1 - a_j - \iota). \tag{52}$$

By definition of $V_1^*$, we choose the optimal policy at each stage. Since $\max_{a \in \mathcal{A}} \langle \boldsymbol{\mu}_h, a \rangle = (d-1)\Delta$, we get

$$V_1^*(x_1) = \sum_{h=1}^{H} (H-h)((d-1)\Delta + \iota)(1 - (d-1)\Delta - \iota)^{h-1} \leqslant H \cdot H \cdot (d\Delta) \qquad (53)$$

where the inequality holds by $0 < 1 - (d-1)\Delta - \iota < 1$ and $\Delta > \iota$. Since the inequality holds by all fixed $x_1$, we also have

$$\mathbb{E}_{x_1 \sim \mu} \left[ V_1^*(x_1) - V_1^{\pi^k}(x_1) \right] \leqslant \mathbb{E}_{x_1 \sim \mu} \left[ V_1^*(x_1) \right] \leqslant dH^2 \Delta \leqslant \frac{dH}{4\sqrt{2}} \sqrt{\frac{H}{K_2}},$$

where the last inequality holds by $\iota = 1/H$. $\qquad \square$

On the one hand, by Lemma B.1, we have

$$\sum_{k=1}^{K_2} \mathbb{E}_{x_1 \sim \nu} \left[ V_1^*(x_1) - V_1^{\pi^k}(x_1) \right] \geqslant c' dH \sqrt{HK_2},$$

where $\pi^k$ denotes the policy for $\mathcal{ALG}_2$ in episode $k$ and $c'$ is a constant number.

On the other hand, by Lemma B.2, we have

$$\sum_{k=1}^{K_1} \mathbb{E}_{x_1 \sim \nu} \left[ V_1^*(x_1) - V_1^{\pi^k}(x_1) \right] \leqslant K_1 \frac{dH}{4\sqrt{2}} \sqrt{\frac{H}{K_2}} = \frac{dH}{4\sqrt{2}} \sqrt{\frac{HK_1}{c}} = \frac{dH}{4\sqrt{2c}} \sqrt{HK_2}.$$

By choosing $c = \max\{1/(2\sqrt{2}c'), 2\}$, we know that for $\mathcal{ALG}_2$ under $\mathcal{M}$,

$$\sum_{k=K_1+1}^{K_2} \mathbb{E}_{x_1 \sim \nu} \left[ V_1^*(x_1) - V_1^{\pi^k}(x_1) \right] \geqslant (c' - \frac{1}{4\sqrt{2}c}) dH \sqrt{HK_2} \geqslant \frac{c'}{2} dH \sqrt{HK_2} = \frac{c'}{2} dH \sqrt{cHK_1}.$$

Because $\pi^k = \pi(K_1)$ in $\mathcal{ALG}_2$ for $K_1 + 1 \leqslant k \leqslant K_2$ and $K_2 - (K_1 + 1) + 1 = (c-1)K_1$, we have

$$\mathbb{E}_{x_1 \sim \nu} \left[ V_1^*(x_1) - V_1^{\pi(K_1)}(s_1) \right] \geqslant \frac{c'\sqrt{c}}{2(c-1)} dH \sqrt{H/K_1}. \qquad (54)$$

Since $\mathcal{ALG}_1$ outputs an $\epsilon$-optimal policy $\pi(K_1)$ with probability at least $1 - \delta$, we also have

$$\mathbb{E}_{x_1 \sim \nu} \left[ V_1^*(x_1) - V_1^{\pi(K_1)}(s_1) \right] \leqslant (1 - \delta) \cdot \epsilon + \delta \cdot \frac{dH}{4\sqrt{2}} \sqrt{\frac{H}{K_2}} \leqslant (1 - \delta) \cdot \epsilon + \delta \cdot \frac{dH}{4\sqrt{2}} \sqrt{\frac{H}{cK_1}}, \tag{55}$$

where the inequality holds by utilizing Lemma B.2, and the second inequality holds by $K_2 = cK_1$. Combining Eq. (54) and Eq. (55) gives

$$(1 - \delta)\varepsilon + \delta \frac{dH}{4\sqrt{2c}} \sqrt{\frac{H}{K_1}} \geqslant \frac{c'\sqrt{c}}{2(c-1)} dH \sqrt{H/K_1}.$$

By setting $\delta$ satisfying $0 < \delta < \min\{1/H, 2\sqrt{2}cc'/(c-1)\}$, we obtains $K_1 \geqslant Cd^2 H^3/\varepsilon^2$ for some positive constant $C$, which completes our proof.

## C   AUXILIARY LEMMAS

### C.1   CONCENTRATION INEQUALITY

This subsection gives the encountered concentration inequalities in our proof.

**Lemma C.1** (Hoeffding inequality for vector-valued martingales, Theorem 1 in Abbasi-Yadkori et al. (2011)). *Let $\{\mathcal{G}_t\}_{t=1}^{\infty}$ be a filtration, $\{\boldsymbol{x}_t, \eta_t\}_{t \geqslant 1}$ be a stochastic process so that $\boldsymbol{x}_t \in \mathbb{R}^d$ is $\mathcal{G}_t$-measurable and $\eta_t \in \mathbb{R}$ is $\mathcal{G}_{t+1}$-measurable.*

*Denote $\mathbf{Z}_t = \lambda \mathbf{I} + \sum_{i=1}^{t} \boldsymbol{x}_i \boldsymbol{x}_i^{\top}$ for $t \geqslant 1$ and $\mathbf{Z}_0 = \lambda \mathbf{I}$. If $\|\boldsymbol{x}_t\|_2 \leqslant L$, and $\eta_t$ satisfies*

$$\mathbb{E}\left[\eta_t \mid \mathcal{G}_t\right] = 0, \quad |\eta_t| \leqslant R$$

*for all $t \geqslant 1$. Then, for any $0 < \delta < 1$, with probability at least $1 - \delta$ we have:*

$$\forall t > 0, \left\|\sum_{i=1}^{t} \boldsymbol{x}_i \eta_i\right\|_{\mathbf{Z}_t^{-1}} \leqslant R\sqrt{d \log\left(1 + tL^2/(d\lambda)\right) + \log(1/\delta)}.$$

**Lemma C.2** (Bernstein inequality for vector-valued martingales, Theorem 7.1 in Hu et al. (2022)). *Let $\{\mathcal{G}_t\}_{t=1}^{\infty}$ be a filtration, $\{\boldsymbol{x}_t, \eta_t\}_{t \geqslant 1}$ be a stochastic process so that $\boldsymbol{x}_t \in \mathbb{R}^d$ is $\mathcal{G}_t$-measurable and $\eta_t \in \mathbb{R}$ is $\mathcal{G}_{t+1}$-measurable.*

*If $\|\boldsymbol{x}_t\|_2 \leqslant L$, and $\eta_t$ satisfies*

$$\mathbb{E}\left[\eta_t \mid \mathcal{G}_t\right] = 0, \quad \mathbb{E}\left[\eta_t^2 \mid \mathcal{G}_t\right] \leqslant \sigma^2, \quad \left|\eta_t \cdot \min\left\{1, \|\boldsymbol{x}_t\|_{\mathbf{Z}_{t-1}^{-1}}\right\}\right| \leqslant R$$

*for all $t \geqslant 1$. Then, for any $0 < \delta < 1$, with probability at least $1 - \delta$ we have:*

$$\forall t > 0, \left\|\sum_{i=1}^{t} \boldsymbol{x}_i \eta_i\right\|_{\mathbf{Z}_t^{-1}} \leqslant 8\sigma\sqrt{d \log\left(1 + tL^2/(d\lambda)\right)\log\left(4t^2/\delta\right)} + 4R\log\left(4t^2/\delta\right)$$

*where $\mathbf{Z}_t = \lambda \mathbf{I} + \sum_{i=1}^{t} \boldsymbol{x}_i \boldsymbol{x}_i^{\top}$ for $t \geqslant 1$ and $\mathbf{Z}_0 = \lambda \mathbf{I}$.*

**Lemma C.3** (Azuma-Hoeffding Inequality). *Let $\{x_i\}_{i=1}^{n}$ be a martingale difference sequence with respect to a filtration $\{\mathcal{G}_i\}_{i=1}^{n+1}$ such that $|x_i| \leqslant M$ almost surely. That is, $x_i$ is $\mathcal{G}_{i+1}$-measurable and $\mathbb{E}\left[x_i \mid \mathcal{G}_i\right] = 0$ a.s. Then for any $0 < \delta < 1$, with probability at least $1 - \delta$,*

$$\sum_{i=1}^{n} x_i \leqslant M\sqrt{2n\log(1/\delta)}$$

## C.2 LINEAR MDP PROPERTY

This subsection gives some indirect results about the estimated parameter $\widehat{\boldsymbol{\mu}}_{k,h}$ in the exploration phase.

**Lemma C.4.** *In Algorithm 1, for any $k \in [K+1]$ and any $h \in [H]$, we have:*

$$\widehat{\boldsymbol{\mu}}_{k,h} - \boldsymbol{\mu}_h = \widehat{\boldsymbol{\Lambda}}_{k,h}^{-1}\left[-\lambda\boldsymbol{\mu}_h + \sum_{i=1}^{k-1} \widehat{\sigma}_{i,h}^{-2}\boldsymbol{\phi}\left(s_h^i, a_h^i\right)\boldsymbol{\epsilon}_h^{i\top}\right] \tag{56}$$

*Proof.* (Lemma C.4) We start from the closed-form solution of $\widehat{\boldsymbol{\mu}}_{k,h}$ :

$$
\begin{aligned}
\widehat{\boldsymbol{\mu}}_{k,h} &= \widehat{\boldsymbol{\Lambda}}_{k,h}^{-1}\sum_{i=1}^{k-1} \widehat{\sigma}_{i,h}^{-2}\boldsymbol{\phi}\left(s_h^i, a_h^i\right)\boldsymbol{\delta}\left(s_{h+1}^i\right)^{\top} \\
&= \widehat{\boldsymbol{\Lambda}}_{k,h}^{-1}\sum_{i=1}^{k-1} \widehat{\sigma}_{i,h}^{-2}\boldsymbol{\phi}\left(s_h^i, a_h^i\right)\left(\mathbb{P}_h(\cdot \mid s_h^i, a_h^i)^{\top} + \boldsymbol{\epsilon}_h^{i\top}\right) \\
&= \widehat{\boldsymbol{\Lambda}}_{k,h}^{-1}\sum_{i=1}^{k-1} \widehat{\sigma}_{i,h}^{-2}\boldsymbol{\phi}\left(s_h^i, a_h^i\right)\left(\boldsymbol{\phi}(s_h^i, a_h^i)^{\top}\boldsymbol{\mu}_h + \boldsymbol{\epsilon}_h^{i\top}\right) \\
&= \widehat{\boldsymbol{\Lambda}}_{k,h}^{-1}\sum_{i=1}^{k-1} \widehat{\sigma}_{i,h}^{-2}\boldsymbol{\phi}\left(s_h^i, a_h^i\right)\boldsymbol{\phi}(s_h^i, a_h^i)^{\top}\boldsymbol{\mu}_h + \widehat{\boldsymbol{\Lambda}}_{k,h}^{-1}\sum_{i=1}^{k-1} \widehat{\sigma}_{i,h}^{-2}\boldsymbol{\phi}\left(s_h^i, a_h^i\right)\boldsymbol{\epsilon}_h^{i\top}
\end{aligned}
$$

$$= \widehat{\boldsymbol{\Lambda}}_{k,h}^{-1} \left( \widehat{\boldsymbol{\Lambda}}_{k,h} - \lambda \mathbf{I} \right) \boldsymbol{\mu}_h + \widehat{\boldsymbol{\Lambda}}_{k,h}^{-1} \sum_{i=1}^{k-1} \widehat{\sigma}_{i,h}^{-2} \boldsymbol{\phi} \left( s_h^i, a_h^i \right) \boldsymbol{\epsilon}_h^i {}^{\top}$$

$$= \boldsymbol{\mu}_h - \lambda \widehat{\boldsymbol{\Lambda}}_{k,h}^{-1} \boldsymbol{\mu}_h + \widehat{\boldsymbol{\Lambda}}_{k,h}^{-1} \sum_{i=1}^{k-1} \widehat{\sigma}_{i,h}^{-2} \boldsymbol{\phi} \left( s_h^i, a_h^i \right) \boldsymbol{\epsilon}_h^i {}^{\top}$$

Rearranging the terms gives Eq. (56). □

### C.3 COVERING NET

**Lemma C.5** (Lemma D.6. in Jin et al. (2020b)). *Let $\widehat{\mathcal{N}}_\varepsilon$ be the $\varepsilon$-covering of $\widehat{\mathcal{V}}(L, B)$ with respect to the distance $\mathrm{dist}\,(V, V') = \sup_x |V(x) - V'(x)|$, where $\widehat{\mathcal{V}}(L, B)$ is defined in Definition A.4. Then*

$$\log |\widehat{\mathcal{N}}_\varepsilon| \leqslant d \log(1 + 4L/\varepsilon) + d^2 \log \left[ 1 + 8d^{1/2} B^2 / \left( \lambda \varepsilon^2 \right) \right].$$

### C.4 ELLIPTICAL POTENTIALS

In this subsection, we present Lemma C.6 from Abbasi-Yadkori et al. (2011), which is an important for establishing the $O(\sqrt{T})$ worst case regret for linear bandits or RL with linear function approximation. Moreover, we also present the conservatism of Elliptical Potentials in Lemma C.7 from Hu et al. (2022), which states that Elliptical Potentials are usually small.

**Lemma C.6** (Lemma 11 in Abbasi-Yadkori et al. (2011)). *Given $\lambda > 0$ and sequence $\{\boldsymbol{x}_t\}_{t=1}^T \subset \mathbb{R}^d$ with $\|\boldsymbol{x}_t\|_2 \leqslant L$ for all $t \in [T]$, define $\mathbf{Z}_t = \lambda \mathbf{I} + \sum_{i=1}^t \boldsymbol{x}_i \boldsymbol{x}_i^{\top}$ for $t \geqslant 1$ and $\mathbf{Z}_0 = \lambda \mathbf{I}$. We have*

$$\sum_{t=1}^T \min \left\{ 1, \|\boldsymbol{x}_t\|_{\mathbf{Z}_{t-1}^{-1}}^2 \right\} \leqslant 2d \log \left( 1 + \frac{TL^2}{d\lambda} \right)$$

**Lemma C.7** (Conservatism of Elliptical Potentials, Lemma D.8 in Hu et al. (2022)). *Given $\lambda > 0$ and sequence $\{\boldsymbol{x}_t\}_{t=1}^T \subset \mathbb{R}^d$ with $\|\boldsymbol{x}_t\|_2 \leqslant L$ for all $t \in [T]$, define $\mathbf{Z}_t = \lambda \mathbf{I} + \sum_{i=1}^t \boldsymbol{x}_i \boldsymbol{x}_i^{\top}$ for $t \geqslant 1$ and $\mathbf{Z}_0 = \lambda \mathbf{I}$. During $[T]$, the number of times $\|\boldsymbol{x}_t\|_{\mathbf{Z}_{t-1}^{-1}} \geqslant c$ is at most*

$$\frac{3d}{\log(1 + c^2)} \log \left( 1 + \frac{L^2}{\lambda \log(1 + c^2)} \right),$$

*where $c > 0$ is a constant.*

## D COMPUTATIONAL TRACTABILITY

In this section, we analyze the space and computational complexity of the LSVI-RFE algorithm. Notice that Algorithm 2 in the planning phase only performs a single run of value iteration to compute the output policy $\{\pi_h\}_{h \in [H]}$, the computational requirement of Algorithm 2 in the planning phase is equivalent to a single-episode run of Algorithm 1. Thus, it suffices to only analyze the space and computation complexity of Algorithm 1 in the exploration phase. Though we consider the linear MDP setting where the size of states $|\mathcal{S}|$ might be infinite, Algorithm 1 is computationally-efficient, i.e., the space and computational complexities are polynomial in $d, H, K$ and $|\mathcal{A}|$, and do not depend on $|\mathcal{S}|$, where $K$ is the number of episodes that Algorithm 1 has run.

### D.1 SPACE COMPLEXITY OF ALGORITHM 1

Though we give the explicit form of $\widehat{\boldsymbol{\mu}}_{k,h} \in \mathbb{R}^{d \times |\mathcal{S}|}$ in Algorithm 1, we do not need to store it directly. In fact, we only need to store $\widehat{\boldsymbol{\mu}}_{k,h} \widehat{\boldsymbol{V}}_{k,h+1} \in \mathbb{R}^d$ since only this term is used in computing the Q-function $\widehat{Q}_{k,h}(s, a)$ for fixed $s$ and $a$. Moreover, we only explore $\{s_h^k : h \in [H], k \in [K]\}$ in Algorithm 1. Thus, Algorithm 1 only needs to store $\widehat{\boldsymbol{\mu}}_{k,h} \widehat{\boldsymbol{V}}_{k,h+1}, \widehat{\boldsymbol{\Lambda}}_{k,h}, \widehat{\sigma}_{k,h}, \boldsymbol{\phi}(s_h^k, a)$ for $h \in [H], k \in [K], a \in \mathcal{A}$. The total space complexity is $O(d^2 H + d|\mathcal{A}|HK)$, where $K$ is the number of episodes that Algorithm 1 has run.

### D.2 COMPUTATIONAL COMPLEXITY OF ALGORITHM 1

In Algorithm 1, Lines 6-12 show the methods to calculate the estimated value function $\widehat{V}_{k,h}$ and policy $\pi_h^k(\cdot)$. Notice that only $\widehat{\boldsymbol{\mu}}_{k,h}\widehat{V}_{k,h+1}$ and $\pi_h^k(s_h^k)$ contribute to the observation steps (Line 15) and calculation of $\widehat{\sigma}_{k,h}$, $\widehat{\boldsymbol{\Lambda}}_{k+1,h}$ and $\widehat{\boldsymbol{\mu}}_{k+1,h}$ (Lines 16-26). We consider the computational cost of the following 3 parts for fixed $k \in [K]$.

1. **Calculating policy** $\pi_h^k(s_h^k)$ **for** $s_h^k$, $h \in [H]$. Assume that we have already observed $s_h^k$ for some $h \in [H]$ and calculated $\widehat{\boldsymbol{\mu}}_{k,h}\widehat{V}_{k,h+1}$. By definition, $\pi_h^k(s_h^k) = \arg\max_{a \in \mathcal{A}} \widehat{Q}_{k,h}(s_h^k, a)$. We can determine $\pi_h^k(s_h^k)$ by calculating $\widehat{Q}_{k,h}(s_h^k, a)$ for all $a \in \mathcal{A}$. Notice that $\widehat{Q}_{k,h}(s_h^k, a) = 3\widehat{\beta}^E \|\boldsymbol{\phi}(s_h^k, a)\|_{\widehat{\boldsymbol{\Lambda}}_{k,h}^{-1}} + \langle \widehat{\boldsymbol{\mu}}_{k,h}\widehat{V}_{k,h+1}, \boldsymbol{\phi}(s_h^k, a) \rangle$, which costs $O(d^2)$ operations for every $a \in \mathcal{A}$. Thus we can calculate $\pi_h^k(s_h^k)$ in $O(d^2|\mathcal{A}|H)$ for every $h \in [H]$.

2. **Calculating** $\widehat{\boldsymbol{\mu}}_{k+1,h}\widehat{V}_{k+1,h+1}$ **for** $h \in [H]$. In Section D.1, we show that we only need to store $\widehat{\boldsymbol{\mu}}_{k,h}\widehat{V}_{k,h+1}$ for every $k \in [K]$ and $h \in [H]$. Thus, instead of calculating $\widehat{\boldsymbol{\mu}}_{k+1,h}$ (Line 26) and $\widehat{V}_{k+1,h}$ (Line 10) in the next episode, we can calculate $\widehat{\boldsymbol{\mu}}_{k+1,h}\widehat{V}_{k+1,h}$ in the next episode by

$$\widehat{\boldsymbol{\mu}}_{k+1,h}\widehat{V}_{k+1,h+1} = \widehat{\boldsymbol{\Lambda}}_{k+1,h}^{-1} \sum_{i=1}^{k} \widehat{\sigma}_{i,h}^{-2} \boldsymbol{\phi}(s_h^i, a_h^i)\widehat{V}_{k+1,h}(s_{h+1}^i).$$

Hence we need to calculate $\widehat{V}_{k+1,h}(s_{h+1}^i)$ for $i \in [k]$, which means we have to calculate $\widehat{Q}_{k+1,h}(s_{h+1}^i, a)$ for any $i \in [k]$ and $a \in \mathcal{A}$. This will take $O(d^2|\mathcal{A}|K)$ operations. With known $\widehat{V}_{k+1,h}(s_{h+1}^i)$, we need $O(d^2K)$ operations to calculate the left term. Thus, calculating $\widehat{\boldsymbol{\mu}}_{k+1,h}\widehat{V}_{k+1,h+1}$ will take $O(d^2|\mathcal{A}|HK)$ operations for every $h \in [H]$ and fixed $k \in [K]$.

3. **Calculating** $\widehat{\sigma}_{k,h}$, $\widehat{\boldsymbol{\Lambda}}_{k+1,h}$. Notice that we can compute $\widetilde{\boldsymbol{\Lambda}}_{k+1,h}$, $\widehat{\boldsymbol{\Lambda}}_{k+1,h}$ by the Sherman-Morrison formula, which takes $O(d^2)$ operations. By Lines 16-24, we know that calculating $\widehat{\sigma}_{k,h}$ will take another $O(d^2)$ operations.

By arguments above, for episode $k$, we need $O(d^2|\mathcal{A}|HK)$ operations. Thus, the full algorithm costs $O(d^2|\mathcal{A}|HK^2)$, which is a polynomial in $d$, $|\mathcal{A}|$, $H$ and $K$, and independent on $|\mathcal{S}|$.

