# OpenReview forum: "Towards Minimax Optimal Reward-free Reinforcement Learning in Linear MDPs"
_ICLR.cc/2023/Conference — ICLR 2023 poster_

### Official Review · Reviewer_nPcg · 2022-10-21

**Confidence:** 3
**Correctness:** 4
**Technical Novelty And Significance:** 3
**Empirical Novelty And Significance:** Not applicable
**Recommendation:** 8

**Clarity, Quality, Novelty And Reproducibility:**

## Clarity

The paper is extremely well written and easy to follow, although a bit on the technical side. I listed some small comments on writing earlier up in the review. The two most useful clarifications which I think could help:
- Provide an overall algorithm sketch before the discussion in section 4 instead of just directly highlighting the differences of the proposed algorithm to the related literature
However, I appreciated the proof sketch, and table that were included in the introduction to help conceptualize the main contributions in thep aper.

## Quality + Novelty

The paper provides an improved analysis of upper and lower bounds for reward-free exploration in linear MDPs.  The theoretical results both help close the gap between what sample complexity is achievable (improving an $H$ factor in the lower bound) and also designing a novel algorithm with improved sample complexity (improving an $H$ factor again).  This helps to slowly close the gap (now only an $H$ factor missing) between the achievable upper and impossibility results from lower bounds in this problem.  Through developing the algorithm the authors propose a novel Bersnstein bound with elliptical potentials in order to improve and obtain the optimal dependency on $d$ for the algorithmic results.

**Strength And Weaknesses:**

## Strengths
1. The authors tighten the lower bound and prove new state-of-the-art sample complexity guarantees for the well-studied reward free learning in linear MDPs.
2. The authors present a simple algorithmic framework using existing techniques (least squares regression of the value function under "bonus"-tuned reward functions) allowing them to improve on the sample complexity required.

## Weaknesses
1. The authors include no discussion on the computational complexity of the algorithm.
2. The authors provide no empirical results of their algorithm's performance
3 The algorithmic design seems like a straightforward extension of prior mechanisms for linear function approximation for the reward-aware setting. The algorithmic contributions could be highlighted more in the writing.


**Summary Of The Paper:**

The authors study reward-free reinforcement learning in linear MDPs.  In this setting, the agent first interacts with the environment for some fixed number of timesteps without knowledge of the downstream reward function it is trying to optimize.  Based on the estimated transition distribution from samples obtained in this exploration phase, the agent is given a reward function and is tasked with outputting an $\epsilon$-optimal policy.  This paper focuses on the linear MDP setting, where the rewards and transition kernel can be written in terms of a known feature representation of an unknown vector (or distribution).  The authors show a novel algorithm with an $H^4 d^2 / \epsilon^2$ sample complexity bound, where $H$ is the episode length and $d$ is dimension of features.  They also show a lower bound of $H^3 d^2 / \epsilon^2$, essentially showing their result is tight up to a factor of $H$.  The algorithm is based on new concentration based on Bernstein confidence sets to improve upon the sample complexity results of prior literature.

More concretely, the authors consider an MDP defined as $(S,A,P,r,H)$ where $S$ and $A$ are state-action space, $P$ transitions, $r$ reward, and $H$ horizon.  They assume MDP is linear, i.e. known features $\phi$ of $S \times A$ which map to a $d$ dimensional vector with $r = \theta^\top \phi$ and $P( = \phi^\top \mu$ where $\mu$ is a distribution over the next states.  The goal is to perform reward free exploration, i.e. design policies to explore the space with unknown reward function, and then given a reward function compute a near optimal policy.

At a high level the authors provide two contributions on this setup:
1. The authors improve the best known algorithmic guarantee to be $H^4 d^2 / \epsilon^2$.  The algorithmic framework follows the standard form (use reward as "exploration" bonuses in the data collection phase), and then solve planning problem over estimated transitions once given a reward function.  However, they modify the algorithmic sketch by introducing novel confidence terms with Bernstein concentration to obtain the new results.
2. The authors improve the lower bound result to be $H^3 d^2 / \epsilon^2$ improving on the best known results.  They highlight how the lower bound is novel, and uses existing techniques from the linear MDP lower bound setup.

## Questions
- Based on the discussion on page 9 - do you believe that the gap remains in the achievability portion (i.e. designing an even better algorithm to save the extra $H$ factor) or the impossibility result?

## Minor Comments
- Defining sample complexity on page 4 should highlight that it should hold for all $r$
- vspace issues on page 6 before algorithm 2

**Summary Of The Review:**

The paper provides strong theoretical contributions for improving upper and lower bounds for reward-free exploration in linear MDPs, which improves the state-of-the-art bounds in this setting.  The paper is well written, although very technical, but highlights the main differences and contributions between the paper and the related work.  However, the paper offers no empirical results (or discussion on computational tractability of their proposed algorithm), although standard in the RL with linear function approximation literature.

---

> ### Author Response · Authors · 2022-11-16
> **Response to Reviewer nPcg (part 1 of 2)**
>
> Thank you very much for your positive and valuable feedback! In the following, we respond to your comments in detail.
> ### Questions
> > Q1: Based on the discussion on page 9 - do you believe that the gap remains in the achievability portion (i.e. designing an even better algorithm to save the extra H factor) or the impossibility result?
>
> We believe that the gap remains in the achievability portion. Yet, it requires to design a new algorithm framework. The reason remains in our algorithm framework, when trying to utilize the Law of Total Variance (LTV) to save off an $H$ factor, we need to estimate the variance of the value concerning the actual reward function, which is infeasible since the actual value function is unknown in the exploration phase. The recent work of Chen et al. (2021) achieved the minmax optimality in reward-free exploration for linear mixture MDPs by LTV. Instead of designing a uniform upper bound of the value function variance, they chose a special exploration reward and estimator of probability transition in every episode from a feasible set to maximize the estimated value function. Chen et al. (2021) shed light on designing a minimax optimal algorithm for reward-free linear MDPs.
> ### Weaknesses
> > W1:  The authors include no discussion on the computational complexity of the algorithm.
>
> Thanks for your insightful comments, we have included the discussion of the computational complexity in Appendix D in our revision.
>
> > W2:  However, the paper offers no empirical results (or discussion on computational tractability of their proposed algorithm), although standard in the RL with linear function approximation literature.
>
> This is a theoretical paper for a dedicated, well-formulated problem, i.e., reward-free exploration in linear MDPs. We focus primarily on providing sound proof for our theoretical results, as done in existing works for reward-free linear MDPs, e.g., Wang et al. (2020a), Zanette et al. (2020c), Chen et al. (2021) and Wagenmaker et al. (2022). We have included the discussion of the computational tractability in Appendix D in our revision.

---

> > ### Author Response · Authors · 2022-11-16
> > **Response to Reviewer nPcg (part 2 of 2)**
> >
> > > W3: The algorithmic design seems like a straightforward extension of prior mechanisms for linear function approximation for the reward-aware setting. The algorithmic contributions could be highlighted more in the writing.
> >
> > The key novelty of LSVI-RFE, i.e., the aggressive variance-aware exploration mechanism, although inspired by prior works including Hu et al. (2022) and Chen et al. (2021), contains critical differences in variance-aware weights, value function monotonicity, and computational tractability compared to these two works, which are clarified better in our revision and illustrated below.
> >
> > 1. The aggressive exploration bonus $b_{k,h}$ guarantees the monotonicity of constructed value functions between two phases, and removes the additional dependency of sample complexity on feature dimension $d$. The design of the aggressive exploration bonus $b_{k,h}$, which contains the variance-aware weights (associating with the weighted linear regression), is inspired by the prior work of Hu et al. (2022) on regret minimization in linear MDPs. However, our exploration bonus $b_{k,h}$ contains the following two significant differences:
> >     - We design different variance-aware weights. The variance-aware weights $\widehat{\sigma}\_{k,h} = \max \\{w\_{k,h}, \widetilde{\sigma}\_{k,h}\\}$ in $b_{k,h}$ only includes two terms, while in Hu et al. (2022), it contains three terms. The significant difference remains that we abandon the variance upper estimator term concerning the real value function based on the actual reward function. Due to the agnosticism of the actual reward function in the exploration phase, a uniform upper bound of the value function (concerning the actual reward function) variance is infeasible.
> >     - We build a different value function monotonicity. Our monotonicity of constructed value functions is built between the exploration phase and the planning phase, i.e., $V\_h^*(\cdot;r)+\hat{V}\_{k,h}(\cdot) \geq \hat{V}\_h(\cdot)$ for any $k \in [K]$, detailed in Lemma A. 15 in Appendix. This is similar to the so-called "over-optimism" in Hu et al. (2022), but with the critical difference that our over-optimism is built between the exploration and the planning phases, instead of each episode in Hu et al. (2022). In addition, it is our over-optimism between the two phases that helps build a sharp self-normalized bound in the reward-free setting, as detailed in Lemma A.12 in Appendix.
> > 2. Moreover, our exploration-driven reward function $r_{k,h}$ in the exploration phase is more aggressive than those in prior works Wang et al. (2020a), Zanette et al. (2020c) by an $H$ factor to avoid overly-conservative exploration. This aggressiveness in the reward function removes the extra dependency of sample complexity on episode length $H$. Although the $H$ factor aggressiveness is inspired by Chen et al. (2021), we extend it from a computationally-inefficient algorithm to a computationally-efficient one, and we incorporate it with a variance-aware exploration mechanism, saving an $H^2$ factor in the sample complexity.
> >
> > > Minor Comments
> >
> > Thanks for your detailed review. We have revised all these comments in the revision.
> >
> > ### Clarity
> > > Provide an overall algorithm sketch before the discussion in section 4 instead of just directly highlighting the differences of the proposed algorithm to the related literature However, I appreciated the proof sketch, and table that were included in the introduction to help conceptualize the main contributions in the paper.
> >
> > Thanks for the helpful suggestions. We have included overall algorithm sketches in the starting paragraph of Section 4.1 and Section 4.2 for the exploration and planning phase, respectively, in our revision.
> > ***
> > ***Finally, we thank the reviewer again for  the detailed and valuable comments. If our response resolves your concerns satisfactorily, we want to kindly ask the reviewer to consider raising the score rating of our work. We will also be happy to answer any further questions you may have during the discussion.***

---

> > > ### Comment · Reviewer_nPcg · 2022-11-19
> > > **Response**
> > >
> > > Thanks for addressing all of my comments. I appreciate the updated discussion in the revision on the modification of the prior work on the linear function approximation for the reward-aware setting.  This helps highlight the theoretical advances required in the analysis.

---

> > > > ### Author Response · Authors · 2022-11-22
> > > > **Thanks**
> > > >
> > > > Thank you very much for your help in improving the paper! We really appreciate your positive and valuable response.

---

### Official Review · Reviewer_Dide · 2022-10-24

**Confidence:** 3
**Correctness:** 4
**Technical Novelty And Significance:** 3
**Empirical Novelty And Significance:** Not applicable
**Recommendation:** 6

**Clarity, Quality, Novelty And Reproducibility:**

The clarity of this paper is good in general. However, its novelty seems to be limited as discussed in the previous section.

#### Questions
- Can you explain how the magnitudes for $w_{k, h}$ are chosen in two different conditions at line 20 and 22 in Algorithm 1?
- Do the techniques and analysis of variance-aware weights and weighted linear regression in this paper have any significant difference from those in [Hu et al. (2022)]?


```
[Hu et al. (2022)] Pihe Hu, Yu Chen, and Longbo Huang. Nearly minimax optimal reinforcement learning with linear function approximation. In International Conference on Machine Learning, pp. 8971–9019. PMLR, 2022.
```

**Strength And Weaknesses:**

### Strengths
This paper proves a first sample complexity bound that is optimal in $d$ by improving over the previous lower bound. Meanwhile, the paper is also well-written with a clear logical flow.

### Weaknesses
It seems most of the technical innovations claimed in Algorithm 1 directly come from [Hu et al. (2022)]. Indeed, it is by no means trivial to derive all details step-by-step. However, it will be better if the paper can clarify whether the techniques in Algorithm 1 that are similar to those in [Hu et al. (2022)] actually contain significant difference.

On the other hand, the improvement seems to be marginal since [Wagenmaker et al. (2022)] has already achieved optimal dependence in $d$. It will be more significant if the results in this paper can also be optimal in $H$.

```
[Hu et al. (2022)] Pihe Hu, Yu Chen, and Longbo Huang. Nearly minimax optimal reinforcement learning with linear function approximation. In International Conference on Machine Learning, pp. 8971–9019. PMLR, 2022.
[Wagenmaker et al. (2022)] Andrew Wagenmaker, Yifang Chen, Max Simchowitz, Simon S Du, and Kevin Jamieson. Rewardfree rl is no harder than reward-aware rl in linear markov decision processes. arXiv preprint arXiv:2201.11206, 2022.
```

**Summary Of The Paper:**

This paper studies the reward-free reinforcement learning in linear MDPs. In particular, by utilizing techniques from the minimax optimal algorithm for ordinary reinforcement learning in linear MDPs, this paper improves the sample complexity to an order of $\widetilde{O}(H^4d^2/\epsilon^2)$. Meanwhile, it also improves the lower bound to an order of $\Omega(H^3d^2/\epsilon^2)$, thus matching the lower bound in terms of dimension $d$.

**Summary Of The Review:**

This paper improves the upper bound and lower bound for reward-free reinforcement learning in linear MDPs to achieve a sample complexity that is optimal in $d$. However, the technical novelty and significance semm to be limited.

---

> ### Author Response · Authors · 2022-11-16
> **Response to Reviewer Dide (part 1 of 3)**
>
> Thank you very much for your valuable feedback. In the following, we respond to your comments in detail.
> ### Weaknesses
> > W1: It seems most of the technical innovations claimed in Algorithm 1 directly come from [Hu et al. (2022)]. Indeed, it is by no means trivial to derive all details step-by-step. However, it will be better if the paper can clarify whether the techniques in Algorithm 1 that are similar to those in [Hu et al. (2022)] actually contain significant difference.
>
> The key novelty of LSVI-RFE, i.e., the aggressive variance-aware exploration mechanism, although inspired by prior works including Hu et al. (2022) and Chen et al. (2021), contains critical differences in variance-aware weights, value function monotonicity, and computational tractability compared to these two works, which are clarified better in our revision and illustrated below.
>
> 1. The aggressive exploration bonus $b_{k,h}$ guarantees the monotonicity of constructed value functions between two phases, and removes the additional dependency of sample complexity on feature dimension $d$. The design of the aggressive exploration bonus $b_{k,h}$, which contains the variance-aware weights (associating with the weighted linear regression), is inspired by the prior work of Hu et al. (2022) on regret minimization in linear MDPs. However, our exploration bonus $b_{k,h}$ contains the following two significant differences:
>     - We design different variance-aware weights. The variance-aware weights $\widehat{\sigma}\_{k,h} = \max \\{w\_{k,h}, \widetilde{\sigma}\_{k,h}\\}$ in $b_{k,h}$ only includes two terms, while in Hu et al. (2022), it contains three terms. The significant difference remains that we abandon the variance upper estimator term concerning the real value function based on the actual reward function. Due to the agnosticism of the actual reward function in the exploration phase, a uniform upper bound of the value function (concerning the actual reward function) variance is infeasible.
>     - We build a different value function monotonicity. Our monotonicity of constructed value functions is built between the exploration phase and the planning phase, i.e., $V\_h^*(\cdot;r)+\hat{V}\_{k,h}(\cdot) \geq \hat{V}\_h(\cdot)$ for any $k \in [K]$, detailed in Lemma A. 15 in Appendix. This is similar to the so-called "over-optimism" in Hu et al. (2022), but with the critical difference that our over-optimism is built between the exploration and the planning phases, instead of each episode in Hu et al. (2022). In addition, it is our over-optimism between the two phases that helps build a sharp self-normalized bound in the reward-free setting, as detailed in Lemma A.12 in Appendix.
> 2. Moreover, our exploration-driven reward function $r_{k,h}$ in the exploration phase is more aggressive than those in prior works Wang et al. (2020a), Zanette et al. (2020c) by an $H$ factor to avoid overly-conservative exploration. This aggressiveness in the reward function removes the extra dependency of sample complexity on episode length $H$. Although the $H$ factor aggressiveness is inspired by Chen et al. (2021), we extend it from a computationally-inefficient algorithm to a computationally-efficient one, and we incorporate it with a variance-aware exploration mechanism, saving an $H^2$ factor in the sample complexity.
>
> > W2:  On the other hand, the improvement seems to be marginal since [Wagenmaker et al. (2022)] has already achieved optimal dependence in d. It will be more significant if the results in this paper can also be optimal in H.
>
> Compared to Wagenmaker et al. (2022), our contributions are twofold and are also clarified better in our revision.
>
> - (1) our sample complexity upper bound $\widetilde{O}(H^4d^2/\epsilon^2)$ is sharper by a factor of $H$, (2) and the lower bound $\Omega(H^3d^2/\epsilon^2)$ is also tighter by a factor of $H$. In fact, there is only an $H$ gap to reach the minimax optimal reward-free reinforcement learning in linear MDPs. We agree with the reviewer that removing the $H$ gap will be an interesting future work.
>
> - Our technical framework is very different from that of Wagenmaker et al. (2022). Unlike traversing a linear MDP to collect samples in a given feature direction in Wagenmaker et al. (2022), we design an aggressive variance-aware exploration mechanism in the exploration phase, which provides a new exploration mechanism to reach optimal dependency on $d$ and better dependency on $H$.

---

> > ### Author Response · Authors · 2022-11-16
> > **Response to Reviewer Dide (part 2 of 3)**
> >
> > > W3: This paper improves the upper bound and lower bound for reward-free reinforcement learning in linear MDPs to achieve a sample complexity that is optimal in $d$. However, the technical novelty and significance seem to be limited.
> >
> > - **Novelty**: The key technical novelty of LSVI-RFE, i.e., the aggressive variance-aware exploration mechanism, although inspired by prior works including Hu et al. (2022) and Chen et al. (2021), contains critical differences in variance-aware weights, value function monotonicity, and computational tractability compared to these two works, which are clarified better in our revision and illustrated in W1.
> > - **Significance**: LSVI-RFE achieves state-of-the-art results for reward-free RL in linear MDPs. It is proved to have a sample complexity upper bound of $\widetilde{O}(H^4d^2/\epsilon^2)$, and we also establish a lower bound of $\Omega(H^3d^2/\epsilon^2)$. This implies LSVI-RFE is the first algorithm to reach minimax optimality for reward-free RL in linear MDPs, up to an $H$ and logarithmic factor.s We agree with the reviewer that removing the $H$ gap will be an interesting future work, which has been discussed in Section 6.1.
> >
> > ### Questions
> > > Q1: How the magnitudes for w_{k,h} are chosen in 2 different conditions at line 20 and line 22 in Algorithm 1?
> >
> > Lines 19-22 are designed for keeping the weighted elliptical potential, i.e., $\\|\widehat{\sigma}\_{k,h}^{-1}\boldsymbol{\phi}(s\_h^k, a\_h^k)\\|\_{\widehat{\mathbf{\Lambda}}\_{k,h}^{-1}}/\widehat{\sigma}\_{k,h}$, small.
> >
> > - If $\\|\widetilde{\sigma}\_{k,h}^{-1}\boldsymbol{\phi}(s\_{h}^k,a\_{h}^k)\\|\_{\widetilde{\mathbf{\Lambda}}\_{k,{h}}^{-1}}$ is small, i.e. $\\|\widetilde{\sigma}\_{k,h}^{-1}\boldsymbol{\phi}(s\_{h}^k,a\_{h}^k)\\|\_{\widetilde{\mathbf{\Lambda}}\_{k,{h}}^{-1}}\le1/d^3$, $w\_{k,h}$ takes default value of $\sqrt{H}$. Thus, $\\|\widehat{\sigma}\_{k,h}^{-1}\boldsymbol{\phi}(s\_h^k, a\_h^k)\\|\_{\widehat{\mathbf{\Lambda}}\_{k,h}^{-1}}/\widehat{\sigma}\_{k,h}$ is small since $\\|\widetilde{\sigma}\_{k,h}^{-1}\boldsymbol{\phi}(s\_{h}^k,a\_{h}^k)\\|\_{\widetilde{\mathbf{\Lambda}}\_{k,{h}}^{-1}}$ is small.
> >
> > - Otherwise, $w\_{k,h}$ is enlaged to $\sqrt{Hd^3}$ such that $\widehat{\sigma}\_{k,h}=\max\\{w\_{k,h},\widetilde{\sigma}\_{k,h}\\}$ is enlarged to keep the weighted elliptical potential, i.e. $\\|\widehat{\sigma}\_{k,h}^{-1}\boldsymbol{\phi}(s\_h^k, a\_h^k)\\|\_{\widehat{\mathbf{\Lambda}}\_{k,h}^{-1}}/\widehat{\sigma}\_{k,h}$ small.
> >
> > These two situations are also detailed in Lemma A.9 in Appendix. The motivation to control the weighted elliptical potential $\\|\widehat{\sigma}\_{k,h}^{-1}\boldsymbol{\phi}(s\_h^k, a\_h^k)\\|\_{\widehat{\mathbf{\Lambda}}\_{k,h}^{-1}}/\widehat{\sigma}\_{k,h}$ small is that this term will appear in the self-normalized bound when building confidence sets, e.g., Lemma A.12 for confidence set $\widehat{\mathcal{C}}\_{h}^{P(2)}$ in Appendix. Thus, the uncertainty in the exploration phase is decreased since the confidence set radius (i.e. the self-normalized bound) is squeezed.

---

> > > ### Author Response · Authors · 2022-11-17
> > > **Response to Reviewer Dide (part 3 of 3)**
> > >
> > > > Q2: Do the techniques and analysis of variance-aware weights and weighted linear regression in this paper have any significant difference from those in [Hu et al. (2022)]?
> > >
> > > Although the aggressive exploration bonus $b\_{k,h}$, which contains the variance-aware weights (associating with the weighted linear regression), is inspired by the prior work of Hu et al. (2022) on regret minimization in linear MDPs, our exploration bonus $b\_{k,h}$ contains the following two significant differences:
> > > - We design different variance-aware weights. The variance-aware weights $\widehat{\sigma}\_{k,h} = \max \\{w\_{k,h}, \widetilde{\sigma}\_{k,h}\\}$ in $b_{k,h}$ only includes two terms, while in Hu et al. (2022), it contains three terms. The significant difference remains that we abandon the variance upper estimator term concerning the real value function based on the actual reward function. Due to the agnosticism of the actual reward function in the exploration phase, a uniform upper bound of the value function (concerning the actual reward function) variance is infeasible.
> > > - We build a different value function monotonicity. Our monotonicity of constructed value functions is built between the exploration phase and the planning phase, i.e., $V\_h^*(\cdot;r)+\hat{V}\_{k,h}(\cdot) \geq \hat{V}\_h(\cdot)$ for any $k \in [K]$, detailed in Lemma A. 15 in Appendix. This is similar to the so-called "over-optimism" in Hu et al. (2022), but with the critical difference that our over-optimism is built between the exploration and the planning phases, instead of each episode in Hu et al. (2022). In addition, it is our over-optimism between the two phases that helps build a sharp self-normalized bound in the reward-free setting, as detailed in Lemma A.12 in Appendix.
> > > ***
> > > ***Finally, we thank the reviewer again for the valuable comments. If our response resolves your concerns satisfactorily, we would like to kindly ask the reviewer to consider raising the score rating for our work. We will also be happy to answer any further questions you may have during the discussion.***

---

> > ### Comment · Reviewer_Dide · 2022-11-25
> > **Response**
> >
> > Thank you very much for your detailed response! My concern has been well-addressed and thus I would like to increase my score.

---

> > > ### Author Response · Authors · 2022-11-26
> > > **Thanks**
> > >
> > > Thank you very much for your help in improving our paper! We really appreciate your valuable comments and positive response.

---

> ### Author Response · Authors · 2022-11-22
> **Reminder to Reviewer Dide**
>
> Dear reviewer,
>
> Thank you for your effort in reviewing our paper!
>
> We wonder whether our reply fully addresses your concerns. If so, could you please consider raising your score for our work?
> Please let us know if you have any further questions. We will be more than happy to discuss this with you and answer any remaining questions.
>
> Thank you very much!

---

### Official Review · Reviewer_FkPf · 2022-10-24

**Confidence:** 3
**Clarity, Quality, Novelty And Reproducibility:** See above
**Correctness:** 3
**Technical Novelty And Significance:** 3
**Empirical Novelty And Significance:** Not applicable
**Recommendation:** 6

**Strength And Weaknesses:**

In general, this paper is an extension from Hu et al. (2022) who studies the regret minimization problem in linear MDPs to the reward free setting. The key technical novelties include the variance-aware weighting mechanism to estimate the transition dynamics, the over-optimistic bonus term in the exploration phase, and a larger exploration function in terms of $H$. Although these techniques have been proposed by Hu et al. (2022) and Chen et al. (2021), it requires effort to adapted them to the reward free setting. However, the results in this paper is not that surprising, given previous works such as Hu et al. (2022) and Chen et al. (2021).

Importantly, I found it hard to understand a few sentences in the main text, mainly because the authors used involved explanations instead of straight forward sentences. For example, what does it mean by "negligible cost" at the bottom of page 5? How does the mentioned term relate to the total uncertainty in the exploration phase? Another example is the first sentence in Remark 4.2. It is confusing to directly say "auxiliary value functions" and "decouple UCB bonuses" since this function and meaning of decouple are not defined at all. I suggest the authors use more straight forward sentences to express their ideas.

Lastly, I want to mention that the line spacing on page 5 and 6 seems to be abnormal, the lines are too much squeezed.

Minor comments and typos:
- Line 9 of Algorithm 1: $\hat{V}_{k,h}$ -- $\hat{V}_{k,h+1}$
- Line 7 of Algorithm 1: it seems that the constant should be 4 according to the proof in Lemma A.15
- Section 4: instatanous -- instantaneous
- Why does the first inequality (the $\sqrt{d}$ term) of Equation (9) hold?

**Summary Of The Paper:**

This paper studies the reward free exploration problem in the linear MDP setting. It proposes a sample-efficient and computational-efficient algorithm LSVI-RFE, which returns an $\epsilon$-optimal policy given any reward function using $\tilde{O}(d^2H^4/\epsilon^2)$ exploration trajectories, which is tighter than previous works. Moreover, it shows that the lower bound of sample complexity of reward free exploration in linear MDPs is at least $\Omega(d^2H^3/\epsilon^2)$.


**Summary Of The Review:**

This paper proposes a reward free exploration algorithm in linear MDPs with improved sample complexity. It also provides a lower bound by reducing the reward free problem to the standard linear MDP exploration problem. The paper provides rigorous proofs for all the statements, but some formulas and sentences are a bit confusing. Overall, I vote for acceptance, but I strongly recommend the authors to clarify their expressions and clean the typos in the technical part of the paper.

---

> ### Author Response · Authors · 2022-11-16
> **Response to Reviewer FkPf (part 1 of 2)**
>
> Thank you very much for your positive and valuable feedback! In the following, we respond to your comments in detail.
> ### Weaknesses
> > W1: However, the results in this paper is not that surprising, given previous works such as Hu et al. (2022) and Chen et al. (2021).
>
> The key novelty of LSVI-RFE, i.e., the aggressive variance-aware exploration mechanism, although inspired by prior works including Hu et al. (2022) and Chen et al. (2021), contains critical differences in variance-aware weights, value function monotonicity, and computational tractability compared to these two works, which are clarified better in our revision and illustrated below.
> 1. The aggressive exploration bonus $b_{k,h}$ guarantees the monotonicity of constructed value functions between two phases, and removes the additional dependency of sample complexity on feature dimension $d$. The design of the aggressive exploration bonus $b_{k,h}$, which contains the variance-aware weights (associating with the weighted linear regression), is inspired by the prior work of Hu et al. (2022) on regret minimization in linear MDPs. However, our exploration bonus $b_{k,h}$ contains the following two significant differences:
>     - We design different variance-aware weights. The variance-aware weights $\widehat{\sigma}\_{k,h} = \max \\{w\_{k,h}, \widetilde{\sigma}\_{k,h}\\}$ in $b_{k,h}$ only includes two terms, while in Hu et al. (2022), it contains three terms. The significant difference remains that we abandon the variance upper estimator term concerning the real value function based on the actual reward function. Due to the agnosticism of the actual reward function in the exploration phase, a uniform upper bound of the value function (concerning the actual reward function) variance is infeasible.
>     - We build a different value function monotonicity. Our monotonicity of constructed value functions is built between the exploration phase and the planning phase, i.e., $V\_h^*(\cdot;r)+\hat{V}\_{k,h}(\cdot) \geq \hat{V}\_h(\cdot)$ for any $k \in [K]$, detailed in Lemma A. 15 in Appendix. This is similar to the so-called "over-optimism" in Hu et al. (2022), but with the critical difference that our over-optimism is built between the exploration and the planning phases, instead of each episode in Hu et al. (2022). In addition, it is our over-optimism between the two phases that helps build a sharp self-normalized bound in the reward-free setting, as detailed in Lemma A.12 in Appendix.
> 2. Moreover, our exploration-driven reward function $r_{k,h}$ in the exploration phase is more aggressive than those in prior works Wang et al. (2020a), Zanette et al. (2020c) by an $H$ factor to avoid overly-conservative exploration. This aggressiveness in the reward function removes the extra dependency of sample complexity on episode length $H$. Although the $H$ factor aggressiveness is inspired by Chen et al. (2021), we extend it from a computationally-inefficient algorithm to a computationally-efficient one, and we incorporate it with a variance-aware exploration mechanism, saving an $H^2$ factor in the sample complexity.

---

> > ### Author Response · Authors · 2022-11-16
> > **Response to Reviewer FkPf (part 2 of 2)**
> >
> > > W2: Importantly, I found it hard to understand a few sentences in the main text, mainly because the authors used involved explanations instead of straight forward sentences. For example, what does it mean by "negligible cost" at the bottom of page 5? How does the mentioned term relate to the total uncertainty in the exploration phase? Another example is the first sentence in Remark 4.2. It is confusing to directly say "auxiliary value functions" and "decouple UCB bonuses" since this function and meaning of decouple are not defined at all. I suggest the authors use more straight forward sentences to express their ideas.
> >
> > Thanks for your constructive suggestions, and we have significantly revised our paper. The above-mentioned points are explained below.
> >
> > - "Negligible cost"：We can use $w_{k,h}$ to dynamically determine the lower bound of $\widehat{\sigma}_{k,h}$ and keep the weighted elliptical potential, i.e.,
> > $\\|\widehat{\sigma}\_{k,h}^{-1}\boldsymbol{\phi}(s_h^k, a_h^k)\\|\_{\widehat{\mathbf{\Lambda}}\_{k, h}^{-1}}/\widehat{\sigma}\_{k,h}$
> > small, which is proved in Lemma A.9. According to Lemma C.7, we only need to enlarge $w\_{k,h}$ in at most $O(H^2d^6)$ episodes, which is not the dominant term in the summation of $\sum\_k \sum\_h \widehat{\sigma}\_{k,h}^2$. Consequently, this operation will not bring an extra dominant term in the sample complexity, as detailed in Lemma A.19. Thus, we say it has "negligible cost" to keep the desired term small.
> >
> > - Relationship to the total uncertainty: The motivation to control the weighted elliptical potential, i.e., $\\|\widehat{\sigma}\_{k,h}^{-1}\boldsymbol{\phi}(s_h^k, a_h^k)\\|\_{\widehat{\mathbf{\Lambda}}\_{k,h}^{-1}}/\widehat{\sigma}\_{k,h}$ small is that this term will appear in the self-normalized bound when building confidence sets, e.g., Lemma A.12 for confidence set $\widehat{\mathcal{C}}\_{h}^{P(2)}$ in Appendix. Thus, the uncertainty in the exploration phase is decreased since the confidence set radius (i.e. the self-normalized bound) is squeezed.
> >
> > - "Auxiliary value functions": The auxiliary value function is defined in Definition A.5, where
> > $\widetilde{V}_h^*(s;r) =\min \\{\max\_{a\in\mathcal{A}} r\_h(s,a)+\mathbb{P}\_h \widetilde{V}\_{h+1}^*(s,a;r) , H \\}$. Since $r\_{k,h}$ is not bounded by 1 in our algorithm, we need this truncated value function to establish the optimism in Lemma A.7.
> >
> > - "Decouple UCB bonuses": We can use the auxiliary value function, i.e., the truncated value function, to build the optimism between the exploration and planning phases in Lemma A.22, with two different UCB bonuses in these two phases. This is a significant difference from the existing works. In this sense, we "decouple UCB bonuses" in two phases since existing works, e.g., Wang et al. (2020a); Zhang et al. (2021), for reward-free RL with linear function approximation, require the same order of bonuses in two phases.
> >
> > > W3: Lastly, I want to mention that the line spacing on page 5 and 6 seems to be abnormal, the lines are too much squeezed.
> >
> > We have revised it in our revision.
> >
> > > W4: I strongly recommend the authors to clarify their expressions and clean the typos in the technical part of the paper.
> >
> > Thanks for your constructive suggestions. We have significantly revised our paper to clarify our expressions better and remove typos in our best.
> >
> > > Minor comments and typos:
> >
> > Thanks for your detailed review, we have revised these typos in our revision.
> >
> > About your forth comment
> > > > Why does the first inequality (the d term) of Equation (9) hold?
> >
> > This equation is detailed in Lemma A.22. The reasons are
> >
> > 1. Since $\widehat{\beta}^E=O(d\sqrt{H})$ and $\widehat{\beta}^P = O(\sqrt{dH})$, we have $\widehat{\beta}^E \geq \sqrt{d}\widehat{\beta}^P$ by neglecting constants.
> >
> > 2. By Line 4 in Algorithm 2, $b\_h(\cdot,\cdot) = \min\\{\widehat{\beta}^P\\|\boldsymbol{\phi}(\cdot,\cdot)\\|\_{\widehat{\boldsymbol{\Lambda}}\_{K+1, h}^{-1}}, H\\}$.
> >
> > 2. By Algorithm 1, we have that $\Lambda\_{K+1,h} - \Lambda\_{k,h} \succeq 0$ is a positive definite matrix, such that $\\|\phi(\cdot,\cdot)\\|\_{\Lambda\_{k,h}^{-1}} \geq \\|\phi(\cdot,\cdot)\\|\_{\Lambda\_{K+1,h}^{-1}}$.
> > ***
> > ***Finally, we want to thank the reviewer again for the detailed and valuable comments. If our response resolves your concerns satisfactorily, we want to kindly ask the reviewer to consider raising the score rating of our work. Please let us know if you have any further questions, and we will be happy to answer them.***

---

> > ### Comment · Reviewer_FkPf · 2022-11-26
> > **Thanks for your clarifications**
> >
> > Thanks for responding to my concerns. For the confusing statements mentioned in my original review, I really appreciate the clarifications made by the authors. On the other hand, by mentioning "not that surprising" I mean there should be differences between known-reward setting and reward-free setting. The problem is whether these differences are significant enough (e.g., by incorporating novel techniques beyond the known literature of reward-free setting). I believe the value function monotonicity mechanism proposed in this paper is a pretty good adaptation from known-reward setting to reward-free setting (based on Hu et al.). However, I still believe the other two novelties mentioned in the rebuttal (i.e., the aggressive rate on $H$ and the revision of the variance-aware weights $\hat{\sigma}_{k,h}$) are "not that surprising" given Hu et al. and Chen et al. The computationally efficient adaptation of the aggressive rate of $H$ seems to be a direct consequence given the (over-optimistic) framework proposed by Hu et al. Dropping the variance upper estimator term of the real value funcions are standard practice in the reward-free setting, instead one turns to construct a confidence set for any possible reward functions (the union bound is infeasible here). That is to say, I would like to keep my score.

---

> > > ### Author Response · Authors · 2022-11-28
> > > **Thanks**
> > >
> > > Thank you very much for your valuable response. Regarding your comment concerning the "not surprising" adaption of techniques, we would like to clarify some points below.
> > >
> > > > The computationally efficient adaptation of the aggressive rate of  $H$ seems to be a direct consequence given the (over-optimistic) framework proposed by Hu et al. (2022).
> > >
> > > Hu et al. (2022) did not provide any aggressive rate of $H$ enlargement in the exploration bonus. Our computationally efficient adaptation of the aggressive rate of  $H$ is inspired by Chen et al. (2021) for linear mixture settings. Yet, their method is computationally inefficient. Thus, incorporating the aggressive rate $H$ into the computationally-efficient LSVI-RFE algorithm is highly non-trivial.
> > >
> > > >  Dropping the variance upper estimator term of the real value functions are standard practice in the reward-free setting, instead one turns to construct a confidence set for any possible reward functions (the union bound is infeasible here).
> > >
> > > We agree with the reviewer that constructing a confidence set for any possible reward functions appears to be a more promising way to utilize the Berstein-type bonus. However, as shown in the existing works of  Chen et al. (2021)  and Zhang et al. (2021) that, constructing a confidence set for any possible reward functions will result in the computationally-inefficient issue in these algorithms. Moreover, a computationally-tractable version (by solving the optimization problem approximately) of these algorithms will lead to an additional $d$ degradation in sample complexity, as discussed in Corollary 4.4 in Zhang et al.(2021). In contrast, our LSVI-RFE algorithm abandons the variance upper estimator term, and our algorithm still reaches a sharper sample complexity with our aggressive variance aware-exploration mechanism. LSVI-RFE achieves optimal dependency on $d$ and is also computationally efficient.
> > >
> > > ***
> > >
> > > In addition to the above, we also would like to highlight our contributions in three aspects.
> > >
> > > - **Technical novelty:**
> > > As stated in our first response, our aggressive exploration bonus $b_{k,h}$ (containing the variance-aware weights $\widehat{\sigma}\_{k,h}$ and the weighted linear regression) and exploration-driven reward function $r_{k,h}$ are both novel for reward-free exploration in Linear MDs, and these novel components contain critical differences in variance-aware weights, value function monotonicity, and computational tractability, compared to Hu et al.(2022) and. Chen et al.(2021). Besides, it is non-trivial to combine and analyze these techniques in reward-free linear MDPs, since Hu et al.(2022) studied the regret minimization in Linear MDP RL, and Chen et al.(2021) studied the reward-free exploration in Linear mixture MDPs.
> > >
> > > - **Sharp results for reward-free linear MDPs:**
> > >  Our paper presents both the sample complexity upper bound $\widetilde{O}(H^4d^2/\epsilon^2)$ and lower bound $\Omega(H^3d^2/\epsilon^2)$ for reward-free exploration in linear MDPs, which are both state-of-the-art statistical results. Apart from the recent work of Wagenmaker et al. (2022), which also achieves the optimal dependency on $d$, we provide a new exploration mechanism to reach optimal dependency on $d$ and better dependency on $H$, which is very different from that collecting samples in a given feature direction in Wagenmaker et al. (2022). Furthermore, it also reveals that there is only a $H$ gap towards minimax optimal sample complexity in reward-free exploration for linear MDPs.
> > >
> > > - **Towards minimax optimality:**
> > > In Section 6.1, we also provide a discussion on the gap of $H$. We also discuss the barriers in the PAC setting, where we know the true reward function. These discussions also potentially improve the understanding of theoretically efficient RL.
> > >
> > >
> > > We hope our responses can clarify our contribution better. We are always happy to answer any questions you may have.

---

> ### Author Response · Authors · 2022-11-22
> **Reminder to Reviewer FkPf**
>
> Dear reviewer,
>
> Thank you for your effort in reviewing our paper!
>
> We wonder whether our reply fully addresses your concerns. If so, could you please consider raising your score for our work?
> Please let us know if you have any further questions. We will be more than happy to discuss this with you and answer any remaining questions.
>
> Thank you very much!

---

### Official Review · Reviewer_tyVT · 2022-10-24

**Confidence:** 4
**Correctness:** 3
**Technical Novelty And Significance:** 2
**Empirical Novelty And Significance:** Not applicable
**Recommendation:** 8

**Clarity, Quality, Novelty And Reproducibility:**

This paper is clearly a good add to the reinforcement learning theory community. I haven't checked the details and the correctness of the paper due to the short review timeframe.

To summarize the contribution of the paper. On the upper bound side, the d^2 sample complexity in reward-free linear MDPs has just recently been shown in the paper by Wagenmaker et al. (2022). This paper further improves the H dependence from H^5 to H^4. On the lower bound side, this paper also provides a H^3 lower bound. Moreover, this paper discusses the difficulty to further bridge this gap.

For the technical part, this paper mostly builds on Chen et al. (2021) and Hu et al. (2022). More specifically, the paper uses the idea of enlarging the exploration bonus by a H factor in Chen et al. (2021) to improve the H dependency. In addition, the authors use the fine-grained control of the magnitude of w_{k,h} to improve the d dependency. Therefore, the algorithm design and the analysis might not be very novel. With that being said, putting them into a single piece and getting a final result is definitely a valuable contribution.

Regarding this part, I have a question. Hu et al. (2022) uses different Q-functions (Over-optimistic Q function, Optimistic Q function, Pessimistic Q function), but only a single function Q-function seems to be used in this paper. Can you comment on why you do not need them? My guess is that it might be related to a the worse in H factor in the upper bound.

Another question is that I feel usually a larger bonus term (beta) implies a larger regret since its sum will enter into the regret bound. But both this paper and Chen et al. (2021) mention that using the larger bonus instead reduces the regret. Probably this has been answered in the paper by Chen et al. (2021), but I would appreciate that if the authors can comment on that.

The proof of the lower bound part mostly follows Chen et al. (2021), and I can also share its intuition.

Finally, I would like to point out a few related works in the reward-free learning setting. [1] is about reward-free learning setting and similar as Zanette et al. (2020c) requires some additional assumptions. [2] can handle the more general low-rank MDPs (linear MDPs + unknown feature phi), but requires a reachability assumption. [3] can handle the general function assumption setting and subsumes linear or low-rank MDPs. There are also more related works in the more restricted block MDPs, e.g., [4] and the reference therein.

[1]Towards Deployment-Efficient Reinforcement Learning: Lower Bound and Optimality

[2]Model-free Representation Learning and Exploration in Low-rank MDPs

[3]On the Statistical Efficiency of Reward-Free Exploration in Non-Linear RL

[4]Efficient Reinforcement Learning in Block MDPs: A Model-free Representation Learning Approach

One minor question: This paper assumes the reward function is linear. Can you handle the case that the reward function is non-linear, but given in the planning phase (assuming that you know the reward function class or simply its size in the online phase)?

**Strength And Weaknesses:**

Strengths:

1. The LSVI-RFE algorithm is computationally efficient and improves the current state of the art bound in reward-free linear MDPs.

Weaknesses:

1. Some literature in reward-free learning could be added to the paper.

**Summary Of The Paper:**

This paper studies reward-free learning in linear MDPs. It proposes a computationally efficient algorithm LSVI-RFE and improves the upper bound to O(H^4d^2/eps^2). An Omega(H^3d^2/eps^2) lower bound is also provided.

**Summary Of The Review:**

Please see other blocks.

---

> ### Author Response · Authors · 2022-11-16
> **Response to Reviewer  tyVT**
>
> Thank you very much for your positive and valuable feedback! In the following, we respond to your comments in detail.
>
> ### Questions
> > Q1: Regarding this part, I have a question. Hu et al. (2022) uses different Q-functions (Over-optimistic Q function, Optimistic Q function, Pessimistic Q function), but only a single function Q-function seems to be used in this paper. Can you comment on why you do not need them? My guess is that it might be related to a the worse in H factor in the upper bound.
> - Optimistic Q function: We did build an optimistic Q function $\widehat{Q}\_h$ in the planning phase, which is the standard procedure of optimistic value iteration as existing works Wang et al. (2020a), Zhang et al. (2021).
> - Over-Optimistic Q function: Though we do not state "over-optimistic" in our paper explicitly, we use a similar over-optimism idea. We did build an over-optimistic Q function $\widehat{Q}\_{k,h}$ in the exploration phase ($\widehat{Q}\_{k,h}$ is over-optimistic due to a factor of "2" enlargement in the exploration bonus $b_{k,h}$). Thus, the monotonicity of value functions is built between the exploration phase value function and the planning phase value function, i.e., $V\_h^*(\cdot;r)+\hat{V}\_{k,h}(\cdot) \geq \hat{V}\_h(\cdot)$ for any $k \in [K]$.
> - Pessimistic Q function: As stated in Section 5.1 by Hu et al. (2022), the pessimistic Q function is used to build the variance estimator. Since we do not estimate the variance of the real value function concerning the actual reward function, the pessimistic Q function is not required. The obstacle for building a variance estimator so as to apply the Law of Total Variance (LTV) does result in an $H$ factor degragation in the upper bound.
>
> > Q2:  I feel usually a larger bonus term (beta) implies a larger regret since its sum will enter into the regret bound. But both this paper and Chen et al. (2021) mention that using the larger bonus instead reduces the regret. Probably this has been answered in the paper by Chen et al. (2021), but I would appreciate that if the authors can comment on that.
>
> Our paper shows that the exploration phase bonus $b\_{k,h}(\cdot,\cdot)=2\widehat{\beta}^E\\|\boldsymbol{\phi}(\cdot, \cdot)\\|\_{\widehat{\mathbf{\Lambda}}\_{k,h}^{-1}}$ (with exploration phase radius $\widehat{\beta}^E=\widetilde{O}(d\sqrt{H})$ ) and the planning phase bonus $b\_{h}(\cdot,\cdot)=\min\\{\widehat{\beta}^P\\|\boldsymbol{\phi}(\cdot, \cdot)\\|\_{\widehat{\mathbf{\Lambda}}\_{K+1,h}^{-1}},H\\}$ (with planning phase radius $\widehat{\beta}^P=\widetilde{O}(\sqrt{Hd})$) can take different orders, as revealed in Eq. (9) and Eq. (10). In particular, as shown in Step 4 in Section 5, only the planning phase radius $\widehat{\beta}^P$ determines the sharpness of the sample complexity. That's why our paper takes a larger-order exploration radius $\widehat{\beta}^E$ but achieves a better sample complexity since the exploration radius $\widehat{\beta}^E$ does not determine the final sample complexity.
>
> > Q3: This paper assumes the reward function is linear. Can you handle the case that the reward function is non-linear, but given in the planning phase (assuming that you know the reward function class or simply its size in the online phase)?
>
> Our algorithm cannot handle arbitrary reward functions. However, apart from the linear function class, some other function classes are likely to be handled by our techniques. In particular, our algorithm is likely to handle function classes which are subsets of a vector space with $d^2$ dimensions, i.e., $\\{r\_h(s,a): h \in [H], s \in S, a \in A\\} \subset V$, where $V$ is a vector space and $\dim V = d^2$. This is because the algorithm is only estimating the transition probability in the exploration phase, and we do not make any structural assumption concerning the reward function given in the planning phase, except that we need to build the covering net argument to analyze the constructed value function $\widehat{V}_h$ in planning phases (detailed in Lemma A.12 in Appendix).
>
> ### Weaknesses
> > W1: Finally, I would like to point out a few related works in the reward-free learning setting.
>
> Thanks for your helpful comments and suggestions, we have included all the mentioned works above and other related works in our revision.
> ***
> ***Finally, we want to thank the reviewer again for the detailed and valuable comments. If our response resolves your concerns satisfactorily, we want to kindly ask the reviewer to consider raising the score rating of our work. Please let us know if you have any further questions, and we will be happy to answer them.***

---

> > ### Comment · Reviewer_tyVT · 2022-11-26
> > **6 to 8**
> >
> > Thank you for your response! I raised my score from 6 to 8.

---

> > > ### Author Response · Authors · 2022-11-26
> > > **Thanks**
> > >
> > > Thank you very much for your help in improving our paper! We really appreciate your valuable comments and positive response.

---

> ### Author Response · Authors · 2022-11-22
> **Reminder to Reviewer tyVT**
>
> Dear reviewer,
>
> Thank you for your effort in reviewing our paper!
>
> We wonder whether our reply fully addresses your concerns. If so, could you please consider raising your score for our work?
> Please let us know if you have any further questions. We will be more than happy to discuss this with you and answer any remaining questions.
>
> Thank you very much!

---

### Author Response · Authors · 2022-11-16
**Revision**

We thank all reviewers for the insightful comments and helpful suggestions. We have revised the paper accordingly, with revised parts marked in blue.
- We have highlighted our technical contributions in Section 4, especially compared with prior works of Hu et al. (2022), Chen et al. (2021) and Wagenmaker et al. (2022).
- We have explained our algorithm design better, i.e. we use more straightforward sentences instead of involved explanations in Section 4.
- We provide a detailed analysis of the computational traceability of our LSVI-RFE algorithm in Appendix D.

---

### Decision · Program_Chairs · 2023-01-20

**Decision:**

Accept: poster

**Justification For Why Not Higher Score:**

The paper is about a minor improvement on a relatively well-studied theoretical problem and is unlikely to be of broad interest to the ICLR community.

**Justification For Why Not Lower Score:**

N/A.

**Metareview: Summary, Strengths And Weaknesses:**

This paper presents an algorithm for reward-free exploration in linear MDPs reaching a minimax optimal sample complexity up to an $H$ and logarithm factor. A number of new analysis tools have been introduced and can be of independent interest. All reviewers lean unanimously toward acceptance so I will recommend so as well.

**Note From Pc:**

if the above contains the word "oral" or "spotlight" please see: "oral" presentation means -> notable-top-5% and "spotlight" means -> notable-top-25%. As stated in our emails, we are disassociating presentation type from AC recommendations